

# Folding and necking across the scales: a review of theoretical and experimental results and their applications

Stefan Markus Schmalholz[1] and Neil Mancktelow[2]

[1]Institute of Earth Sciences, University of Lausanne, Switzerland

[2]Department of Earth Sciences, ETH Zurich, Switzerland

*Correspondence to*: S.M. Schmalholz (stefan.schmalholz@unil.ch)

**Abstract.** Shortening and extension of mechanically-layered ductile rock generates folds and pinch-and-swell structures (also referred to as necks or continuous boudins), which result from mechanical instabilities termed

folding and necking, respectively. Folding and necking occur in layered rock because the corresponding mechanical work involved is less than that associated with a homogeneous deformation. The effective viscosity of a layered rock decreases during folding and necking, even when all material parameters remain constant. This mechanical softening due to viscosity decrease is solely the result of fold and pinch-and-swell structure development and is hence termed structural softening (or geometric weakening). Folding and necking occur over the whole range of

geological scales, from microscopic up to the size of lithospheric plates. Lithospheric folding and necking are evidence for significant deformation of continental plates, which contradicts the rigid-plate paradigm of plate tectonics. We review here some theoretical and experimental results on folding and necking, including the lithospheric scale, together with a short historical overview of research on folding and necking. We focus on theoretical studies and analytical solutions that provide the best insight into the fundamental parameters controlling

folding and necking, although they invariably involve simplifications. To first order, the two essential parameters to quantify folding and necking are the dominant wavelength and the corresponding maximal amplification rate. This review also includes a short overview of experimental studies, a discussion of recent developments involving mainly numerical models, a presentation of some practical applications of theoretical results, and a summary of similarities and differences between folding and necking.



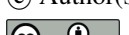

## 1    Introduction

About 200 years ago Sir James Hall made his famous analogue experiments on layer-parallel shortening of
linen and woollen cloth layers with vertical confining pressure provided by "a door (which happened to be off the
hinges)" loaded with weights (Hall, 1815; Figure 1). During shortening the layers deflected laterally (orthogonal to
the shortening direction), which is a process termed buckling (mainly used for elastic material) or folding (mainly
used for viscous material; Figure 2A, B). James Hall was probably the first to show that natural folds in rock are the
result of a horizontal compression. However, some of the first scientific observations on folds were already made
more than 100 years earlier, by Marsili and Scheuchzer in 1705, in the European Alps around Lake Uri in
Switzerland (see Hantke, 1961; Ellenberger, 1995; Vaccari, 2004; Figure 3A). If a competent layer is extended,
rather than shortened, it does not usually deflect but either breaks (brittle or fracture boudinage) or locally thins in a
ductile manner (necking; Figure 2C, D). The term boudinage goes back to Lohest (1909) but necking structure
(often termed pinch-and-swell structure) was probably first described by Ramsay in 1866 (Ramsay, 1866; Cloos,
1947; Lloyd, et al., 1982; Figure 3B). The finite strain deformation geometry of a competent layer is thus
fundamentally different for layer-parallel shortening and extension (Figure 4), although the initial stages of both
folding and necking instabilities can be mathematically explained with the same theory (e.g. Smith, 1975, 1977).
The mechanical processes controlling the different behaviour of competent layers under compression and extension
are the main focus of this review.

Folding and necking are processes that result from instabilities in elastic, plastic and viscous material caused
by layer-parallel compression and extension, respectively, of mechanically competent layers. In this review, the
overall deformation behaviour is assumed to be ductile and continuous so that fracturing does not play any
controlling role. However, after some amount of ductile necking a layer often fractures around the necked region,
which is a process termed ductile fracture (e.g. Dieter, 1986), and necking can act as a ductile precursor to brittle
boudinage. It is also possible in nature that brittle precursors can localize shearing (Segall and Simpson, 1986;
Mancktelow and Pennacchioni, 2005; Pennacchioni and Mancktelow, 2007) and hence trigger the formation of
pinch-and-swell structure (Gardner et al., 2015).

The structures resulting from folding and necking can have a wide variety of different geometries, especially in
multilayers (Ramberg, 1955; Ramsay and Huber, 1987; Price and Cosgrove, 1990; Goscombe et al., 2004; Figure 5).
As noted by Ramsay and Huber (1987), "folds are perhaps the most common tectonic structure developed in

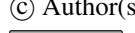



deformed rocks" and a thorough understanding of folding is therefore essential to understand the deformation of

typically layered or foliated rocks. Folds can also be generated by passive flow or bending but here we focus only on

folds resulting from mechanical instability due to layer shortening. Pinch-and-swell structure seems to be less

frequent in nature than folds and this review will provide potential reasons to explain this observation. Folding and

necking (and also brittle boudinage) can occur simultaneously, with necks and/or boudins commonly forming in the

limbs of folds (Ramberg, 1959; Figure 5). In nature, folding and necking are always three-dimensional (3D)

processes (Figure 6), but most theoretical results are based on 2D models.

Folding and necking are important tectonic processes because they occur over the whole range of geological

scales, from microscopic dimensions up to the size of lithospheric plates. Lithospheric folding and necking in

tectonic plates contradict the paradigm of plate tectonics *sensu strictu*, which states that tectonic plates are rigid and

deformation only occurs at plate boundaries. Lithospheric folding can occur on a length scale of several thousands

of kilometres, such as in Central Asia (Burov et al., 1993; Figure 7A). Lithospheric necking is also important, for

example, in the formation of rift basins (Zuber and Parmentier, 1986) and of magma-poor passive continental

margins, because many of these margins are characterised by so-called necking domains in which the crustal

thickness decreases from normal crustal thickness (30-35 km) to ca. 5-10 km (Sutra, et al., 2013). Lithospheric

necking can also generate crustal-scale pinch-and-swell structure (Fletcher and Hallet, 1983; Gueguen et al., 1997;

Figure 7B) and is a first-order process during slab detachment (Lister et al., 2008; Schmalholz, 2011; Duretz et al.,

2012; Bercovici et al., 2015).

The literature on the mechanics of folding and necking is vast because these processes (i) occur from mm to

km scale, (ii) were studied for a variety of constitutive equations (rheologies) such as elastic, plastic, viscous,

viscoelastic or viscoelastoplastic, (iii) can be driven by imposed boundary displacements, velocities, or stresses, or

by gravity, (iv) were studied for different bulk deformation geometries such as pure shear, simple shear or

wrenching, (v) were studied for single- or multilayer configurations, (vi) were studied for isotropic and anisotropic

materials, (vii) were studied in 2D and 3D and (viii) were studied using analytical solutions, laboratory (analogue)

experiments, or numerical simulations. We present here only a small selection of studies and results. Further

information on the mechanics of folding and necking in rock can be found in textbooks (Price and Cosgrove, 1990;

Johnson and Fletcher, 1994; Pollard and Fletcher, 2005) and in other review articles (Hudleston and Treagus, 2010;

Cloetingh and Burov, 2011).



We focus on studies that investigated particular mechanical aspects of folding and necking and on studies that applied the results to geological observations and problems. Particular geological questions concerning folding and necking are, for example, (i) which parameters control the observed geometry of folds and pinch-and-swell

structures, (ii) how much shortening or extension is required to generate the observed finite (high) amplitude folds and pinch-and-swell structures, (iii) does folding and necking change the overall (effective) strength of a rock unit, and (iv) how much force or stress is required to generate observed folds and necks, particularly on the lithospheric scale?

## 2 Folding

### 2.1 Theoretical results

Theoretical studies and analytical solutions invariably involve simplifications but can provide the best insight into the fundamental parameters controlling a mechanical process. The aim of such studies is thus to determine the (hopefully) small number of (non-dimensionalized) controlling parameters and to investigate their specific

influence.

#### 2.1.1 Single-layer folding

Some of the first mathematical studies on bending of beams were performed by Galilei (1638) who studied the strength of beams under beam-orthogonal loading (Figure 8A). A beam is a 2D layer which is much longer than thick and a number of simplifications can therefore be made for the geometrical description of the bending. A first

beam theory was developed in the 18[th] century with major contributions from Euler and Bernoulli and is hence often termed the Euler-Bernoulli beam theory (see Timoshenko, 1953; Szabo, 1987). It is assumed that the central (neutral) line in the beam is neither extended nor shortened and that the inner side of the beam is shortened while the outer side is extended, that is, there are both layer-parallel extensional and compressional strains in the beam due to bending. A major result of the Euler-Bernoulli beam theory is that it can relate the layer-parallel strain, $\varepsilon_{xx}$, due to

bending of the beam to the amplitude, $A$, of the deflection of the beam (Figure 9)

$$\varepsilon_{xx} = -y \frac{d^2 A}{dx^2} \tag{1}$$

In this equation, $y$ is the orthogonal (vertical) coordinate measured from the middle line of the beam and $x$ is the coordinate along the beam (Figure 9). The second spatial derivative represents the curvature of the beam. For elastic



material, the total layer-parallel stress due to bending is $\sigma_{xx} = K\varepsilon_{xx}$, where $K$ is a material property. For example, in

a modern plane stress formulation, it corresponds to $E/\left(1-\upsilon^2\right)$, with $E$ being Young's modulus (with units of Pa)

and $\upsilon$ being Poisson's ratio (dimensionless). Frequently used symbols with a consistent meaning throughout the text

are listed in Table 1.

The bending moment associated with the flexure of the beam is

$$M = \int_{-H/2}^{H/2} y\sigma_{xx}dy = -\int_{-H/2}^{H/2} y^2 K \frac{d^2A}{dx^2}dy = -\frac{KH^3}{12}\frac{d^2A}{dx^2} = -D\frac{d^2A}{dx^2} \qquad (2)$$

where $H$ is the thickness of the beam and $D$ is usually termed the flexural rigidity. If the beam is also compressed,

then a horizontal compressional load, $F$, is present within the layer (with units of N/m in 2D). Euler (1744) solved

this first buckling problem by equating the flexural moment due to bending of the beam, $M$, to the moment caused

by compression of the deflected beam, $FA$ (Fig. 9), to give

$$D\frac{d^2A}{dx^2} + FA = 0 \qquad (3)$$

Euler's famous result for the smallest load for which the beam buckles, the so-called Euler load (see also, for

example, Bazant and Cedolin, 1991), is then

$$F_E = \left(\frac{2\pi}{L}\right)^2 D \qquad (4)$$

The length, $L$ (or wavelength), of the deflection is controlled by the initial length of the beam, assuming that both

ends of the beam are fixed in vertical and horizontal position but can rotate freely. For loads smaller than $F_E$ the

beam does not buckle, assuming that it is initially perfectly straight.

Equation (3) describes the balance of moments acting on a compressed beam that can deflect freely because

the beam is not embedded in another material and gravitational stresses arising due to the deflection are also not

considered. If there is an additional stress, $q$, which resists the vertical deflection of the beam, either due to an

embedding medium or gravity, then Eq. (3) has to be expanded to (e.g. Smoluchowski, 1909; Biot, 1961)

$$\underbrace{D\frac{\partial^4 A}{\partial x^4}}_{\substack{\text{Horizontal stress} \\ \text{due to bending}}} + \underbrace{F\frac{\partial^2 A}{\partial x^2}}_{\substack{\text{Horizontal stress} \\ \text{due to compression}}} + \underbrace{q}_{\substack{\text{Vertical stress due to} \\ \text{embedding medium} \\ \text{and/or gravity}}} = 0 \qquad (5)$$





with the convention here, opposite to Biot (1961), that $q$ is positive when acting against the positive direction of the deflection $A$. Equation (5) represents an approximate force balance equation in 2D and the derivation of this approximate equation from the full set of 2D force balance equations is given in Appendix 1. In Eq. (5), $D$ is assumed constant and all terms (summands) have units of Pa. In a geological context, the above equation was

probably first solved by Smoluchowski (1909), who considered crustal folding and used $q = \rho g A$, where $\rho$ is the crustal density and $g$ the gravitational acceleration (see section 2.1.3 on lithospheric folding). The results of Smoluchowski were re-derived by Goldstein (1926), who also applied them to folding of the Earth's crust. In engineering applications, the resistance of an elastic medium below the layer (often termed the foundation) commonly has the form $q = kA$, where $k$ is termed an elastic foundation modulus. The resistance of an elastic

foundation is hence directly proportional to the deflection, $A$. The terms describing the resistance of an elastic foundation and the resistance due to gravity are mathematically identical if we identify $k$ with $\rho g$.

Equation (5) is a 1D equation that can be used to study 2D folding (buckling) because of the assumption that the horizontal strain can be expressed by the second spatial derivative of the vertical deflection, $A$ (Eq. (1)). Many studies on folding, involving elastic, viscous and viscoelastic material (see below), are based on this

simplified equation and we refer to the approach of using such a simplified "beam equation" or "thin-plate equation" as the thin-plate approach. The thin-plate approach can also be applied to 3D folding of plates. The thin-plate equations for 2D and 3D have been continuously developed and derived in the 18[th], 19[th] and 20[th] centuries (Timoshenko, 1953; Szabo, 1987). The derivation of a general thin-plate theory applicable for 3D buckling of plates was finalized by Kirchhoff (1855).

Sander (1911) observed that, for multilayers within a quartz phyllite, the size of individual folds is related to their layer thickness and that folds become systematically smaller as their layer thickness becomes thinner (e.g., Figure 10). He termed this observation the "law of fold size" (in original German: Gesetz der Stauchfaltengrösse). Several authors applied variations of Eq. (5) to folded rocks to derive formulas for the size or wavelength, $L$, of folds (e.g. Gunn, 1937; Kienow, 1942). These studies applied a foundation term, $q$, which is proportional to $A$ but

independent of the wavelength, $L$, of the deflection. Biot (1937) showed that if the embedding medium is more correctly considered as a two-dimensional elastic half-space, then the stress resistance of the embedding medium is

$$q = b\frac{2\pi}{L}A \qquad (6)$$

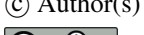



where $b$ is here a material parameter. The importance of this result is that the resistance of the embedding medium not only depends on the amplitude of the deflection but also on the wavelength of the deflection (Figure 11).

Based on Biot's correspondence principle (1954a, 1956a – see Biot, 1961), the thin-plate equation for elastic material can be easily applied to viscous and viscoelastic materials (Biot, 1957). For incompressible viscous material, the horizontal total stress to calculate the moment, $M$, is $\sigma_{xx} = 4\eta D_{xx}$, where $\eta$ is the viscosity and the strain rate due to bending, $D_{xx}$, can be calculated from the time derivative of Eq. (1)

$$D_{xx} = \frac{d\varepsilon_{xx}}{dt} = -y \frac{d^3 A}{dx^2 dt} \tag{7}$$

The factor 4 for the total horizontal stress ( $\sigma_{xx} = -P + 2\eta D_{xx}$ with $P$ being the pressure) arises because it is assumed that the layer can expand freely vertically, so that $\sigma_{yy} = -P + 2\eta D_{yy} = 0$, which yields $P = 2\eta D_{yy}$. Using the incompressibility condition $D_{yy} = -D_{xx}$ yields $P = -2\eta D_{xx}$ and $\sigma_{xx} = 4\eta D_{xx}$ (Biot, 1961). From $\sigma_{xx} = 4\eta D_{xx}$, it also follows that the horizontal load is $F = 4\eta \bar{D}_{xx} H$, where $\bar{D}_{xx}$ is the bulk rate of shortening (i.e. the shortening between the two ends of the layer, indicated with an overhead bar). In passing, it should be noted that there is always

a "tectonic" or "dynamic" overpressure in the shortening layer that is directly proportional to the viscosity $\eta$ and bulk shortening rate $\bar{D}_{xx}$ in the layer ( $P = -2\eta \bar{D}_{xx}$; e.g. Mancktelow, 1993, 2008). For extension of the layer, the sign is simply reversed and there is an underpressure of equal magnitude, an observation that is relevant when considering why boudins often show more brittle behaviour. Assuming that the viscous layer is embedded in a viscous half-space, the net resistance of the embedding medium acting on the layer $q = -4\eta_M \frac{2\pi}{L} \frac{dA}{dt}$ (Biot, 1961;

see Appendix 2) with $\eta_M$ being the viscosity of the embedding medium. The thin-plate equation for folding of a viscous layer embedded in a viscous medium is then (Biot, 1961)

$$-\frac{\eta H^3}{3} \frac{\partial^5 A}{\partial x^4 \partial t} + 4\eta \bar{D}_{xx} H \frac{\partial^2 A}{\partial x^2} - 4\eta_M \left(\frac{2\pi}{L}\right) \frac{\partial A}{\partial t} = 0 \tag{8}$$

In direct analogy to Eq. (5), the first term corresponds to the stress due to bending of the layer, the second to the stress due to compression, and the third to stresses due to the resistance of the surrounding medium. To solve the

above partial differential equation (PDE) one assumes a solution of the form (Biot, 1961)

$$A = A_0 \exp\left(-\alpha \bar{D}_{xx} t\right) \cos\left(\frac{2\pi}{L} x\right) \tag{9}$$



for which the derivative with respect to time, $t$, and coordinate, $x$, can be easily taken. The parameter $\alpha$ is the non-dimensional dynamic amplification rate of any initial deflection $A_0$ (the kinematic component of amplification is ignored in the thin-plate approach; see below) and the minus sign in the exponent accounts for the convention that

mathematically a shortening strain rate is negative (Figure 11). Substituting (9) into (8) and taking the derivatives transforms Eq. (8) into an algebraic equation of the form

$$\frac{\eta H^3}{3}\alpha \bar{D}_{xx}\left(\frac{2\pi}{L}\right)^4 A - 4\eta \bar{D}_{xx}H\left(\frac{2\pi}{L}\right)^2 A + 4\eta_M\left(\frac{2\pi}{L}\right)\alpha \bar{D}_{xx}A = 0 \tag{10}$$

which can be simplified by introducing $s = 2\pi H / L$ (the non-dimensional wavenumber) and $R = \eta / \eta_M$ (the viscosity ratio) to


$$\frac{R}{3}\alpha s^3 - 4Rs + 4\alpha = 0 \tag{11}$$

The amplitude, $A$, the bulk strain rate, $\bar{D}_{xx}$, and one term $s$ could be dropped from (10) because they are multiplied with every term in the equation. The algebraic Eq. (11) provides an expression for $\alpha$ as function of $L$ (within $s$)

$$\alpha = \frac{1}{\dfrac{s^2}{12} + \dfrac{1}{sR}} \tag{12}$$

As seen from plots of this relation in Figure 12, all wavelengths are amplified but there is a wavelength, the so-

called dominant wavelength, $L_d$, for which $\alpha$ is at a maximum. The value of $L_d$ can be found by taking the derivative of $\alpha$ with respect to $L$, setting this derivative to zero and solving for $L$, which yields

$$L_d = 2\pi H\left(\frac{R}{6}\right)^{\frac{1}{3}} \tag{13}$$

from which it is immediately obvious that the dominant wavelength is directly proportional to the layer thickness and increases with increasing viscosity ratio of layer to matrix (Figure 12). This famous dominant wavelength

expression was first derived by Biot (1957) using the thin-plate approach described above. The maximum growth rate corresponding to $L_d$ is found by substituting (13) into (12) and is

$$\alpha_d = \left(\frac{4}{3}R\right)^{2/3} \tag{14}$$





This value also increases with the viscosity ratio. Eq. (14) already indicates that the thin-plate approach is inaccurate

for small values of $R$ because $\alpha_d \approx 1.21$ for $R=1$ but for $R=1$ (no viscosity difference between layer and matrix)

the amplification rate should be zero (since passive layer thickening is not considered in the thin-plate approach).

For comparison with a viscous layer, the dominant wavelength for an elastic layer embedded in a linear

viscous medium is $L_d = 2\pi H \left( G / \bar{\sigma}_e \right)^{1/2}$ and the corresponding maximal amplification rate is

$\alpha_d = \bar{\sigma}_e \left( \bar{\sigma}_e / G \right)^{1/2} / \left( 6\eta_M \right)$, where $G$ is the shear modulus of the layer and $\bar{\sigma}_e$ is the elastic compressive stress in the

layer (Biot, 1961; Schmalholz and Podladchikov, 2001a). For an elastic layer, the dominant wavelength is

independent of the viscosity of the embedding medium. If the layer is viscoelastic and its rheology is described by a

Maxwell model (i.e. elastic and viscous element connected in series), then the layer behaves as effectively viscous

for $\lambda < 1$ and effectively elastic for $\lambda > 1$, whereby $\lambda$ is the ratio of the dominant wavelength for a viscous layer to

the dominant wavelength for an elastic layer, that is $\lambda = \left( R / 6 \right)^{1/3} \left( \bar{\sigma}_e / G \right)^{1/2}$ (Schmalholz and Podladchikov, 1999).

If the viscoelasticity of the layer is described by a viscous and elastic element connected in parallel (Kelvin model),

then the layer behaves as effectively viscous for $\lambda > 1$ and effectively elastic for $\lambda < 1$ (Schmalholz and

Podladchikov, 2001b).

As discussed in some detail by Biot (1961), the selectivity of amplification, and therefore the tendency to

develop a clear sinusoidal form with a wavelength approximating that of the dominant wavelength, depends on the

relative bandwidth of the amplification rate curve, which he defined as the wavelength difference at half the

maximum amplification rate $\Delta L$ divided by $L_d$ (Fig. 6 in Biot, 1961). He found that

$$\frac{\Delta L}{L_d} \approx \frac{1.36}{\sqrt{\log\left(\alpha_d\right)}} \tag{15}$$

and that the selectivity therefore depends only on the maximum amplification rate (and thus the viscosity ratio) but

that this dependence is relatively weak, as can be qualitatively seen in Figure 12. At lower viscosity ratios (e.g. 15 in

Figure 12A), fold trains will be more irregular and the initial perturbation geometry will have an increasing

influence (Abbassi and Mancktelow, 1990, 1992; Mancktelow and Abbassi 1992; Mancktelow, 2001). It should be

noted that the analysis presented above is specifically for infinitesimal amplitudes, so that the dominant wavelength

and amplification rate derived represent initial values when the fold amplitude is very small.

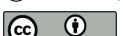



A different approach to derive $L_d$ was presented by Ramberg (1962). He considered the full 2D force

balance equation of a viscous fluid and solved the equations with a stream function approach (see Appendix 2). The

stream function approach provides the complete 2D velocity and stress fields for a viscous layer embedded in a

viscous medium. From the stress field, Ramberg calculated the horizontal force required to fold the layer as a

function of the wavelength and determined the wavelength for which this force is minimal. The stream function

approach provides an expression for $L_d$ that is identical to the one of Eq. (13) (Ramberg, 1962). Ramberg (1962), in

his Eq. 32, has a factor 4 in the expression for $L_d$, because he used the half-layer thickness for $H$.

The stream function approach was later used by Fletcher (1974) to perform a hydrodynamic stability

analysis for single-layer folding of a power-law viscous layer. Detailed mathematical descriptions of the stream

function approach in combination with a hydrodynamic stability analysis are given in Fletcher (1974, 1977), Smith

(1975, 1977), Johnson and Fletcher (1994) and Pollard and Fletcher (2005). We term this approach here stability

analysis, but this approach has also been termed perturbation method or thick-plate analysis. We do not use the term

thick-plate analysis to avoid confusion with the thick- or shear-deformable plate theory, which is an elaboration of

the thin-plate equation that considers also shearing within the layer (e.g. Wang, et al., 2000). As will be shown later,

the same stability analysis can also be applied to study necking (Smith, 1975, 1977). The stability analysis assumes

that geometrical perturbations are superposed on a flat layer that is shortening and thickening by pure shear (the

basic state). In a stability analysis, the initial deflection, $A_0$, of the thin-plate approach corresponds to the amplitude

of a sinusoidal geometrical perturbation superposed on a flat layer boundary. The stability analysis provides an

expression for the time derivative of $A$ of the general form

$$\frac{dA}{dt} = -[1+\alpha]\bar{D}_{xx}A \qquad (16)$$

where the negative sign accounts for the convention that shortening strain rates are negative. In the stability analysis,

the kinematic (or passive) amplification velocity due to the basic state pure shear thickening is taken into account

and is given by $-\bar{D}_{xx}A$. This corresponds to a passive growth rate of 1, which is added to the dynamic growth rate

$\alpha$ within the square brackets of Eq. (16). Kinematic amplification is neglected in the thin-plate approach because

the layer thickness is assumed to be constant during folding. The exact infinitesimal amplitude solution from

stability analysis for the dynamic amplification rate $\alpha$ is given by



$$\alpha = \frac{-2\left(1 - R^{-1}\right)}{\left(1 - R^{-2}\right) - \left[\left(1 + R^{-1}\right)^2 e^s - \left(1 - R^{-1}\right)^2 e^{-s}\right]/2s} \tag{17}$$

(Fletcher, 1974, 1977; Smith, 1977), where $s$ is again the non-dimensional wavenumber $s = 2\pi H / L$ as defined

above. A so-called closed form solution for the dominant wavelength cannot be derived because the mathematical

expression in Eq. (1.17) is just too complicated. The above expression for $\alpha$ can be simplified by performing a

Taylor expansion for $L / H \gg 1$ and keeping only dominating terms in $L / H$. The resulting expression for $\alpha$ is

identical to the one derived by the thin-plate approach (Eq. (12)) and of course the dominant wavelength is therefore

also identical. In Figure 12, the exact solution of Eq. (17) is plotted in comparison to the simplified thin-plate result,

and it is seen that the thin-plate approximation always overestimates the amplification rate, the dominant wavelength

is shorter, and the thin-plate solution does not tend to a dynamic amplification rate of zero as the viscosity ratio

approaches 1, as it clearly should in reality. The thin-plate solution becomes a better approximation with increasing

viscosity ratio.

Fletcher (1974) also derived a solution for $\alpha$ for power-law viscous fluids, which is given by his Eq. (8) as

$q(k)$ in that publication. The general solution for the amplification rate of folding, and also for necking, of power-law

viscous layers embedded in power-law viscous medium is (Smith, 1977; Pollard and Fletcher, 2005)

$$\alpha = \theta\left[-1 + \frac{2n\left(1 - \frac{1}{R}\right)}{1 - Q^2 + \theta \frac{\sqrt{n-1}}{2\sin(\beta k)}\left(1 + Q^2\right)\left(e^{as} - e^{-as}\right) + 2Q\left(e^{as} + e^{-as}\right)}\right] \tag{18}$$

$$\theta = sign\left(\bar{D}_{xx}\right), \quad a = \sqrt{\frac{1}{n}}, \quad \beta = \sqrt{1 - \frac{1}{n}}, \quad s = \frac{2\pi H}{L}, \quad Q = \sqrt{\frac{n}{n_M}}\frac{1}{R}$$

For folding the signum function of the bulk shortening rate $sign\left(\bar{D}_{xx}\right) = \theta = -1$ and for necking $\theta = 1$. Plots of the

amplification rate against $L/H$ in Figure 13 show that, in comparison to the linear viscous case, the amplification rate

is greater, the dominant wavelength is shorter and the selectivity is greater (i.e. narrower curve). In the exact power-

law infinitesimal amplitude solution, there is oscillation in the amplification rate curve as $L/H$ approaches zero, as

was discussed by Johnson and Fletcher (1994) with regard to their Fig. 8.2. Numerical models determining the

amplification rate of folding in power-law viscous materials, as can be carried out very easily with the Folder

package of Adamuszek et al. (2016), reproduce this oscillating growth curve (Fig. 13C) and it is not an artefact.

However, considering the very low amplification rates (in part negative) and normalized wavelengths, it has little





practical influence on fold development in power-law materials. The dominant wavelength for folding of a power-law viscous layer and the corresponding maximal amplification rate depend on both the viscosity ratio and the power-law stress exponent (Figure 14). The lower limit for the dominant wavelength to thickness ratio is ~4 (Figure

14).

With power-law viscous rheology, the approximate formula (for $L/H \gg 1$) for the dominant wavelength is

$$L_d \approx 2\pi H \left( \frac{R}{6} \frac{n_M^{1/2}}{n} \right)^{\frac{1}{3}} \tag{19}$$

where $n$ and $n_M$ are the power-law stress exponents of the layer and matrix, respectively. The approximate maximum

value of $\alpha$ corresponding to $L_d$ is (Fletcher, 1974)

$$\alpha_d \approx 1.21 \left( n\, n_M \right)^{1/3} R^{2/3} \tag{20}$$

A power-law viscous behaviour of the layer and/or the embedding medium ($n$ and/or $n_m > 1$) hence increases the amplification rate and consequently the folding instability. For a power-law viscous flow law such as $\tau_{xx} = B D_{xx}^{1/n}$ ($B$ is a material property), the effective viscosity (ratio of stress to strain rate) can be expressed as a function of the

square root of the second invariant of the strain rate tensor, $D_{II}$, (Johnson & Fletcher, 1994)

$$\eta = \eta_C \left( D_{II} \right)^{\frac{1}{n}-1} \tag{21}$$

A viscosity formulation with an invariant is required so that the flow law is independent of the chosen coordinate system (so-called material objectivity). For 2D incompressible flow the strain rate invariant reduces to

$D_{II} = \sqrt{D_{xx}^2 + D_{xy}^2}$ (e.g. Johnson & Fletcher, 1994). Using Eq. (21), the power law viscous flow law is then

$\tau_{xx} = 2\eta D_{xx}$ where $\eta_C = B/2$. The coefficient $\eta_C$ is not a viscosity because it has units Pas$^{1/n}$ and not Pas. It is useful to reformulate the above equation and to scale $D_{II}$ by the absolute value of the bulk shortening rate, $\left| \bar{D}_{xx} \right|$,

$$\eta = \eta_C \left( \left| \bar{D}_{xx} \right| \right)^{1-\frac{1}{n}} \left( \frac{D_{II}}{\left| \bar{D}_{xx} \right|} \right)^{\frac{1}{n}-1} = \eta_R \left( \frac{D_{II}}{\left| \bar{D}_{xx} \right|} \right)^{\frac{1}{n}-1} \tag{22}$$

The parameter $\eta_R$ is a viscosity with units Pas and is the reference viscosity corresponding to a homogeneous pure shear with a constant strain rate corresponding to $\left| \bar{D}_{xx} \right|$, since for this homogeneous pure shear rate $D_{II} = \left| \bar{D}_{xx} \right|$





(Figure 15). The viscosities used in the formulas for the dominant wavelength and corresponding amplification rates

for power-law viscous material are typically the reference viscosities corresponding to the basic state of

deformation, which is typically a homogeneous pure shear. Due to folding, strain rates deviate locally from the bulk

shortening rate and cause local variation in the effective viscosity. Strain rates higher than the basic state strain rates

cause a decrease in the effective viscosity and strain rates smaller than the basic state rate an increase in effective

viscosity.

Stability analysis is performed with linear equations and hence the nonlinear power-law flow law must be

linearized. This is done by assuming that every quantity, such as strain rate (e.g. $D_{xx}$) or deviatoric stress (e.g. $\tau_{xx}$)

can be expressed as the sum of a quantity representing the basic, pure shear state of deformation (e.g. $\bar{D}_{xx}$ and $\bar{\tau}_{xx}$)

and a quantity representing the perturbation (deviation) flow from the basic state of pure shear (e.g. $\tilde{D}_{xx}$ and $\tilde{\tau}_{xx}$),

that is, for example $D_{xx} \approx \bar{D}_{xx} + \tilde{D}_{xx}$ (Figure 15). The nonlinear equations are then linearized by performing a Taylor

expansion around the basic state, keeping only those terms that are linear in the perturbation quantities. The resulting

flow laws for the perturbed, horizontal and shear stresses are then (e.g. Fletcher, 1974; Smith, 1977; Fletcher and

Hallett, 1983; Johnson and Fletcher, 1994; Pollard and Fletcher, 2005)

$$\tilde{\tau}_{xx} = 2\frac{\eta_R}{n}\tilde{D}_{xx}$$
$$\tilde{\tau}_{xy} = 2\eta_R\tilde{D}_{xy}$$

(23)

where $\eta_R$ is the reference viscosity for the basic state of pure shear deformation given in Eq. (22). The flow laws for

the perturbing flow, that is the deviation from pure shear, are thus anisotropic, because the viscosity for the normal

stresses (but not the shear stresses) is divided by $n$, which is a result of the linearization (Taylor expansion). This

can also be seen qualitatively in Figure 15, reproduced in a modified form after Fig. 3 of Smith (1977). The implicit

anisotropy in power-law viscous materials means that the effective viscosity for normal (perturbing) strains is

smaller than for (perturbing) shear strains. The anisotropy occurs because the second strain rate invariant,

$D_{II}^2 = D_{xx}^2 + D_{xy}^2$, which controls the effective viscosity (Eq. (21)) is only sensitive to perturbations in normal strain

rate $\tilde{D}_{xx}$ but not to shear strain rates $\tilde{D}_{xy}$. Using the assumptions $D_{xx} \approx \bar{D}_{xx} + \tilde{D}_{xx}$ and $D_{xy} \approx \bar{D}_{xy} + \tilde{D}_{xy}$ in $D_{II}^2$ and

performing a multiple Taylor expansion for small $\tilde{D}_{xx}$ and $\tilde{D}_{xy}$ yields $\bar{D}_{II}^2 = \bar{D}_{xx}^2$ and $\tilde{D}_{II}^2 = 2\bar{D}_{xx}^2\tilde{D}_{xx}^2$ because

$\bar{D}_{xy} = 0$ for the applied pure shear basic state of deformation (Smith, 1977). The anisotropy in the perturbing flow is

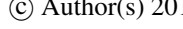

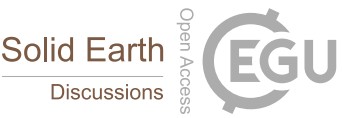

responsible for the difference between linear viscous and power-law viscous folding (and also necking; Smith,

1977). A detailed derivation of the linearization of the power law flow law and the separation into basic state and

perturbing flow can be found in Pollard and Fletcher (2005; their section 11.2.3).

   A note on nomenclature: for linear viscous fluids ($n = 1$) the stress continuously increases with increasing

strain rate and the viscosity is constant. For power-law viscous fluids with $n > 1$ the stress also continuously

increases with increasing strain rate but the effective viscosity continuously decreases with increasing strain rate

(Eq. (21), Figure 15). In the geological literature, such power-law viscous fluids have been termed both strain-rate

softening (due to the viscosity decrease; Smith, 1977) and strain-rate hardening (due to the stress increase;

Schmalholz and Maeder, 2012). We use here the terminology strain-rate hardening, which follows the nomenclature

in the material science literature (e.g. Ghosh, 1977). The term softening is usually applied to flow laws for which the

stress decreases with increasing strain (i.e. strain softening) or strain rate (i.e. strain-rate softening). Strain-rate

softening in power-law viscous materials is then expressed by a negative power law exponent ($n < 0$; e.g. Montesi

and Zuber, 2002).

   Stability analysis is of great practical importance in nearly all branches of mechanics because it shows

whether mathematically and mechanically correct solutions are possible (or stable) in nature. For example, pure

shear shortening and thickening of a perfectly straight (rectangular) and competent viscous layer embedded in a less

viscous medium, which takes place without folding, is a mathematically and mechanically correct solution.

However, this solution is not stable because any small geometrical perturbations, which are always present in natural

materials, amplify with faster velocities than the corresponding pure shear velocities and cause folding.

Homogeneous pure shear deformation of a competent viscous layer therefore does not occur in nature, although the

mathematical solution is correct. Stability analysis is thus essential to determine whether correct mechanical

solutions are applicable to natural processes. For pure shear shortening of a layer of thickness $H$ with geometrical

perturbations of amplitude $A$, the kinematic (passive) velocity due to shortening/thickening is $V_k = (H / 2 + A)\bar{D}_{xx}$

(assuming the vertical coordinate is zero in the middle of the layer) and the dynamic velocity due to active folding is

$V_a = \alpha \bar{D}_{xx} A$. Both velocities are equal when $A / H = 1 / \left[ 2(\alpha - 1) \right]$ (Schmalholz and Podladchikov, 2001a), which

corresponds to the transition between passive kinematic shortening/thickening and active ("explosive") folding (cf.

Ramsay, 1967, Fig. 7-37). For active folding, $V_a > V_k$ and hence $A / H > 1 / \left[ 2(\alpha - 1) \right]$. For example, for $\alpha = 20$,





active folding occurs for $A/H > \sim 0.026$, that is, when the amplitude is larger than ~2.6% of the layer thickness. If, for $\alpha = 20$, the initial amplitude $A_0$ is $< 0.026H$ then an initial deformation phase dominated by layer thickening occurs, but if $A_0$ is $> 0.026H$ then the deformation is immediately dominated by "explosive" folding.

The derivation of the theoretical dominant wavelength and corresponding amplification rates for linear and power-law viscous materials are only strictly valid for infinitesimal amplitudes but have been verified by both laboratory deformation experiments and numerical simulations at small fold amplitudes (e.g. Biot et al., 1961; Mancktelow and Abbassi, 1992; Schmalholz, 2006). The further development of fold geometries and the "selection" and "locking" of a fold wavelength have been discussed and analysed in a number of studies (Sherwin and Chapple,
1968; Fletcher, 1974; Fletcher and Sherwin, 1978; Hudleston and Treagus, 2010). However, a detailed discussion of their results is beyond the scope of this review.

        The theories outlined above are valid for single layers with the imposed layer-parallel displacement, velocity or load directly applied at the ends of the layer. However, folded veins or dikes are not infinitely long and
therefore a layer-parallel load is usually not directly applied at their ends. Schmid et al. (2004) considered folding of power-law viscous layers with a finite length, essentially corresponding to isolated inclusions removed from the lateral boundaries and embedded in a more extensive linear viscous medium. They considered the layers as ellipses with large aspect ratios (i.e. length to thickness ratio). If the viscous medium is shortened in a direction parallel to the long axis of the ellipse, the stresses in the surrounding viscous medium cause deformation and folding of the
isolated, elongate ellipse. Finite-length single-layer folding is controlled by the dimensionless ratio $D_a = \eta_e / (\eta_M a)$, where $a$ is the aspect ratio, $\eta_e = \eta \left( 1 + [2\eta/\eta_M a]^{n-1} \right)$ for $n \geq 1$ (with $\eta_e = \eta$ for $n = 1$), and $\eta$ is the effective viscosity of the layer calculated with the bulk shortening rate of the medium (i.e. $D_{II} = |\bar{D}_{xx}|$ in Eq. (21)). $D_a$ controls whether the theories assuming an infinite layer (for $D_a \ll 1$) can be applied or whether a modified theory has to be used. The modified theory is based on Muskhelishvili's complex potential method and shows that the
dominant wavelength for the general case of a finite length layer is

$$L_d = 2\pi H \left( \frac{1}{6n} \frac{\eta_e}{\eta_M} \right)^{\frac{1}{3}} \tag{24}$$



and the corresponding maximum amplification rate is

$$\alpha_d = \frac{1}{1+2D_a}\left(\frac{4n}{3}\frac{\eta_e}{\eta_M}\right)^{2/3} \tag{25}$$

For a linear viscous layer, $L_d$ is identical to the one for an infinite layer (Eq. (13)). The analysis further shows that

for finite-length layers ($D_a \gg 1$), the amplification rates are substantially reduced relative to the bulk shortening

rate and that the growth of large wavelength to thickness ratio folds is suppressed.

Both the standard thin-plate approach and the stability analysis assume that the fold amplitude grows

exponentially with time, which can be seen in Eq. (9) or results from time integration of Eq. (16). The solution in

Eq. (12) provides the amplification rate for all possible wavelengths (or Fourier components) and with this solution

the evolution of fold shapes can be calculated analytically for certain initial layer geometries, such as an initial

isolated bell-shaped geometry (Biot et al., 1961). The analytical treatment is possible because the bell-shaped

geometry can be represented as an infinite cosine series by a known Fourier integral expression

$$y = \frac{b}{1+\left(\frac{x}{a}\right)^2} = ab\int_0^\infty \exp(-ak)\cos(kx)\,dk \tag{26}$$

where $x$ and $y$ are the horizontal and vertical coordinates (with the origin on the central symmetry axis, see Fig. 14),

$k = 2\pi / L$ is the wavenumber, $b$ is here the amplitude of the bell-shape on its central symmetry axis (i.e. $y = b$ for $x$

= 0), and $a$ controls here the width of the bell-shape. If it is assumed that for very small initial amplitudes the

wavelength components are amplified independently, the amplified shape of the perturbation can be calculated at

any time by simple linear superposition using (Biot et al., 1961)

$$y = ab\int_0^\infty \exp(\alpha t - ak)\cos(kx)\,dk \tag{27}$$

where $\alpha$ is the amplification rate given by Eq. (12) and $t$ is time. The integral in Eq. (27) can be calculated

numerically and Eq. (27) allows the fold shape to be calculated for any time $t$ (or strain; Figure 16). In Fig. 16, the

analytically predicted fold evolution is compared with the evolution calculated by a numerical simulation based on

the finite-element method. The comparison shows that the analytical prediction is valid up to moderate fold

amplitudes (or fold limb dips), but also that the analytically predicted amplification becomes too large once the

amplitudes exceeded a certain value. In the case of the bell-shaped perturbation of Fig. 16, the infinitesimal





amplitude, linear superposition approach is a good approximation up to ca. 20% bulk shortening, with a maximum limb dip of ca. 25° (Fig. 16C).

As noted above, there is always a passive or kinematic component of layer parallel shortening and thickening that dominates at very small amplitudes before "explosive" folding manifests itself due to the exponential amplification rate. Indeed, as also presented above, in the limit of a perfectly planar layer, folds never develop regardless of the viscosity ratio and the layer will simply shorten and thicken. The result of this effect, which is not specifically considered in the thin-plate or stability analysis solutions above, is that the non-dimensional wavenumber, $s$, of a sinusoidal fold will increase during bulk shortening (Sherwin and Chapple, 1968) at a rate given, to first order, by $ds/dt = -2\bar{D}_{xx}s_0$, with $s_0$ being the initial non-dimensional wavenumber (e.g. Fletcher, 1974). Combining this with differential Eq. (16) for the instantaneous growth in fold amplitude and numerically integrating through time allows the calculation of a "preferred wavelength" (Sherwin and Chapple, 1968; Fletcher and Sherwin, 1978) of maximum amplification for a particular imposed bulk shortening (Sherwin and Chapple, 1968; Fletcher 1974; Fletcher and Sherwin, 1978; Johnson and Fletcher, 1994; Adamuszek et al, 2013).

It is of course expected that the initial exponential amplification of Eq. (16) must eventually break down, because fold amplitudes cannot grow exponentially at the same rate forever during shortening, the dynamic rate ($\alpha$) must decrease, but the passive component (the "1" in Eq. (16)) will remain. Indeed, both numerical calculations (Chapple, 1968; Zuber & Parmentier, 1996) and analogue experiments (Hudleston, 1973; Mancktelow and Abbassi, 1992) showed that the exponential amplification only occurs when the fold amplitudes are small and that fold amplification slows down with increasing fold amplitude or fold limb dip. Analytical solutions for the finite amplitude evolution of single layer folds have been derived by, for example, Mühlhaus et al. (1994) and Schmalholz and Podladchikov (2000). The analytical solutions are derived by considering geometrical nonlinearities due to finite amplitudes in the thin-plate Eq. (8). For example, Schmalholz and Podladchikov (2000) argue that the bulk shortening rate is no longer a good measure for the shortening rate of the layer (and hence the load $F$) when the amplitude is finite because, due to layer deflection, the layer is shortening less than a horizontal line parallel to the direction of bulk shortening. Shortening of the layer at finite amplitudes is more accurately described by the shortening of the fold arc length (or span), $S$. Therefore, the value of $\bar{D}_{xx}$ which controls the layer-parallel load is




not given by the change of the horizontal wavelength, that is $\bar{D}_{xx} = \dfrac{1}{L}\dfrac{dL}{dt}$, but by the change of the arc length, that is

$\bar{D}_{xx} = \dfrac{1}{S}\dfrac{dS}{dt}$ . The fold arc length can be related to the fold amplitude by


$$S \approx L + \pi^2 \frac{A^2}{L} \qquad (28)$$

The above relation can be derived with a Taylor expansion for the geometrical formula for the arc length. Due to the

resulting nonlinearity (i.e. the $A^2$ term) in the thin-plate equation, an explicit solution for $A$ as function of the

shortening $S$ (or time $t$) cannot be derived but an implicit solution is possible (Mühlhaus et al., 1994; Schmalholz

and Podladchikov, 2000). The finite amplitude solution for amplification of the dominant wavelength component, as

derived by Schmalholz and Podladchikov (2000), is then

$$\frac{L_0}{L} = \left(\frac{A\,L_0}{L\,A_0}\right)^{\frac{1}{2+\alpha_d}}\left(\frac{S\,L_0}{L\,S_0}\right)^{\frac{\alpha_d}{2+\alpha_d}} \qquad (29)$$

The subscripted index 0 indicates the initial value of the corresponding quantity. The solution (29) provides the

evolution of $A/L$ with progressive shortening quantified by $L_0/L$ (Figure 17). Values of $L_0/L$ can be calculated

for the corresponding value of $A/L$ (from which values of $S/L$ can be calculated from Eq. (28)) and hence a

curve $A/L$ versus $L_0/L$ can be plotted. A more accurate finite amplitude solution can be constructed if in Eq. (28)

$\pi^2$ is replaced by $\pi^2 / \left[1 + 3\left(A/L\right)^2\right]$, which is a term calibrated with numerical simulations. The breakdown of the

exponential amplification and the deviation from exponential amplification occurs approximately at a ratio of

amplitude to wavelength of (Schmalholz and Podladchikov, 2000; Schmalholz, 2006)

$$\frac{A}{L} = \frac{1}{\pi\sqrt{2\alpha_d}} \qquad (30)$$

Assuming a sinusoidal fold shape the ratio $A/L$ corresponds to a maximal limb dip of $\mathrm{atan}\left(2\pi A/L\right)180/\pi$ .

Hence, using Eq. (14) for linear viscous folding the breakdown of the exponential solution occurs at ~24 degrees

limb dip for a viscosity ratio of 25 and at ~15 degrees limb dip for a viscosity ratio of 100.

For the non-sinusoidal, bell-shaped initial perturbation of Figure 16, with a viscosity ratio of 75,

comparison with the numerical model indicates that the approximation is very good up to a maximum limb dip of

ca. 25 degrees (Figure 16c). The arc length in this example is determined by the finite shape and therefore by a non-



linear function of the amplitude and wavelength of the full infinite cosine Fourier series. It is clear that a simple

linear superposition approach, as presented by the theoretical curve in Figure 16, must therefore break down at finite

amplitude. A solution for multiple waveforms represented by a Fourier series, such as the case for the bell-shaped

perturbation, was derived by Adamuszek et al. (2013, their Appendix B). However, it should be noted that the plots

of their improved (and complex) solution relative to numerical models, presented in their Fig. 12, are not such a

dramatic improvement when compared to the simple linear superposition employed in Figure 16.

Adamuszek et al. (2013) combined the two previously proposed corrections to the original linearized

theories outlined above, namely the effect of homogeneous shortening and thickening on the non-dimensional

wavenumber and the need to consider the shortening rate along the layer at finite amplitude, rather than the

shortening rate imposed at the boundaries, into a combined and more accurate model for large amplitude folding

(LAF). Their solution allows the quite exact prediction of fold geometries up to moderate limb dips of ~20 degrees.

However, they also show that numerical solutions are required to accurately predict the shape of folds with larger

limb dips or finite amplitudes, or more complicated geometries due to initial irregularities.

The finite amplitude solutions outlined above are still fundamentally based on an elaboration of the linear

thin-plate approach. It is also possible to further elaborate the linear first-order stability analysis in order to obtain

results that are more accurate for finite, but still small amplitudes, by considering sinusoidal terms up to third-order

(Fletcher, 1979; Johnson & Fletcher, 1994). This higher-order stability analysis provides results which are valid up

to moderate limb dips of ~30 degrees and in particular provides more accurate fold geometries than the first-order

analysis and the thin-plate approach, especially for small to moderate viscosity ratios (< ~50).

One particular result of the finite amplitude solution is that the layer-parallel load, $F$, required to drive

folding decreases with increasing amplitude and hence shortening. This decrease in $F$ with increasing shortening is

only due to geometrical effects and is usually termed structural softening (or geometric softening/weakening; e.g.

Schmalholz et al., 2005; Schmalholz and Schmid, 2012). Structural softening can be quantified by determining the

evolution of the effective viscosity of the entire layer-medium system. The effective viscosity of a rock unit

consisting of competent layers embedded in a weaker medium during shortening with a constant bulk rate of

deformation can be calculated by the ratio of the area-averaged stress (e.g. the second invariant of the deviatoric

stress tensor) to the bulk rate of deformation. If shortening was accommodated by homogeneous pure shear at a

constant rate, the layer would deform by homogeneous shortening and thickening, and the effective viscosity of the



layered rock unit would remain constant. However, if folding takes place, the effective viscosity decreases during

bulk shortening, because the area-averaged stress is smaller during folding than during pure shear thickening (Figure

18).

Implicit in developing the analytical solutions above and in many analogue and numerical models is that

there are basically three stages of fold development. As is obvious from Eq. (16), the amplification velocity $dA/dt$

is directly proportional to both $A$ and $1+\alpha$. If the initial amplitude of irregularities in the layer is small and/or the

dynamic amplification rate is low (low viscosity ratio $\eta/\eta_M$), there will be an initial stage of dominantly layer-

parallel shortening and thickening without obvious fold development. Eventually, as the amplitude increases, this

will be followed by an "explosive" period of growth to finite amplitude (Ramsay, 1967), during which the bulk

shortening is predominantly accommodated by the folding. This cannot go on indefinitely, and the dynamic

amplification rate will decay toward zero, especially at limb dips > 45 degrees, in which case the limbs are now

infinitesimally stretched rather than shortened. Based on the finite amplitude solution of Eq. (29), Schmalholz

(2006) therefore separated the amplification of single layer folds into three stages: (1) a nucleation stage, during

which the amplification velocity is continuously increasing (acceleration); (2) an amplification stage, in which the

amplification velocity is decreasing but the dynamic amplification rate is still greater than the kinematic

amplification rate (i.e. $\alpha > 1$); and (3) a third, kinematic stage, where the dynamic amplification rate is less than the

kinematic one (i.e. $\alpha < 1$) and fold amplification is predominantly passive, that is, controlled by the basic state of

pure shear deformation.

An important, and still controversial question, is the magnitude of the effective viscosity ratio implicit in

observed natural single-layer folds (e.g. Hobbs et al., 2008; Schmid et al. 2010). Using Eq. (16) for the amplification

of the dominant wavelength with the maximal amplification rate, integrating it in time with $A_0$ being the initial

amplitude (at $t = 0$) and rearranging the terms yields

$$\alpha_d = \frac{\ln(A/A_0)}{-\bar{D}_{xx}t} - 1 \qquad (31)$$

The product $\bar{D}_{xx}t$ quantifies the bulk shortening. Folding could be considered significant if amplification

$A/A_0 > 10$ can be achieved within 20% shortening ($\bar{D}_{xx}t = 0.2$), which requires that $\alpha_d > 10$. Although the

development of distinct finite-amplitude folds still depends on the initial magnitude of irregularities in the layer ($A_0$),



it is a reasonable conclusion that the dominant amplification rates, which can be calculated for various scenarios

described above, should be at least larger than 10 for significant folding to take place. If the material properties are

known, one can then calculate and predict whether significant folding would take place or not. On the other hand, if

one can observe folds, it can be deduced that $\alpha_d$ was at least larger than 10 and make estimates for the material

parameters, such as estimating the minimum viscosity ratio for linear viscous folding. For example, folding of a

viscous layer in a viscous medium requires a viscosity ratio larger than ~25 to achieve $\alpha_d > 10$ (see Eq. (20)).

However, if both layer and embedding medium are power-law viscous, then for $n$, $n_m = 3$ an effective viscosity ratio

of $> ~8$ is sufficient (e.g. Schmid et al., 2010).

    As a concise summary, the simpler solutions for the dominant wavelength and corresponding maximal

amplification rate derived above are listed in Table 2.

    3D viscous folding has also been studied analytically with both the thin-plate (Ghosh, 1970) and the

stability analysis for linear viscous (Fletcher, 1991) and power-law viscous fluids (Fletcher, 1995; Mühlhaus et al.,

1998). The finite amplitude solution in Eq. (29) has been elaborated for 3D folding to take into account the ratio of

the two orthogonal, layer-parallel shortening (or extension) rates and was verified with 3D numerical simulations

(Kaus and Schmalholz, 2006).

### 2.1.2    Multilayer folding

    Multilayer folds (Figure 5, Figure 10) are more frequent in nature than single-layer folds. The mechanics of

elastic multilayer buckling was already discussed in Smoluchowski (1909) and we provide here only a short review.

Viscous multilayer folding is also discussed in detail in Johnson and Fletcher (1994) and a review of multilayer

folding can be found in Hudleston and Treagus (2010).

    The thin-plate approach shows that if a multilayer is composed of $m$ layers with identical thickness then the

dominant wavelength increases by the factor of $\sqrt[3]{m}$ (Biot, 1961). Ramberg (1962) applied the 2D stream function

approach to viscous multilayer folding and introduced the concept of contact strain. This concept states that strain

due to folding is negligible outside a zone about one initial wavelength wide on either side of the folded sheet. This

means that layers spaced at a distance greater than the wavelength of embryonic folds do not interact and can hence

fold independently (so-called "disharmonic" folds, e.g. Ramsay and Huber, 1987). An important first result of these

multilayer studies is that a multilayer with many stiff layers folds with a faster amplification rate than a single stiff





layer in the same medium, that is, folding instability increases with the number of competent layers (Biot, 1961;

Ramberg, 1963). Another important result is that the dominant or preferred (Ramberg, 1962) wavelength increases

but the selectivity decreases (i.e. increasing bandwidth) as the relative thickness ratio of the lower viscosity to the

higher viscosity layers decreases away from infinity (the single-layer case considered previously above) and that

below a critical value there is no real wavelength selection and the preferred wavelength is as long as the body

happens to be (e.g. see Fig. 5 of Ramberg, 1962; Fig. 4 of Ramberg, 1964; Fig. 7-40 of Ramsay, 1967).

In a series of articles, Biot developed a theory of internal buckling and multilayer folding (Biot, 1964a;

Biot, 1964b; Biot, 1965a; Biot, 1965b). In his theory of internal buckling, he considered multilayers as a confined

anisotropic or stratified medium under compressive stress (Biot, 1964a; Biot, 1965a). This simplifies the

mathematical analysis, because the individual deformation of each competent and incompetent layer is neglected.

Application of this internal buckling theory to viscous multilayers with competent and incompetent layers of equal

thickness, $H$, and a viscosity ratio larger than 50, yields an approximate formula for the dominant wavelength of

folds that develop in the confined multilayers of total thickness, $H_{tot}$ (Biot, 1964):

$$L_d = 1.9\sqrt{H\,H_{tot}} \tag{32}$$

The striking result is that the dominant wavelength is independent (or extremely insensitive in the non-approximated

formula) of the viscosity ratio between competent and weak layers. Biot presented several modifications of the

dominant wavelength in confined multilayers, depending on various underlying assumptions, for example:

$$L_d = 1.9\left(1 + 3.63\frac{\eta}{\eta_M}\frac{a^2}{m}\right)^{1/6}\sqrt{H\,H_{tot}} \tag{33}$$

where $m$ in this case is half the number of layers and $a = H_M / (H_M + H)$ with $H_M$ being the thickness of the

incompetent layer. If $a = 0$, Eq. (33) reduces to Eq. (32), which means that the dominant wavelength of Eq. (32)

represents multilayers where the incompetent layers have essentially zero thickness but are incompetent enough that

there is negligible shear stress between the competent layers (i.e., the contact between the competent layers is

perfectly lubricated). The results for the multilayer dominant wavelength presented above all show an extremely

weak sensitivity on the viscosity ratio, with the immediate implication that multilayer fold geometries are not

suitable for estimating viscosity ratios.



Biot further showed that in a homogeneous anisotropic (orthotropic) medium essentially two types of

internal instability can occur, essentially corresponding to either folding or localized kink-band formation (Biot,

1965b). Following Biot's theory of internal instability, Cobbold et al. (1971) performed analytical analyses and

laboratory experiments for compression of multilayers representing a homogeneous, anisotropic material and could

validate the application of the analytical results for an anisotropic medium to true multilayers consisting of

competent and incompetent layers. They further concluded that (i) the instability due to multilayer compression is

mainly governed by the degree of anisotropy rather than rheological properties such as elasticity or viscosity, (ii) the

form of the internal structure depends on the angle between layer orientation (or anisotropy orientation) and

compression direction, (iii) sinusoidal folds (developing for low degrees of anisotropy) and conjugate kink-bands

(developing for high degree of anisotropy) are end-members of a range of fold structures which can develop in

anisotropic material, (iv) chevron folds (i.e. folds with long straight limbs and short angular hinges) can form by

either convergence of conjugate kink bands or by progressive straightening of limbs during amplification of initially

sinusoidal folds, and (v) the scale of an internal structure in an anisotropic (but statistically homogeneous) rock

depends on the scale of the elements which cause the anisotropy.

Johnson and Fletcher (1994) present solutions based on stability analysis for selective amplification during

low-amplitude folding for many examples of multilayers with different configurations of stiff and soft layers and

embedding medium. They show that the strength of the folding instability generally increases with (i) the number of

stiff layers involved, (ii) increasing viscosity ratios between the stiff and soft layers and (iii) larger thickness of the

soft medium embedding the multilayer (see also Ramberg, 1962, his Fig. 12). They further show that the

amplification of multilayer folds with all layers having free slip interfaces is significantly stronger than for

multilayers with all layers having no slip (bonded) interfaces (Johnson and Fletcher, 1994). The difference in

interface condition (free slip versus bonded) is much more significant for folding of a multilayer than for folding of

a single layer.

A third-order stability analysis of viscous multilayer folding for small but finite amplitudes was performed

by Johnson and Pfaff (1989) to study fold shapes in multilayers. They distinguished three end-member forms in

multilayers: parallel, constrained and similar folds. Parallel (concentric) folds develop in multilayers confined by

soft medium whereas constrained (internal) forms develop in multilayers confined by very competent medium.

Similar (chevron) forms develop if wavelengths are short relative to the thickness of the multilayer.



Schmid and Podladchikov (2006) show that multilayer folding can occur in essentially three ways: as an effective single layer, as a true multilayer or as real single-layer folding (Schmid and Podlachikov, 2006). They consider a multilayer embedded in a weak medium. The multilayer is made of strong layers with equal thickness, $H$, and weak layers of equal thickness, $b$ (generally with $b \neq H$), alternating with the strong layers. Effective single layer folding occurs for $H/b < 1/L_d$, true multilayer folding for $1/L_d < H/b < L_d$ and real single-layer folding for $L_d < H/b$ where $L_d$ is the dominant wavelength given in Eq. (13).

A particular problem of multilayer folding is the mechanism of formation of asymmetric parasitic folds (S and Z folds) or "polyharmonic" folds (e.g. Ramberg, 1963, 1964; Ramsay and Huber, 1987). As outlined above, studies have consistently shown that multilayers fold more strongly than individual single layers. However, in the field, individual small folds are often observed on the limbs of larger folds, indicating that the smaller, shorter wavelength folds developed in the thin layers must grow faster than the larger folds that involve more layers, so that the small folds can be amplified sufficiently to be observable on the limbs of larger folds (Treagus and Fletcher, 2009). Frehner and Schmalholz (2006) performed 2D numerical simulations and showed that thin layers develop first symmetric folds during multilayer folding and that these folds are later transformed into an asymmetric shape due to relative shearing between fold limbs (Ramberg 1963; Frehner and Schmalholz, 2006). They argued that thinner layers have larger initial ratios of amplitude to thickness than thicker layers and that hence the "explosive" folding of thinner layers starts before "explosive" folding of the thicker layers. An alternative explanation was proposed by Treagus and Fletcher (2009) who studied analytically the amplification rates of individual layers within a multilayer for various configurations. They found that in viscous multilayers with a central thinner layer, the folds in the thin layer will initiate with greater amplification rates than larger folds of the whole multilayer if the multilayer is narrowly confined, and/or if the thin layer is the stiffest layer (Treagus and Fletcher, 2009).

Some solutions for the dominant wavelength and maximal amplification rates for multilayers are summarized in Table 2.

### 2.1.3 Lithospheric folding

Lithospheric folding is of general geodynamic importance because it demonstrates that large regions of the interior of tectonic plates are deformable. Internal deformation of continental plates contradicts a basic principle of plate tectonics *sensu strictu*, which states that tectonic plates are essentially rigid and deformation should only occur





at plate boundaries. Continental lithospheric folding and necking can therefore be considered as a specific and

important component of continental tectonics.

Probably the first large-scale folding analysis was performed by Smoluchowski (1909), who considered an

elastic beam and applied Eq. (5). Smoluchowski equated the load term $q$ to $\rho g A$, where $\rho$ is the crustal density. For

elastic beams under gravity, a critical load, similar to the Euler load, is required to cause buckling (Smoluchowski,

1909). However, if the beam is viscous it will fold no matter how small the applied compressive force (Biot, 1961).

For a small compressive load the fold amplification rate will be negligibly small and Biot suggested that a horizontal

stress $> 9\Delta\rho g H$ is required for folding to take place (Biot, 1961), where $\Delta\rho$ is the density difference between the

material below and above the layer and $H$ is layer thickness. Applying this formula to folding of the oceanic

lithosphere using $\Delta\rho = 3300 - 1000 \mathrm{kgm^{-3}}$ and $H = 20\mathrm{km}$ gives an unrealistically high stress $>\sim 4$ GPa. This stress,

if vertically integrated over the thickness of 20 km, corresponds to an unrealistically high force per unit length of

$\sim 8 \times 10^{13}$ N/m. All studies on elastic lithospheric folding have shown that the stress required for folding is

unrealistically high and hence folding of the lithosphere was considered impossible. However, McAdoo and

Sandwell (1985) showed that for an elastoplastic layer, with lithospheric strength based on experimental rock

mechanics, the average stress required to fold the oceanic lithosphere is reduced to 600 MPa, which could be a

naturally realistic stress magnitude. They therefore concluded that the observed basement topography and geoid

height in the northern Indian Ocean results from folding of the oceanic lithosphere, caused by the India-Asia

collision.

Biot (1961) also derived the dominant wavelength for a viscous layer floating on an inviscid medium in the

field of gravity, which can be written as

$$L_d = 2\pi H \sqrt{\frac{2\eta |\bar{D}_{xx}|}{\Delta\rho g H}} = 2\pi H \sqrt{\frac{1}{Ar_F}}; \quad Ar_F = \frac{\Delta\rho g H}{2\eta |\bar{D}_{xx}|}. \tag{34}$$


This dominant wavelength solution was rederived by Ramberg and Stephansson (1964) with a slightly different

approach, and in addition was verified by them with laboratory experiments. In Eq. (34), $Ar_F$ is the Argand number

for folding and represents the ratio of gravitational stress acting against folding to compressive stress driving folding

(Schmalholz et al., 2002). The Argand number was originally introduced in a slightly different form for thin viscous

sheet models (England and McKenzie, 1982). As mentioned above, the main difference between folding of an



elastic layer and a viscous layer is that for an elastic layer a certain load must be exceeded to initiate folding, that is, there is a critical load (e.g. Euler load) below which no instability appears, whereas for a viscous layer folding takes place for any compressive load. The formulation for the dominant wavelength for viscous and elastic folding under gravity is identical if the term $2\eta\left|\bar{D}_{xx}\right|$ in the viscous case is replaced by half the compressive load applied to the

elastic layer (Biot, 1961). However, if the load is small then the maximal amplification rate $\alpha_d$ is less than one order of magnitude larger than the bulk shortening rate $\left|\bar{D}_{xx}\right|$ (i.e. $\alpha_d < 10$) and hence folding is insignificant. Values of $\alpha_d$ should be at least $> 10$ for folding to be significant and observable for typical tectonic shortening of several tens of percent (see Sect. 2.1.1 above). Biot (1959, 1961) argued that $4\eta\left|\bar{D}_{xx}\right|/\Delta\rho gH >\sim 9$ is required for viscous folding under gravity to be significant. The maximum value of the amplification rate for folding of a power-

law viscous layer under gravity is (Schmalholz et al., 2002)

$$\alpha_d = \frac{6n}{Ar_F} = 6n\frac{2\eta\left|\bar{D}_{xx}\right|}{\Delta\rho gH} \tag{35}$$

The dominant wavelength for viscous and power-law viscous layers resting on an inviscid medium under gravity is the same (Schmalholz et al., 2002). Values of $n$ for dislocation creep are usually in the range 3 - 5. The result for power-law viscous folding can also be applied to the oceanic lithosphere. Assuming $n = 5$, a total

horizontal stress $4\eta\left|\bar{D}_{xx}\right| = 350$ MPa, a density difference between mantle lithosphere and water $\Delta\rho = 2300$ kgm$^{-3}$ and $H = 25$ km yields $\alpha_d \approx 9$, which is not sufficient for significant folding. However, if we assume that the mechanically strong (upper) region of the mantle lithosphere is deforming by low-temperature plasticity (Peierls creep), then the effective (or apparent) value of $n$ is in the range 10 – 25 (Dayem et al., 2009; Schmalholz and Fletcher, 2011) and we get $\alpha_d \approx 19-47$, a value sufficient for significant folding. Furthermore, if we assume that

the competent level of the oceanic lithosphere is not overlain by water, but by unconsolidated sediments, then the density difference reduces to $\Delta\rho = \sim 1000$ kgm$^{-3}$ (Martinod and Molnar, 1995) and $\alpha_d \approx 43-107$. These amplification rates are sufficient for significant folding of the oceanic lithosphere. The assumed total stress of 350 MPa integrated over an assumed 25 km thickness yields a force per unit length of $8.75 \times 10^{12}$ Nm$^{-1}$, which is a reasonable value (Martinod and Molnar, 1995). Also, the dominant wavelength for the applied values with $\Delta\rho$

$= 1000$ kgm$^{-3}$ is $\sim 137$ km, which is within the range of wavelengths observed in the northern Indian ocean of 130 –





250 km (McAdoo and Sandwell, 1985). It follows that the simple thin-plate based solution for folding of a power-law viscous layer under gravity supports the proposal that folding of the oceanic lithosphere is feasible for reasonable compressive stresses. The observed fold wavelength can also be used to estimate the Argand number, $Ar_F$ (Eq. (34)), which requires an assumed value for the effective thickness of the lithospheric level that is actively

folding. Assuming 25 km for this thickness, wavelengths between 130 and 250 km in the Indian ocean correspond to $Ar_F$ between 1.5 and 0.4, respectively. Assuming an effective thickness of 15 km gives $Ar_F = 0.14 - 0.53$. For central Asia, Burov et al. (1993) estimated the wavelength due to folding of the mantle lithosphere to be between 300 and 360 km (see also Figure 7). Note that for folding of the continental mantle lithosphere, which is mechanically decoupled from folding of the crust (Burov et al., 1993), the density difference relevant to folding is

the difference between mantle and crustal density. Assuming effective thickness between 20 and 50 km for the folding mantle lithosphere with wavelength between 300 and 360 km gives $Ar_F = 0.12 - 1.1$. These simple estimates suggest that values of $Ar_F$ are in the order of 0.1 to 1 for folding of oceanic and continental lithosphere. If, for both oceanic and continental lithosphere, it is the upper, cold level of the mantle lithosphere that controls folding and if this level deforms plastically or by low-temperature plasticity with effective values of $n = 10 - 25$, then values of

$Ar_F = 0.1 - 1$ correspond to significant folding based on Eq. (35). Cloetingh and Burov (2011) compiled wavelengths of folded lithosphere, including crustal, mantle and whole lithosphere folding, for 17 regions worldwide, including Central Asia (Burov et al., 1993), Central Australia (Lambeck, 1983), the NE European platform (Bourgeois et al., 2007), the Tibet/Himalayan syntaxes belt (Shin et al., 2015) and the Indian oceanic lithosphere (McAdoo and Sandwell, 1985; Krishna et al., 2001), and showed that all fold wavelengths are between

40 and 700 km. Not surprisingly, wavelengths for crustal folding are smallest and wavelengths for whole lithosphere folding are largest. They also showed that the wavelengths increase with the thermo-tectonic age of the lithosphere, because older lithosphere is mechanically stronger and exhibits larger effective thickness of the folded levels. Cloetingh and Burov (2011) also discussed lithospheric folding as mechanism of sedimentary basin formation.

Lithospheric folding has also been studied with the stability analysis (Zuber, 1987; Martinod and Davy,

1992; Burov et al., 1993; Martinod and Molnar, 1995), which yields more accurate (but also more complicated) solutions for the amplification rate without changing the first-order results and conclusions of studies based on the thin-plate approach. The solution in Eq. (35) is strictly valid only for infinitesimal amplitudes and $\alpha_d$ decreases





with increasing fold amplitude according to the finite amplitude solution (Schmalholz and Podladchikov, 2000).

Nevertheless, Eq. (35) indicates that plastic behaviour ($n > \sim 10$) can significantly increase the growth rate of a

folding instability. Indeed, numerical simulations of lithospheric shortening considering representative viscoplastic

yield strength profiles for the continental and oceanic lithosphere indicate that lithospheric folding most likely takes

place during lithospheric compression (Zuber and Parmentier, 1996; Burg and Podladchikov, 1999; Cloetingh et al.,

1999; Gerbault, 2000). The intensity of folding depends mainly on the applied bulk shortening rate and the

temperature at the Moho, which controls the integrated strength of the lithosphere (Schmalholz et al., 2009).

Martinod and Molnar (1995) performed a stability analysis in which they considered a power-law viscous

rheology and Mohr-Coulomb plastic yield strength of the Indian oceanic lithosphere. They argued that the oceanic

lithosphere is overlain by unconsolidated sediments with average density of 2300 $\text{kgm}^{-3}$, which gives

$\Delta\rho = 1000\text{kgm}^{-3}$. Their result indicates that a force per unit length of $4.8\times10^{12}$ ($\pm1.3\times10^{12}$) N/m is sufficient to fold

the oceanic lithosphere, which is a value significantly smaller than the ones obtained from simpler models based on

folding of beams. This value is important because the lateral variation of the gravitational potential energy, caused

by the lateral crustal thickness variation between the Indian foreland and the Tibetan plateau, generates a force per

unit length of $\sim7\times10^{12}$ N/m (Molnar and Lyon-Caen, 1988), which means that the growth of the Tibetan plateau

could in principle have caused the folding of the Indian oceanic lithosphere (Molnar et al., 1993). The analysis of

Molnar and co-workers caused some controversy because other authors using thin viscous sheet models argued that

the values for the force per unit length related to the Tibetan plateau from Molnar and co-workers were

overestimated by a factor of two and, hence, that the Tibetan plateau alone could not be responsible for folding of

the Indian ocean lithosphere (Ghosh et al., 2006; Ghosh et al., 2009). However, the analysis of Molnar and co-

workers is based on total stress and differential stress (difference between maximal und minimal principal stress),

whereas the results of thin viscous sheet models are based on deviatoric stress, which is half the differential stress in

the thin viscous sheet model (Schmalholz et al., 2014). Furthermore, the force (or force per unit length in 2D)

driving folding is controlled by the total stress ($4\eta\left|\overline{D}_{xx}\right|$) and not by the deviatoric stress ($2\eta\left|\overline{D}_{xx}\right|$; see Eq. (8)), so

that the application of Molnar and co-workers of the force per unit length (calculated from differential stress) due to

lateral variations in gravitational potential energy to folding of the oceanic lithosphere is correct.

A frequently applied model for viscous deformation of the continental lithosphere is the thin viscous sheet

model (England and McKenzie, 1982). This model assumes that lithospheric folding is negligible and that the



lithosphere deforms by homogeneous, kinematic thickening. Thin-sheet models consequently assume that vertical

velocities due to folding are less than vertical velocities due to kinematic thickening and hence they assume that

amplification rates for folding $\alpha_d < 1$. Thin-sheet models are useful to explain the first-order response of the

continental lithosphere due to shortening on the scale of 1000's of kilometres but they are not suitable for estimating

the deformation on the 100 km scale, because on this scale folding is likely to be important and may control the

lateral variation of vertical velocities (Lechmann et al., 2011). For example, thin sheet models are useful to predict

the average topography of Central Asia (Figure 7A) but not to predict the characteristic fold-like topography on the

100's of kilometre scale and the related vertical velocities.

**2.2    Experimental results**
        In most branches of science, proposed analytical solutions can be tested by direct observation or experiment.

However, in the case of folding and necking of rock layers as considered here, this is impossible due to the long

times and large forces, pressures, temperatures, and (often) length scales involved. By necessity, the progressive

development of such structures can only be studied by analogue and, more recently, numerical modelling. Initially

these analogue models were only qualitative but progressively became more quantitative with the application of

correct scaling laws (Hubbert, 1937) and the use of materials more rheologically similar to rocks but deformable at

lower stresses, temperatures and pressures. The pioneer in analogue modelling of folding was Hall (1815), who

"conceived that two opposite extremities of each bed being made to approach, the intervening substance, could only

dispose of itself in a succession of folds, which might assume considerable regularity, and would consist of a set of

parallel curves, alternatively convex and concave towards the centre of the earth". To test this premise he carried out

his now famous experiments using layers of cloth to demonstrate that the folds he observed in nature could develop

by shortening of horizontally layered rocks by application of a horizontal force (Figure 1). The experiments were

entirely qualitative but established the basic principle. Since then, a large number of analogue experimental studies

have investigated the influence of material contrast (e.g. viscosity ratios), rheology (elastic, linear and power-law

viscous, plastic and different combinations), material anisotropy, and initial perturbation geometry on the initiation

and development of single- and multi-layer folds and boudins. Only a limited selection are presented here as

examples.





In a companion paper to Biot (1961), Biot et al. (1961) presented a series of analogue models aiming to provide experimental verification of the analytical results for folding of stratified viscoelastic media (Biot, 1957;

Biot, 1961). These experiments considered layer parallel shortening of both elastic and viscous layers embedded in a viscous matrix. Biot's thin-plate theory is for a layer of infinite length and predicts an amplification rate as a function of normalized wavelength (or wavenumber) given by Eq. (12) with a dominant wavelength, corresponding to the maximum amplification rate, given by Eq. (13). In an analogue model, a layer of infinite length is unattainable and an initial infinitesimal amplitude perturbation spectrum of perfect random white noise (all wavelength components present and with equal amplitude) is also unrealistic. A novel alternative approach proposed by Biot et

al. (1961) was to consider the amplification of an initial isolated bell-shaped perturbation. This can be represented as an infinite cosine series by a known Fourier integral expression given in Eq. (26) and the evolving fold geometry with time (strain) can be calculated with Eq. (27). Biot et al. (1961) used this approach in a numerical evaluation of the time history of folding and sideways propagation away from an isolated perturbation but not in their analogue

models. In these models, they used thin plates of aluminium or cellulose acetate butyrate (elastic layers) or roofing tar (viscous layers) in a corn syrup viscous matrix, without any prescribed initial perturbation, to establish a good correspondence with theory – at least for the very high viscosity ratios ( $2.22 \times 10^3$ and $4.28 \times 10^4$ ) and corresponding long dominant wavelength to thickness ratios considered (45 and 121, respectively). As predicted by Biot's theory, for such large contrasts in properties, amplification rates were high and wavelength selection strong,

so that a relatively clear sinusoidal wave-train was rapidly developed. However, such high wavelengths are not typical of natural examples, where common wavelength to thickness ratios are between 2 and 16 with a mean value at ~6.5 (e.g. Hudleston and Treagus, 2010).

Ramberg and Stephansson (1964) performed laboratory experiments on folding of a viscous plate (made from molten mixtures of colophony and diethyl phthalate) floating on an aqueous solution of potassium-iodide to

verify the dominant wavelength for folding under gravity given in Eq. (34). They showed that the value of $L_d / H$ developed in the experiments is linearly proportional to the ratio of $\sqrt{\sigma / \Delta \rho g H}$ , where $\sigma$ is the compressive load applied in the experiments and corresponds to the value of $4\eta \left| \bar{D}_{xx} \right|$ in the theoretical analysis. The experiments hence verified the theoretical result for the dominant wavelength, which states that $L_d / H$ is directly proportional to

$\sqrt{4\eta \left| \bar{D}_{xx} \right| / \Delta \rho g H}$ .





Ghosh (1966) studied single-layer buckle folding under simple shear, using combinations of modelling

clay, putty and wax. He noted that the fold axis developed parallel to the major axis of the strain ellipse on the

surface of the layer (i.e. perpendicular to the principal component of shortening within the layer), which, for

generally oblique layering, is not necessarily parallel to a principal axis of the applied bulk strain. He also noted that

the single layer folds are, at least initially, generally symmetric despite the simple shear boundary conditions. This is

consistent with the later, general observation of Lister and Williams (1983) that single layer buckle folds are good

examples of coaxial spinning deformation (their Fig. 4) and agrees with results of numerical models (Viola and

Mancktelow, 2005; Llorens et al., 2013a). Ghosh (1968) also did analogue experiments on multilayer folding to

develop rough qualitative constraints on the transition from conjugate to chevron to concentric folds. Currie et al.

(1962) had previously also qualitatively investigated single- and multi-layer folding in elastic materials using

photoelastic gelatin. With this experimental technique they could not only investigate the influence of layer

thickness and ratios in elastic properties on fold wavelength, but also analyse the stress trajectories in the layer and

matrix during folding. Their experiments provided an excellent visual representation of the zone of contact strain

around a folding layer and the consequent development of disharmonic or harmonic folding depending on the

spacing between layers (their Plate 2).

Hudleston (1973) performed experiments to study the development of single-layer folds with shortening

parallel to the layer. The material used for both layer and matrix were mixtures of ethyl cellulose in benzyl alcohol,

which, at the low concentrations used in his experiments, is effectively linear viscous. The viscosity ratios

considered were between 10 and 100 and thus much lower than those used by Biot et al. (1961). One of the aims of

the experiments was to establish that folding to finite amplitude with such low ratios, and correspondingly short

wavelength to thickness ratios, was possible, in contrast to what was implied in the original papers of Biot (1961)

and Biot et al. (1961). In these experiments, Hudleston (1973) also specifically investigated layer-parallel shortening

and thickening and the transition to rapid ("explosive") fold amplification, as well as making harmonic analyses of

the experimental fold shapes.

Cobbold (1975a) carried out analogue experiments to study the sideways propagation of folds away from

an initial isolated perturbation in a single layer undergoing layer-parallel shortening, using a pure-shear deformation

rig (Cobbold, 1975b; Cobbold and Knowles, 1976). Materials used were well-calibrated paraffin waxes of different

melting points, with power-law stress exponents of ca. 2.6 and an effective viscosity ratio between layer and matrix



of ca. 10. Conceptually this was an experimental investigation of the process considered theoretically and
numerically by Biot et al. (1961) with an initial isolated bell-shaped perturbation, but for power-law viscous

materials and a much lower (and more realistic) viscosity ratio. However, Cobbold (1975a) used a cylindrical form
for the initial perturbation, rather than a bell-shape with the known Fourier integral representation of Eq. (26), and
did not consider the propagation in terms of amplification of Fourier spectral components. Instead, he introduced the
important concept of the perturbation flow lines (Passchier et al., 2005) to qualitatively investigate the sideways
spread of the folding instability.

Gairola (1978) made single-layer fold experiments with plasticene layers embedded in putty to investigate
the effects of progressive deformation on fold shape and particularly on the internal strain within the layer and on
the varying position of the neutral surface. He found that the appearance of the neutral surface depends on the
"ductility contrast" between the layer and matrix, and the amount of strain. A neutral surface may not appear at all if
the contrast between layer and matrix is very small, due to the strong component of layer-parallel shortening, which

agrees with recent results of numerical simulations (Frehner, 2011).

Neurath and Smith (1982) performed folding and necking experiments with wax models, measured the
effective viscosities and power-law exponents for the wax models, and compared the experimentally determined
amplifications rates for folding and necking with the corresponding theoretical rates. They showed that for folding,
theoretical and measured amplification rates more or less agreed with the theoretical rate as derived by Smith (1975,

1977, 1979) and the equivalent results of Fletcher (1974, 1977).

Abbassi and Mancktelow (1990) investigated the influence of initial perturbation shape on fold shape,
establishing that markedly asymmetric folds, even with overturned limbs, could develop by amplification of a small
initially asymmetric irregularity, despite the fact that the imposed boundary condition was layer parallel shortening
in a pure shear deformation rig (Mancktelow, 1988a). Abbassi and Mancktelow (1992) and Mancktelow and

Abbassi (1992) employed the isolated bell-shaped perturbation technique originally developed by Biot et al. (1961)
directly in analogue experiments, both to investigate the effects of perturbation geometry on fold shape and lateral
propagation (Cobbold, 1975a) and to experimentally determine fold amplification rates. Instead of calculating a
numerical forward model for a specific amplification rate curve as done by Biot et al (1961), they reversed the
approach and used the changing shape of an initial bell-shaped perturbation with known initial values of *a* and *b* (see

Eq. (26) for the meaning of *a* and *b*) to determine, via Fourier analysis, the amplification rates for folding in well-



calibrated power-law viscous materials (paraffin waxes of different melting temperatures; Cobbold, 1975a;

Mancktelow, 1988b). The amplification rate curves determined in this way were directly comparable to those

predicted theoretically (Fletcher, 1974; Smith, 1975; Fletcher, 1977; Smith, 1977) but for short wavelengths and

particularly for narrow initial perturbations, the observed amplification rates were generally higher than theoretical

values. This could reflect the strain softening behaviour of the layer, as also suggested by Neurath and Smith (1982)

for their boudinage experiments (but significantly not for their folding experiments). The experiments indicate that

bonding of the matrix-layer interface may have a much greater effect on the amplification rate curve than is

theoretically predicted, at least for the low to moderate effective viscosity ratios investigated (8 and 30). For better

bonding between layer and matrix, the amplification rate decreases and consequently the amount of initial layer

parallel shortening increases. Abbassi and Mancktelow (1992) observed that the influence of the initial perturbation

is greater for broader irregularities, when the average wavelength component is longer than the dominant

wavelength, than for narrow isolated irregularities.

Marques and Podladchikov (2009) placed a thin layer of either plasticine or polyethylene between viscous

polydimethylsiloxane (PDMS, Dow Corning SGM36) below and Fontainebleau quartz sand above. The PDMS

represents the ductile part of the lithosphere, the quartz sand the brittle parts and the thin layer of either plasticine or

polyethylene the thin elastic core, which is easily flexed but unstretchable/unbreakable. Their results show that a

very thin, elastic layer between an overlying brittle and underlying viscous medium produces folding as the

dominant deformation mechanism during shortening, and not brittle faulting or viscous homogeneous thickening.

Recently, Marques and Mandal (2016) have made experiments to investigate the buckling and post-

buckling behaviour of an elastic single layer (cellophane, plasticine, or polyethylene film) in a linear viscous

medium (PDMS silicone putty). The experiments were performed in two stages: a first stage of buckling by layer-

parallel shortening at different rates and a second stage of buckling relaxation with fixed lateral boundaries. They

found major contradictions between their experimental results and both the analytical results of Biot (1961) for the

buckling phase and with the analytical solutions and conclusions of Sridhar et al. (2002) for the buckling relaxation

stage. Their results have still to be explained by theoretical models.

Analogue experiments on single- and multi-layer folding have generally investigated a geometry where the

principal bulk shortening direction is within the layer and the principal extension axis is perpendicular to the layer.

Experiments with oblique layers are technically difficult because the layer ends tend to slide along the boundaries.



Oblique loading of the ends also introduces unavoidable additional perturbation components, because the planar

boundary is no longer parallel to the axial plane of the developing folds. Grujic and Mancktelow (1995) carried out

pure and simple shear analogue experiments, where the intermediate axis was perpendicular to the layer (i.e. both

the principal shortening and extension directions were within the layer). Models were generally constructed of

power-law ($n$ = 2-3) viscous paraffin waxes of different melting temperatures, but in some cases a matrix of linear

viscous PDMS silicone putty was used to allow observation. Folds developed parallel to the stretching direction but

significant amplification was only possible for a (very) high effective viscosity ratio (i.e. only for ca. 600 and not for

ca. 30). In rotational simple shear experiments, Grujic and Mancktelow (1995) observed that, in high viscosity ratio

experiments, the amplifying folds develop initially approximately parallel to the infinitesimal stretching direction.

With increasing shear and amplification, the fold axes remain fixed to the same material points and therefore rotate

as a passive line, rotating toward but not strictly tracking the finite extensional axis. As a result, there is a component

of antithetic shear along the axial plane of these folds. The observation that fold hinges remain fixed to material

points may reflect material damage and strain softening along the fold hinges, which correspond by definition to

lines of maximum layer curvature. The paraffin waxes employed are strain softening (Mancktelow, 1988b), so that

increased strain in the hinge will tend to subsequently localize further strain.

Davy and Cobbold (1991) modelled the lithosphere as 2, 3 or 4 layers: brittle crust (quartz sand), ductile

crust (silicone), brittle mantle (quartz sand) and ductile asthenosphere (sugar solution). Variation in the rheology

with depth (e.g. temperature dependence of viscosity) was not considered in the simplified model but the potential

effects of erosion were. They investigated the interplay between buckling and lithospheric thickening, showing that

thickening style is mainly dependent on mantle behaviour, as well as demonstrating the effect of low degrees of

coupling, when the brittle crust can detach and buckle independently of mantle layers.

Martinod and Davy (1994) modelled the development of periodic instabilities in continental and oceanic

lithosphere under compression. The lithosphere was modelled as a stack of alternating brittle (quartz sand) and

ductile (silicone putty) layers. As with Davy and Cobbold (1991), there was no vertical variation within the layers

themselves. For small strain, the deformation modes mainly depend on the spatial distribution of the brittle layers

and the amplitude of buckling is an exponential function of horizontal strain, as would be expected for folding (Eq.

(9)).



## 3    Short history of necking

The terms "boudin" and "boudinage" were first introduced by Lohest et al. (1908) and Lohest (1909) as a descriptive term for sausage-like structures (hence "boudin", which is a French word for blood sausage) that they

observed in the High-Ardenne Slate Belt, which were developed in psammitic layers embedded within a more pelitic matrix. However, recent studies now interpret these classic "boudins" of Lohest and co-authors to be in fact "mullions" (Urai et al., 2001; Kenis et al, 2002, 2004), developed due to layer shortening. "Pinch and swell" was already used by Matson (1905) as a purely descriptive term for the geometry of peridotite dykes from near Ithaca, New York, but without a sketch and with the implication that this was an original intrusive rather than tectonic

structure. A short but relatively comprehensive summary of early literature on boudinage is given by Cloos (1947). By this time, there were already descriptions in the literature of more ductile pinch-and-swell structures (e.g. Ramsay 1866; Harker 1889; Walls 1937), but Cloos (1947) concentrated more on examples involving fracture and interpreted the initial fractures as tension joints normal to the direction of extension. However, he notes that "the barrel shape of the classical boudins is somewhat puzzling but seems to be a function of incipient flowage in the

competent layer". Fracture development producing rectangular or barrel-shaped is promoted by the dynamic (or tectonic) underpressure inherently developed in an extending competent layer, in contrast to the overpressure developed if the layer is shortened (Mancktelow, 1993, 2008). This under- or overpressure is associated with corresponding refraction in the principal stress axes in the more competent layer (Mancktelow, 1993), so that extensional fractures are nearly perpendicular to layering, as typically reported for brittle boudins (e.g. Cloos, 1947).

As discussed by Rast (1956), the difference in behaviour between barrel-shaped and lozenge-shaped boudinage directly reflects the mechanical response of the layer: if the layer is effectively elasto-plastic (i.e. "brittle") it develops extensional fractures (joints) and rectangular or barrel-shaped boudins; if viscous flow dominates (at least initially), mechanical instability will lead to necking and the development of pinch-and-swell or lozenge shapes.

### 3.1    Theoretical results

We focus here on studies investigating ductile necking instability. Many studies on boudinage consider brittle boudinage or study deformation with an initial configuration where the competent layer is already broken or already includes weak layers separating the layer. Such studies are useful to investigate the kinematic evolution of boudins





but yield no insight into the necking instability. An extensive review of boudinage and necking is also provided in

Price and Cosgrove (1990).

### 3.1.1    Single layer necking

Galilei (1638) performed one of the first experiments to test the tensile strength of columns (Figure 8) and

the first mathematical study of necking was probably by Considère (1885) (see also Dieter, 1986). Assuming a

homogeneous layer with thickness $H$, the extensional load (here force per unit length) is $F = \sigma H$ with $\sigma$ being the

total stress. We assume in addition that the material is strain hardening, that is, the stress and hence the load-carrying

capacity increases with increasing strain. During extension at an imposed constant rate, the stress also increases due

to the decrease in layer thickness. Necking or localized deformation begins when the increase in stress due to

decrease in layer thickness becomes greater than the increase in load-carrying ability of the layer due to strain

hardening. At the onset of extension, the load required to extend the strain-hardening layer is increasing. The

maximum load is achieved when the rate of change of the load during extension is zero, that is,

$dF = d\sigma H + \sigma dH = 0$. The ratio $dH / H$ corresponds to the vertical shortening and is identical to the negative of

the horizontal (layer-parallel) extension, $-dL / L = -d\varepsilon$ with $L$ being the layer length, assuming mass conservation

and an incompressible material. The variation of the load can be reformulated to

$$dF = d\sigma + \sigma \frac{dH}{H} = d\sigma - \sigma d\varepsilon = 0 \quad \Rightarrow \quad \frac{d\sigma}{d\varepsilon} = \sigma \qquad (36)$$

The above equation is known as the Considère criterion and states that the load is at a maximum when the rate of

strain hardening, $d\sigma / d\varepsilon$, is equal to the stress, $\sigma$. When $d\sigma / d\varepsilon > \sigma$ then $dF > 0$ and the extension is stable,

whereas when $d\sigma / d\varepsilon < \sigma$ then $dF < 0$ and unstable necking takes place. Introducing the dimensionless strain-

hardening coefficient $\beta = d\sigma / d\varepsilon\sigma$, the Considère criterion predicts onset of necking instability for $\beta < 1$. The

Considère criterion can be used to predict the amount of extension at which necking takes place. For example, if a

material follows a strain hardening stress-strain relation of the form $\sigma = K\varepsilon^m$, with $K$ and $m$ being here material

parameters ($K > 0; 0 < m < 1$), then $d\sigma / d\varepsilon = Km\varepsilon^m / \varepsilon$. Substituting the expressions for $\sigma$ and $d\sigma / d\varepsilon$ in the

Considére criterion, $\sigma = d\sigma / d\varepsilon$, yields $\varepsilon = m$. This means that necking begins when the extensional strain $\varepsilon$ is

equal to the strain-hardening exponent $m$.

The analysis above, which is based on the early work of Considère, assumes that the flow stress is only

dependent on strain. A similar analysis can be done for a material that is both strain and strain-rate sensitive. The





strain-rate sensitivity is described by a standard, strain-rate hardening power-law viscous flow law, that is,

$\sigma = \eta_C \dot{\varepsilon}^{1/n}$ ($n > 1$) where here $\dot{\varepsilon}$ is the strain rate. The result of the stability analysis shows that the onset of necking

instability takes place when (Hart, 1967; see also Dieter, 1986)

$$\beta + \frac{1}{n} < 1 \qquad (37)$$

If only strain-rate sensitivity is considered ($\beta = 0$), then the stability criterion reduces to $1/n < 1$ and indicates that

necking in power-law viscous material only takes place if $n > 1$, which means that in a linear viscous material (

$n = 1$) a necking instability does not occur. This result for power-law viscous material has been confirmed by a

slightly different analysis of Emerman and Turcotte (1984) and by the analysis of Smith (1975, 1977) presented in

more detail below. The stability criterion of Hart (1967) in Eq. (37) is controversial and has been much discussed in

the engineering literature because it is not universally valid for any kind of initial perturbation or imperfection

(Ghosh, 1977; Hutchinson and Obrecht, 1978). Indeed, there is a very extensive engineering literature on necking in

strain and strain rate sensitive materials due to its importance, for example, for metal forming, but a review of the

non-geological literature is beyond the scope of this review. The interested reader is referred to (Hill, 1952; Ghosh,

1977; Hutchinson and Neale, 1977; Hutchinson and Obrecht, 1978; Tvergaard et al., 1981; Hutchinson and Neale,

1983; Dieter, 1986).

Smith (1975, 1977) applied the stability analysis to both folding and necking of linear and power-law

viscous layers embedded in a linear and power-law viscous medium. He showed that the dominant wavelength

solution for folding and necking is identical (for the same material parameters), but that the corresponding

amplification rates for folding and necking are different (Figure 14). The maximal amplification rate for necking,

that is the maximum from Eq. (18) for $\theta = 1$, which corresponds to the dominant wavelength, can be approximated

(Smith, 1977) by

$$\alpha_d \approx (n-1) \qquad (38)$$

The result shows that for linear (Newtonian) viscous fluids $\alpha_d = 0$ and there is no active component of necking and

therefore that necking does not occur in linear viscous fluids, in agreement with Eq. (37) for $\beta = 0$. Hence, pinch-

and-swell structure is an excellent paleo-rheology indicator, because rocks developing a pinch-and-swell instability

behaved as power-law viscous fluids (or more generally as nonlinear viscous fluids) during pinch-and-swell

formation.



Neurath and Smith (1982) performed folding and necking experiments with wax models, measured the effective viscosities and power-law exponents for the wax models and compared the measured amplifications rates for folding and necking with the corresponding theoretical amplification rates. For folding, theoretical and measured amplification rates more or less agreed but for necking the measured amplification rates where significantly higher (a factor 2 - 3) than the theoretical ones (Neurath and Smith, 1982). They suggested that the discrepancy could be due to some kind of strain-softening by which the power law exponent would increase with increasing strain. They show analytically that strain softening can be described with an effective power-law stress exponent

$$\frac{1}{n_{eff}} = \frac{1}{n} - \frac{2}{\sqrt{3}}\frac{1}{\alpha_d}\frac{1}{\varepsilon^*} \tag{39}$$

where $\varepsilon^*$ is a measure for the strain during softening. Note that in Neurath and Smith (1982) values of $n_{eff}$ remained positive during strain-softening so that the material remained strain-rate hardening.

A simple 1D analytical solution for the evolution of thinning during necking of a power-law layer can be found by assuming that the layer is free and that plane sections in the layer remain plane during necking (Emerman and Turcotte, 1984; Schmalholz et al., 2008). The horizontal extension rate can then be expressed by a change of the layer thickness, that is $D_{xx} = -\frac{1}{H}\frac{dH}{dt}$. The power-law constitutive equation is $D_{xx} = B\tau_{xx}^n$ where $\tau_{xx}$ is the horizontal deviatoric stress and $B$ is a material constant. Assuming in addition a constant horizontal extensional force, $F$ (in units N/m), the deviatoric stress is $\tau_{xx} = F/2H$ (note that the factor 2 appears again because force is related to total stress and for a free layer the deviatoric stress is half the total stress). Equating the two above expressions for $D_{xx}$ and using $\tau_{xx} = F/2H$ yields a nonlinear ordinary differential equation (ODE) for $H$

$$H^{n-1}\frac{dH}{dt} = -B\left(\frac{F}{2}\right)^n \tag{40}$$

Integrating both sides of the equation with respect to time and using the initial condition $H(t=0) = H_0$ yields the solution (Schmalholz, et al., 2008; Schmalholz, 2011)

$$\frac{H}{H_0} = \left(1 - n\frac{t}{t_C}\right)^{\frac{1}{n}} \tag{41}$$



where the characteristic time $t_c = 1/\left(B\tau_{xx0}^n\right)$ with $\tau_{xx0} = F/2H_0$. The thinning of the layer, quantified by $H/H_0$,

with progressive (dimensionless) time, $t/t_C$, depends only on $n$. Eq. (41) shows that when $t/t_C$ reaches the value

of $1/n$, then $H$ is zero and the layer has been separated by necking (Figure 17B). Hence, the time necessary to

separate a layer by necking is $t_N = 1/\left(nB\tau_{xx0}^n\right)$. For example, if a necking experiment for $\tau_{xx} = 250$ MPa would be

performed with Madoc dolomite following the flow law of Davis et al. (2008), given by $D_{xx} = \varepsilon\mu^{-n}e^{\frac{-Q}{RT}}\tau_{xx}^n = B\tau_{xx}^n$

with $\varepsilon = 10^{29}$ s$^{-1}$, $\mu = 45.6$ GPa, $Q = 420$ kJmol$^{-1}$, $R = 8.31$ kJmol$^{-1}$K$^{-1}$, $T = 800$ °C and $n = 7$, then $B = 8.1422 \times 10^{-67}$ Pa$^{-n}$s$^{-1}$ and it would take nearly a year to separate the dolomite by necking because $t_N = \sim328$ days.

The above simple analytical solution provides reasonably accurate results for the evolution of thinning (

$H/H_0$) with progressive extension up to $H/H_0 = 0.2$, for effective viscosity ratios larger than 100 (Schmalholz et

al., 2008). The evolution of thinning was also studied with numerical simulations (Schmalholz et al., 2008). The

results show that initially straight vertical (layer-orthogonal) passive lines across the layer remain straight and

vertical during necking (i.e. plane sections remain plane), which means that there is essentially no layer-parallel

shear around the necking region. Furthermore, the amplification rates of initial geometrical perturbations decrease

with increasing extension, similar to finite amplitude folding. Similar to folding, necking is also associated with

structural softening (Figure 18).

Solutions for the dominant wavelength and maximal amplification rate for necking are listed in Table 2.

### 3.1.2    Multilayer necking

Theoretical studies on multilayer necking are rare in the geological literature. Most analytical multilayer

necking studies have been applied to large-scale necking and lithospheric extension (see below). Most theoretical

studies have considered brittle boudinage in order to calculate the stress field in multilayers under extension or to

calculate the stress fields for layers with pre-existing vertical fractures, in which case the initial fracturing process

itself has not been investigated (Stromgard, 1973; Mandal et al., 2000).

Cobbold et al. (1971) showed that if the theory of internal instability for folding, as developed by Biot

(1957, 1964), is used for a compression direction orthogonal to the anisotropy orientation, then structures can form

that are similar to pinch-and-swell structure (they also used the term internal boudins).



### 3.1.3 Lithospheric necking

Artemjev and Artyushkov (1971) were probably the first to suggest that rift systems are caused by crustal

thinning due to a necking instability during lithospheric extension. It was subsequently shown that lithospheric

necking for slow spreading rates (1-3 cm/yr) is feasible for creep flow laws considered typical for the lithosphere

(Tapponnier and Francheteau, 1978). Later, the stability analysis (described above for folding) has been applied to

study necking instability during lithospheric extension (Fletcher and Hallet, 1983; Ricard and Froidevaux, 1986;

Zuber and Parmentier, 1986), including lithospheric models with two competent layers (upper crust and upper

mantle) separated by a weak lower crust (Zuber et al., 1986). Compared to small-scale necking models, models of

lithospheric necking are usually more complex because they consider (i) gravity, (ii) one or more competent layers

with a very high power-law stress exponent mimicking effectively plastic deformation, (iii) a viscosity which decays

exponentially with depth in the weak layers to mimic the temperature dependence and (iv) some kind of stress limit

to mimic the brittle yield strength of rock.

The impact of gravity on necking can be also quantified by an Argand number, that is, the dimensionless

ratio of gravitational stress to extensional stress (Fletcher and Hallet, 1983)

$$Ar_N = \frac{\Delta \rho g H}{2\tau_y} \qquad (42)$$

where $\Delta \rho$ is the density difference between the material below and above the competent layer, $g$ the gravitational

acceleration, $H$ the thickness of the competent layer and $\tau_y$ the representative extensional yield stress in the

competent layer. The Argand number $Ar_N$ is similar to the one that has been introduced by (England & McKenzie,

1982) to scale the gravitational stress to the horizontal stress during lithospheric thickening and is similar to the

Argand number applicable to lithospheric folding (Schmalholz et al., 2002). Fletcher and Hallet (1983) showed that

for a wide range of creep flow laws and an extension rate on the order of $10^{-15}$ s$^{-1}$, the necking instability is strong (

$\alpha_d > \sim 40$) and that the dominant wavelength $L_d = 30 - 90$ km. The ratio of $L_d$ to the depth of the brittle-ductile

transition (representing the thickness of the competent layer) ranges between 3 - 4 and typical values of $Ar_N$ are

between 2 - 6. Several subsequent studies have applied the stability analysis to study necking in an extending

lithosphere and showed, for example, the strong impact of the rheological assumptions and stratification on necking





(Bassi and Bonnin, 1988; Martinod and Davy, 1992). Fletcher and Hallet (2004) and Pollard and Fletcher (2005; their section 11.2.4) presented large-scale analytical necking solutions that also consider the effect of erosion, which is described by a diffusion-type law. Fletcher and Hallet (2004) show that erosion can significantly increase the

necking instability.

Recent studies on magma-poor rifted margins have identified so-called necking zones in which the crustal thickness is strongly reduced from a normal thickness of 30 – 35 km to about 5 – 10 km (Peron-Pinvidic and Manatschal, 2009). These necking domains separate the proximal domain from the hyperextended domains in which the continental crust is strongly thinned (e.g. Sutra et al., 2013). Recent studies on magma-poor margins indicate that

the continental lithosphere can be significantly extended and necked over several hundreds of kilometres without a lithospheric breakup that would result in the formation of new oceanic crust at a mid ocean ridge (Sutra et al., 2013). The observed width of necking zones ranges from 20 to 100 km for passive margins worldwide (Chenin, 2016). Within these necking zones the crustal thickness is reduced from about 30 km to about 10 km. Assuming that pre-rift (initial) geometrical perturbations of crustal thickness have an amplitude ($A_0$) on the order of 100 m requires an

amplification, $A / A_0$, of $10^5$ to thin an initially 30 km thick crust to 10 km. Using typical amplification rates (scaled by the bulk extension rate) for the continental lithosphere in the range 40 – 100 (Fletcher and Hallet, 1983), the bulk extension required to achieve $A / A_0 = 10^5$ can be calculated by the formula $\ln \left( A / A_0 \right) / \left( \alpha_d - 1 \right)$ (compare with Eq. (31)), which gives an extension of about 30% and 12% for amplification rates of 40 and 100, respectively. Applying these extension values to the 25 – 123 km range of dominant wavelengths derived by (Fletcher and Hallet, 1983) for

typical continental rocks provides a corresponding range of "extended" wavelengths between 28 – 166 km. The necking zone corresponds to half the extended wavelength and hence ranges between 14 – 83 km, which agrees with the observed widths of necking zones of 20 – 100 km (Chenin, 2016). The agreement between observed and predicted width of necking zones suggests that the observed necking zones at passive margins are indeed the result of mainly viscous necking.

Lithospheric extension, rifting and associated sedimentary basin formation in a number of regions worldwide have been attributed to mainly lithospheric necking, such as the region around the Porcupine and Rockhall basins in the southern North Atlantic (Mohn et al., 2014), the Baikal rift (Artemjev and Artyushkov, 1971) or the Western Mediterranean back-arc basin (Gueguen et al., 1997). Furthermore, most kinematic or semi-kinematic (including flexure) models of lithospheric thinning and associated sedimentary basin formation implicitly



assume a continuous necking of the lithosphere (McKenzie, 1978; Kooi et al., 1992). Such thinning models are of

practical importance for the assessment of hydrocarbon reservoir potential in extensional sedimentary basins (see

applications).

Necking has also been suggested to be the controlling process for slab detachment (Lister et al., 2008;

Duretz et al., 2012). Detachment of an oceanic slab usually occurs when the corresponding ocean is closed and

continental collision has started. The heavy oceanic slab is then hanging more or less vertically in the mantle and is

attached to the overlying continent. The downward extension is controlled by the negative buoyancy of the cold slab

in the warmer mantle. The analytical necking solution of Eq. (41) has been applied to show the feasibility of slab

detachment (by using the buoyancy as the driving force $F$) for the Hindukush region, as an example (Schmalholz,

2011). The simple analytical solution can accurately describe the thinning of the lithospheric slab during slab

detachment, which was also numerically simulated with 2D thermo-mechanical models considering

viscoelastoplastic rheologies, heat transfer by conduction and advection, and thermo-mechanical coupling by shear

heating (Duretz et al., 2012). The first-order agreement between the simple 1D analytical solution and the 2D

thermo-mechanical numerical solution indicates that the 1D necking solution captures the first-order processes of

slab detachment. The simple ODE in Eq. (40) was elaborated into a system of ODE's to study the impact of

coupling between grain-sensitive rheology and grain-size evolution with damage on slab detachment (Bercovici et

al., 2015). The more elaborated system of ODE's had to be solved numerically. Bercovici et al. (2015) show that

weakening due to grain size reduction and damage in polycrystalline rock can significantly accelerate necking and

hence slab detachment, so that detachment can occur in about 1 My.

**3.2    Experimental results**

There are fewer experimental studies on the development of viscous pinch-and-swell necking for several

reasons. Firstly, as shown theoretically by Smith (1975, 1977), Emerman and Turcotte (1984), and Eq. (37) above,

the dynamic growth rate of necking in linear viscous materials is zero. Whereas for folds the kinematic or passive

growth rate due to the homogeneous component of background strain is +1, reversing the sign of this bulk strain

relative to the layer also reverses the sign of the passive growth rate: in contrast to folds there is a passive

deamplification of initial perturbations due to stretching of the layer. It follows that viscous necking should not

develop in linear viscous materials as used in many analogue experiments. Smith (1977) did predict dynamic growth



of necks in strain-rate hardening power-law viscous materials, with the amplification rate increasing for higher

values of the power-law stress exponent, especially in the layer. However, there are technical difficulties with

developing necks in stiff power-law viscous layers in analogue experiments. As discussed above in relation to Eq.

(22), the effective viscosity in a power-law viscous material is a function of strain rate and, when the strain rate is

zero, the effective viscosity should be infinite. In theoretical studies such as that of Smith (1977), the layer is

infinitely long and the strain rate is taken as a given parameter. In an experiment, the layer is of finite length, is

poorly bonded to the model boundary, and initially has a zero strain rate. In pure shear folding experiments,

shortening directed along the length of the layer means that the layer ends have no alternative other than to move

inwards with the model walls. However, this constraint is not present when the walls extend away from the layer

ends in a pure shear necking experiment. In such a model configuration, the layer represents a finite-length inclusion

in a weaker matrix, equivalent to the case considered by Schmid et al. (2004) for folding of layers of finite length.

Extending the discussion of Mancktelow and Pennacchioni (2010) on isolated power-law inclusions, for strictly

power-law viscous materials, a finite-length layer with initially zero strain rate and infinite effective viscosity should

behave rigidly and detach from the model walls, with the matrix simply flowing around the layer ends. However,

because no material has a perfect power-law rheology and the effective viscosity is generally asymptotically limited

to a non-infinite value with decreasing strain rate, the situation is not as bad in practice as in theory. Also, following

Schmid et al. (2004), increasing the length to thickness of the modelled layer(s) is advantageous, but there are

realistic limits on the length of model rigs and long thin layers are more difficult to prepare accurately and to

observe in sufficiently fine detail.

Ramberg (1955) performed compression experiments orthogonal to layering of layered cakes of putty,

plasticene (sic), and cheese (!), with either 1D or 2D compensating extension. The resulting structures are similar to

natural boudinage and pinch-and-swell structure, but such models, like the models of Hall (1815) on folding, were

more illustrative than quantitative. Griggs and Handin (1960) studied the mechanisms of deep earthquakes and

performed extension experiments with natural rock, but necessarily scaling length, time and temperature. Amongst

others, they performed experiments with dolomite (Hasmark and Luning) and Eureka quartzite layers embedded in

Yule marble for confining pressures of 200 and 500 MPa (2 and 5 kbar) and temperatures of 800 °C. They showed

that, depending on the confining pressure and temperature, three macroscopic deformation processes take place:

extension fracturing, faulting and uniform flow (necking). Extension fracturing takes place in the brittle regime at



lower confining pressures and temperatures, faulting (i.e. shear failure) takes place at the transition between brittle

and ductile deformation and uniform flow in the ductile regime at higher confining pressures and temperatures. For

confining pressures of 200 and 500 MPa and 800 °C, the dolomite layers were necking while the quartzite layer was

fracturing.

In addition to their experiments on folding in power-law viscous materials, Neurath and Smith (1982) also

performed necking experiments. The measured amplification rates where significantly higher (a factor 2 - 3) than the

theoretical ones and they suggested that the discrepancy could be due to some kind of strain-softening, by which the

power law exponent would increase with increasing strain. They showed analytically that strain softening can be

described with an effective power-law stress exponent given in Eq. (39). Ghosh (1988) conducted experiments with

plaster of Paris resting on a substrate of pitch with equal stretching of the layer in all directions to investigate 2D

chocolate–tablet structure. The study was designed to consider the geometry during progressive development and

from the materials chosen could only develop brittle boudins rather than the pinch-and-swell structures considered

here. Kidan and Cosgrove (1996) used the same rig employed in earlier folding experiments (Cobbold 1975a,

1975b; Cobbold and Knowles, 1976) to investigate multi-layer boudinage, using layers of paraffin wax and

plasticine. Their experiments generally developed rectangular boudins due to (sequential) fracturing but internal

pinch-and-swell structure in some cases developed on a larger scale, reflecting the overall anisotropy.

        More recent experimental studies on boudinage in 2D or 3D have used specially designed rigs (Zulauf et

al., 2003) and power-law materials with high stress exponents, such as plasticine with n = ~7 (McClay, 1976; Zulauf

and Zulauf, 2004; Zulauf et al., 2011). As shown by Schöpfer and Zulauf (2002), plasticine never flows at steady

state but is strongly strain hardening, with the stress exponent also increasing (in some mixtures markedly) with

increasing strain. Both of these effects promote heterogeneous deformation and localization (Hobbs et al., 1986),

with the development of ductile shears, as noted by McClay (1976). Strain hardening is also a pre-requisite for the

onset of necking according to the Considère criterion, as discussed in detail above. The experiments of Schöpfer and

Zulauf (2002) with plasticine layers in a plasticine/oil mixture matrix developed pinch-and-swell structures even for

remarkably low effective viscosity contrasts of ~1.5, with more distinct boudins at ratios of ca. 2.0 -2.5. Their results

were consistent with the theoretical dominant wavelength predicted by Smith (1977) for such low effective viscosity

ratios. However, the experiments of Schöpfer and Zulauf (2002) also suggest that at these low viscosity ratios the



dominant wavelength is approximately constant. Only the boudin geometry is sensitive to the viscosity ratio, with

pinch-and-swell geometries developing at the lower values.

Marques et al. (2012) used layers of viscoelastoplastic clay or elastic soft paper in linear viscous PDMS

silicone putty to investigate the influence of layer thickness and bulk strain rate on the average boudin width for

brittle boudins. Although their natural measurements from south-west Portugal show a clear linear relationship

between layer thickness and boudin width, as would be expected from elastic theory, the average boudin width in

their experiments shows an exponential dependence on layer width and a power-law dependence on bulk strain rate.


## 4    Newer developments

### 4.1    Nonlinear terms in the folding equation and localized folding
Equations (5) and (8) for elastic and viscous folding, respectively, are linear and the corresponding solutions

are periodic, that is, they can be expressed with trigonometric functions such as cosine or sine. However, most

natural fold systems are not strictly periodic but irregular and sometimes localized. Localized folding is

characterized by the existence of large amplitudes only over a small region of a shortened layer (after Wadee, 1999).

The reason for irregular and localized fold geometries has been controversially discussed in the last decades (e.g.

Zhang et al., 1996; Mancktelow, 1999; Schmid et al., 2010; Hobbs and Ord, 2012). There are essentially two

fundamental reasons for irregular and localized fold geometries: (1) geometrical heterogeneities (including material

heterogeneities), and (2) material softening.

          Considering the first reason, if linear equations for folding are considered, then irregular and localized fold

geometries can result from (i) an irregular and localized initial geometry of the layer or (ii) from non-homogeneous

material properties. Concerning (i), in the thin-plate approach one usually assumes that the layer has initially a

constant thickness but that the layer is initially not perfectly straight, for example, it can have the shape of a bell-

shaped function (Eq. (26); Figure 16). Stability analysis can consider initial irregularities either as deviation from a

straight layer having constant thickness or as initial variations in the layer thickness. The thin-plate approach and the

stability analysis can also include the impact of nonlinear rheologies, such as a power-law flow that is strain-rate

hardening (stress increases with increasing strain rate), but these flow laws are in practice linearized to provide

accurate solutions provided fold amplitudes are small (i.e. limb dips smaller than ~20 degrees; Chapple, 1968;

Schmalholz, 2006). Analytical results, numerical simulations and analogue experiments have shown that initial



irregular layer geometries can generate a wide variety of irregular, non-periodic and localized fold geometries (e.g. Cobbold, 1975a; Abbassi and Mancktelow, 1990, 1992; Mancktelow and Abbassi 1992; Mancktelow, 1999, 2001; Schmalholz, 2006; Schmid et al. 2010). Concerning (ii), heterogeneities, such as stronger or weaker inclusions, in a perfectly straight layer or in the matrix close to the layer will cause local perturbations of the deformation and hence

cause local deformations in the layer, which in turn cause a deviation from the straight geometry. These geometric variations can then also cause irregular or localized fold shapes. One of the first localized folding solutions was presented by Smoluchowski (1909) for folding of an elastic layer under gravity. The solution is described by a sinusoidal waveform whose amplitude at one end of the layer is exponentially decaying along the (one-sided) infinite layer. The initial amplitude at one end of the layer is interpreted as result of local deviations from isostatic

equilibrium (Smoluchowski, 1909).

Considering the second reason, if nonlinear equations for folding are considered then a much wider spectrum of solutions is possible. Non-linearities arise essentially due to two reasons: geometrical nonlinearities or material nonlinearities. Geometrical nonlinearities have been considered to describe the finite amplitude evolution because the linear solutions based on exponential amplitude growth break down when fold limb dips exceed ~20 degrees.

Material nonlinearities, such as due to power-law flow laws, are often linearized and, as mentioned above, the resulting fold geometries can be explained by the corresponding linearized equations. Therefore, the fundamental finite amplitude fold geometries (i.e. regular or localized) due to geometrical and material non-linearities for hardening behaviour can be well estimated with linearized equations because these linearised equations are valid up to limb dips of ~20 degrees.  However, other types of non-linearities have also been studied with the thin-plate

approach, whereby the resistance of the material in which the layer is embedded (often termed the matrix or foundation) is assumed to be nonlinear. The linear term for the matrix resistance in Eq. (5) is usually $q = kA$ but in the nonlinear analysis it is usually expanded to $q = kA + c_1 A^2 + c_2 A^3 + \dots$ (Tvergaard and Needleman, 1980; Wadee, 1999; Hunt et al., 2000). Typically, expressions like $q = kA - cA^2$ or $q = kA - cA^3$ have been considered where $c$ is a constant. These nonlinearities describe a material softening of the matrix resistance because the matrix resistance

becomes smaller as the amplitude becomes larger. Also nonlinear and viscoelastic behaviour of the matrix has been investigated using $q = 4G \dfrac{2\pi}{L} \left( \dfrac{d/dt}{d/dt + G/\eta_M} \right) \left( A - cA^3 \right)$, where G is the shear modulus of the embedding medium and $c$ is a positive constant (Hunt et al., 1996). Some of these nonlinear folding equations are mathematically



identical to the nonlinear equations that have been studied in the framework of nonlinear dynamics. For example, the

nonlinear ODE for the buckling of a free elastic beam (i.e. Eq. (3) in which the deflection is quantified by $\theta$, that is,

the angle between the horizontal x-direction and the beam and not by amplitude $A$) is

$$D\frac{d^2\theta}{dx^2} + F\sin(\theta) \tag{43}$$

This nonlinear (due to the sinus function) folding equation is mathematically similar to the nonlinear ODE

describing a pendulum (Hunt et al., 1989)

$$mL\frac{d^2\theta}{dt^2} + mg\sin(\theta) \tag{44}$$

where $\theta$ is the deviation from the vertical direction of gravity, $m$ is the mass, $L$ the length of the pendulum and $g$

the gravitational acceleration. Eqs. (43) and (44) become exactly equivalent if we identify $D$ as $mL$, $F$ as $mg$ and $x$ as

$t$ (Hunt et al., 1989). This equivalence between the deformation of a beam and the motion of a pendulum can be

traced back to Kirchhoff (1859) and is known as Kirchhoff's kinetic analogue (see also Love, 1927). The pendulum

equation is a simple example of a nonlinear dynamical system, which is typically described by a system of nonlinear

ODEs and the derivatives are typical time derivatives. There exist many mathematical tools to describe and quantify

the behaviour of dynamical systems, such as phase plane, phase paths, limit cycles or homoclinic orbits (e.g. Jordan

and Smith, 1999). These tools are useful to describe the behaviour of a dynamical system without actually explicitly

solving the nonlinear ODE. Also, the so-called chaos theory is based on the analysis of dynamical systems (e.g.

Guckenheimer and Holmes, 1983). Because of the mathematical equivalence between equations describing

dynamical systems and folding of beams, the folding equations including nonlinear terms for the matrix resistance

have been analysed with the tools of phase plane etc. as mentioned above (e.g. Champneys, 1998; Hunt et al., 1989).

Furthermore, some solutions for these nonlinear folding equations can also be expressed with non-periodic functions

such as hyperbolic secant ( $\mathrm{sech}(x) = 1/\cosh(x) = 2/(\mathrm{e}^x + \mathrm{e}^{-x})$ ), which is also a solution for solitary waves (or so-

called solitons; e.g. Drazin and Johnson, 1989).

Geometrical and material heterogeneities are intuitive reasons for observed irregular fold geometries

because natural rock layers are never perfectly straight or homogeneous before folding. Geometrical nonlinearities

are intrinsic for folding because they arise naturally due to the deviations of the folded layer from the initially

straight layer. Linearized equations can predict the fold shapes up to amplitudes for which the final irregularities can



already be seen, such as for an initial bell-shaped perturbation (Figure 16). Nonlinearities due to material softening,

such as a nonlinear matrix resistance, are more difficult to justify, and especially quantify, in a straightforward

manner. Nonlinear matrix resistance is usually justified by some kind of material strain softening (e.g. Hobbs and

Ord, 2012). However, this softening process is usually defined *a priori* and it is not clear what micromechanical

processes actually causes such particular nonlinearities related to softening. Typical candidates responsible for

softening are, for example, grain size reduction, mineral reactions or fluid-rock interaction. The impact of strain-rate

softening on folding has been investigated also with numerical simulations (e.g. Hobbs et al., 2011)

### 4.2    Numerical simulations and coupled models

This review focuses on analytical solutions, with some reference to the analogue models that were often used

to qualitatively or (semi-)quantitatively test these analytical solutions. However, since the late 1960s more and more

numerical studies of folding and necking have been performed. One of the first numerical simulations of folding in a

geological context was carried out by Dietrich (1969) and Dieterich and Carter (1969) and Stephansson and Berner

(1971) already applied the finite element method to various tectonic processes such as folding, deformation of

isolated boudins, isostatic adjustment and spreading at the mid-Atlantic ridge.

Numerical simulations are essential to study folding and necking scenarios for which analytical solutions

cannot be derived or for which only approximate analytical solutions exist. Such scenarios are for example (i) the

finite amplitude evolution of folding and necking in 2D and 3D for which only approximate analytical solutions are

available (Chapple, 1968; Kaus and Schmalholz, 2006; Schmalholz, 2006; Schmalholz et al., 2008; Schmid et al.,

2008; Grasemann and Schmalholz, 2012; Fernandez and Kaus, 2014; Frehner, 2014; von Tscharner et al., 2016), or

(ii) the numerical solution of nonlinear folding equations (see section 4.1.; e.g. Hunt et al., 1997; Wadee, 1999). A

typical application for numerical simulations is, for example, the study of fold propagation (or serial folding) where

folding in a competent layer starts from a localized geometrical perturbation and new folds develop sequentially

away from the initial perturbation along the layer. Such fold propagation has been studied in 2D in single- (e.g.

Cobbold, 1977; Mancktelow, 1999; Zhang et al., 2000) and multilayers (Schmalholz and Schmid, 2012) and in 3D

in single layers (Frehner, 2014).

Many numerical simulations of folding consider a layer-parallel compression of the layers and the bulk

deformation of the model is close to pure shear. Folding of layers under bulk simple shear has been studied





numerically for single layers (Viola and Mancktelow, 2005; Llorens et al., 2013a) and multilayers (Schmalholz and

Schmid, 2012, Llorens et al., 2013b). A main result of the simple shear studies is that folding under bulk simple

shear does not generate asymmetric fold shapes but more or less symmetric fold shapes similar to the ones generated

1265 under bulk pure shear (cf. Lister and Williams, 1983). Also, when layers rotate in a simple-shear zone they can be

first shortened until the fold train is more or less orthogonal to the simple-shear zone. Further shear and rotation,

however, extends the fold train which can unfold the layers again (Llorens, et al., 2013b). Laboratory experiments of

such single layer folding and unfolding under bulk simple shear have been already performed by Ramberg (1959).

 For active folding, a continuous competent layer is actually not required. Adamuszek et al. (2013a) showed

1270 that it is sufficient for folding to have competent inclusions (can be of various size) aligned and clustered in a way to

form a "layer" of inclusion-clusters. If this "layer" of inclusions is embedded in a weaker viscous medium then the

layer-parallel shortening also generates folding of the "layer" consisting of individual inclusions.

 Numerical simulations of the extension of power-law viscoplastic (von Mises; Schmalholz and Maeder, 2012)

and power-law viscous (Duretz and Schmalholz, 2015) multilayers embedded in weaker power-law viscous medium

1275 showed the formation of individual shear zones that crosscut the entire multilayer. The shear zones form after some

period of distributed multilayer necking and only occur (i) when the weak inter-layers are power-law viscous and (ii)

when the spacing between the strong layers is less than or approximately equal to the thickness of the strong layers.

The shear zones crosscutting the entire multilayer form due to the alignment of individual necks in different layers,

which is a finite amplitude effect when individual necking zones can form a connected network of weak zones.


 The numerical studies mentioned above investigated fundamental mechanical folding and necking processes,

but numerical solutions are also useful to study the coupling of folding and necking with other processes such as (i)

the generation of heat during folding due to dissipative rock deformation (shear heating) and the related thermal-

softening caused by thermo-mechanical feed-back with temperature-dependent rock viscosity (Hobbs et al., 2007,

1285 2008; Burg and Schmalholz, 2008), (ii) the conversion of macroscale mechanical work into microscale mechanical

work during the reduction and growth of mineral grain size and related softening due to grain size reduction (Peters

et al., 2015), (iii) the impact of metamorphic reactions on rock deformation (Hobbs et al., 2010), (iv) coupling of

crustal folding or necking with erosion in 2D (Burg and Podladchikov, 2000; Burov and Poliakov, 2001) and 3D

(Collignon et al., 2015), or (v) the coupling of folding with salt diapirism (Fernandez and Kaus, 2014). A detailed





outline of a coupled thermodynamic approach to study rock deformation and the resulting structures is given in the recent textbook by Hobbs and Ord (2014). The impact of shear heating and grain size reduction on lithospheric folding and necking can be significant because both processes cause a mechanical softening of the rock (e.g. Regenauer-Lieb and Yuen, 1998; Regenauer-Lieb et al., 2006). For example, during shortening of the continental lithosphere, shear heating and thermal softening can cause a transition from distributed folding to localized ductile

thrusting (Burg and Schmalholz, 2008; Schmalholz et al., 2009; Jaquet et al., 2016).

       Numerical simulations are based on a basic set of partial differential equations resulting from the concepts of continuum mechanics. These equations are useful to describe continuous deformation and strain localization by shear bands (with no loss of velocity continuity). Elaborated numerical algorithms based on continuum mechanics,

the so-called extended finite element method or XFEM (Belytschko et al., 2001), are additionally able to model discontinuous fracture, for example due to 3D folding (Jäger et al., 2008). In geological studies it is more common to apply so-called discrete element methods to study brittle deformation and fracturing. In simple words, these discrete models assume that a material consists of particles which are connected by elastic springs. The force balance is controlled by Newton's law (force equals mass times acceleration) which is an ODE (no spatial

derivatives) and not a PDE. A fracture appears when the stress in a spring connecting two particles exceeds the yield strength and the spring connection between the two corresponding particles is then removed. Discrete element modelling has been applied, for example, to study fracturing during detachment folding (Hardy and Finch, 2005) or to study the evolution of brittle boudinage in 2D and 3D (Abe and Urai, 2012; Abe et al., 2013).

Recently, Adamuszek et al. (2016) developed the MATLAB© based software termed Folder, which can be used to numerically model folding and necking in power-law viscous single- and multilayers. Folder is freely available under http://geofolder.sourceforge.net. Folder also includes all relevant analytical solutions for the amplification rates for folding and necking. Figure 12 and Figure 13 have been generated with the results of Folder. Folder is easy-to-use software with a user-friendly graphical interface. 200 years after the analogue experiments of

James Hall, any student or researcher in geology can now easily perform similar experiments on his/her personal computer.





## 5    Fundamental similarities and differences between folding and necking

### 5.1    Similarities

The stability analysis (Fletcher, 1974; Smith, 1975, 1977) can be used to study the initial (small amplitude)

stages of both folding and necking. Folding and necking result from the same type of mechanical instability. This

instability causes initial geometrical perturbations on the layer interface to amplify with velocities that are faster

than the velocities corresponding to the applied bulk deformation (e.g. pure shear). The dominant wavelength for

folding and necking is identical for the same material parameters (Figure 14). Amplification rates for folding and

necking increase with increasing viscosity ratio and with increasing power-law stress exponent in both the layer and

the embedding medium (Figure 14).

Folding and necking are processes that can take place in single and multilayers and can also act on all scales.

For large-scale folding and necking, gravity decreases the intensity of the folding and necking instabilities. The

impact of gravity on folding and necking is usually quantified by some kind of Argand number, which is the ratio of

the gravitational stress to the layer-parallel stress driving compression or extension, respectively.

Folding and necking are both associated with structural softening (also termed geometric softening or

geometric weakening; Figure 18). The effective viscosity of a rock unit consisting of competent layers embedded in

a weaker medium during shortening and extension with a constant bulk rate of deformation can be calculated by the

ratio of the area-averaged stress to the bulk rate of deformation. If the shortening and extension would be

homogeneous pure shear at a constant rate and the layer would deform by homogeneous thickening and thinning,

then the effective viscosity of the layered rock unit would remain constant. However, if folding or necking takes

place the effective viscosity decreases during bulk shortening and extension, respectively, because the area-averaged

stresses are smaller during folding and necking than during pure shear thickening and thinning. Related to the stress

decrease is a decrease in dissipation (stress times strain rate) and the integral of the dissipation over the time

(duration) of the deformation represents the work (or energy) necessary to perform the deformation. The structural

softening related to folding and necking hence reduces the mechanical work required to deform the layered rock

unit. Therefore, folding and necking are the preferred deformation modes of mechanically layered rock units

because folding and necking minimizes the work required for the deformation. During structural softening the

material properties remain constant and for both linear and power-law viscous material the flow laws are always

strain-rate hardening, that is, the stress increases with increasing strain rate. Hence, structural softening is

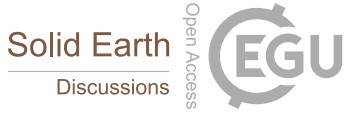

fundamentally different to material softening, where some material property (e.g. cohesion, friction angle or effective viscosity) decreases with progressive strain.

### 5.2 Differences

During folding the layer thickness remains more or less constant and the shortening is compensated by a lateral deflection (Figure 4). During necking the local variation in layer thickness is significant and the extension is compensated by localised thinning of the layer while the central axis of the layer remains more or less straight (Figure 4).

    In folding, a particular wavelength can be selected and "locked in" if the fold arc length does not vary

significantly during fold amplification, which is the case for large viscosity ratios (> ~100). During necking a wavelength cannot be "locked in" because the necking zone is continuously extending during bulk extension.

    The maximal amplification rates for folding and necking for the same material parameters are significantly different (Figure 14). Amplification rates for folding are larger than the ones for necking for the same material parameters. While folding occurs for linear and power-law viscous rheologies, necking only occurs for power-law

viscous rheologies. Since the amplification rates for necking are smaller than the ones for folding, significant necking ($\alpha_d > 10$) in rock requires higher viscosity ratios and/or higher power-law stress exponents than significant folding. Hence, the range of material parameters for significant necking is smaller than the parameter range for significant folding (see text below Eq. (31)).

    The yield stress for brittle fracture is typically described by a Mohr-Coulomb failure criterion, which is based

on parameters determined by Byerlee (1978; Byerlee's law). These yield stresses are significantly larger during compression than during extension (Sibson, 1974), due to the implicit development of dynamic over- and underpressure respectively (Mancktelow 1993, 2008). Hence, layers under layer-parallel compression can deform viscously up to much larger stresses than layers under layer-parallel extension before fracture occurs. The available stress range for folding without fracturing is therefore much larger than the stress range for necking without

fracturing.

    The range of material parameters and flow stresses for significant necking is significantly smaller than the corresponding range for folding, which may be the main reason why pinch-and-swell structure is less frequent in nature than folding, and also why brittle boudinage is more frequent than pinch-and-swell structures.



## 6    Some applications

The main direct applications of analytical and numerical solutions for folding are the estimation of (i) the bulk shortening that was necessary to generate an observed fold and (ii) the viscosity ratio during the formation of the observed fold (Sherwin and Chapple, 1968; Talbot, 1999; Hudleston and Treagus, 2010). Natural rock viscosities are commonly estimated using laboratory derived flow laws but the extrapolation from laboratory ($10^{-4}$ – $10^{-6}$ s$^{-1}$) to tectonic ($10^{-12}$ - $10^{-15}$ s$^{-1}$) strain rates makes such estimates uncertain. Also, flow laws are mainly determined for single minerals or specific rock types whereas natural, polymineralic rocks are usually more heterogeneous. Therefore, independent viscosity estimates based on, for example, analysis of isostatic rebound (e.g. Haskell, 1937), mullions (Kenis et al., 2004) or fold structures are useful to test viscosity estimates based on laboratory-derived flow laws (e.g. Karato, 2008). Observed single-layer fold geometries in folded veins of calcite, quartz and pegmatite on the mm to cm scale suggest that average effective viscosity ratios are between 17 – 70 and power-law exponents of the layer are between 2 – 5 (Hudleston and Treagus, 2010). Schmalholz and Podladchikov (2001) presented a diagram that enables the bulk shortening and viscosity ratio to be estimated from measured values of $A/L$ and $H/L$ for single layer folds. Adamuszek et al. (2011) developed a MATLAB© based software, the fold geometry toolbox, which determines automatically the values of $A/L$ and $H/L$ from fold shapes digitized from fold photos. Yakovlev (2012 and references therein) also presented a method to estimate bulk shortening from fold shapes and further developed a method to reconstruct the tectonic evolution of folded regions, which he mainly applied to the Caucasus. A problem with the estimation of viscosity ratio from folds is that the same amplification rates and hence similar fold amplifications can be obtained for different combinations of viscosity ratio and power-law exponents (Figure 14A). Lan and Hudleston (1995) presented a method to estimate the power-law exponent from observed fold shapes. Estimates of the viscosity ratios from fold shapes in combination with microstructural analysis have also been applied to estimate the stress levels during folding (Trepmann and Stöckhert, 2009). Trepmann and Stöckhert (2009) estimated that stresses in folded quartz veins in fine-grained high pressure–low temperature metamorphic greywackes of the Franciscan Subduction Complex at Pacheco Pass, California, to have been between 100 and 400 MPa.



Fold geometries can also be used to estimate the dominant folding mechanism. Schmalholz et al. (2002)

distinguished three types of folding mechanism depending on the controlling material parameters: (i) matrix-

controlled folding (controlled by viscosity ratio between layer and matrix), (ii) detachment folding (controlled by the

thickness of the weak layer below a strong layer), and (iii) gravity folding (controlled by the ratio of gravity to

viscous stress, that is, the Argand number, Eq. (34)). They presented a diagram which allows estimation of the

dominant folding mechanism from the fold geometry alone.

         Numerical simulations of necking have shown that during necking initially straight and vertical lines

remain vertical and straight (Schmalholz et al., 2008). This feature justifies the application of thermo-kinematic

models to lithospheric necking and the associated formation of sedimentary basins (e.g. McKenzie, 1978; Kooi et

al., 1992). These thermo-kinematic models subdivide the lithosphere laterally into a series of vertical columns

whose independent thinning is quantified by thinning factors. Such models have been applied to reconstruct the

thermo-tectonic history of extensional sedimentary basins, which is useful to evaluate the potential of hydrocarbon

reservoirs. 2D thermo-kinematic models of lithospheric extension are significantly faster to compute than 2D

thermo-mechanical models and can hence be used efficiently in combination with automated inversion or

optimization methods (Poplavskii et al., 2001; White and Bellingham, 2002; Ruepke et al., 2008).

The mathematical solutions for folding and necking have also been used to assess the deformation style of

the outer shell of the moons of Jupiter. Dombard and McKinnon (2001) investigated the grooved terrain of

Ganymede and argued that the regular structural periodicity found in this grooved terrain could be the result of an

extensional necking instability. Dombard and McKinnon (2006) also argued that topographic undulation, with a ca.

25 km wavelength, observed on Jupiter's icy moon Europe could be due to contractional folding.


## 7    Summary and conclusions

         Significant progress has been made in understanding and quantifying the mechanical processes of folding and

necking since the pioneering folding experiments of Hall and the pinch-and-swell observations of Ramsay. The

geometry and mechanical evolution of many fold trains can be explained by the dominant wavelength theory of Biot

and Ramberg and its elaboration to power-law viscous rheology by Fletcher and Smith. Folding and necking in

viscous layers are the result of a hydrodynamic instability. Folding and necking takes place because these processes

minimize the mechanical work required to shorten or extend mechanically-layered rock on all scales. The most important quantities to analyse folding and necking are the dominant wavelength and the corresponding maximal amplification rate. The two quantities allow the estimation of fundamental parameters relevant for folding and

necking, such as the effective viscosity ratio or the Argand number, and also allow an evaluation of whether folding or necking instabilities are sufficient to generate observable fold or pinch-and-swell structures.

Folding and necking instabilities are most likely always active when ductile, layered rocks are shortened or extended on all scales. Observable folds and necks (pinch-and-swell structure) are usually generated when the amplification rate, $\alpha$, is more than an order of magnitude larger than the absolute value of the bulk deformation

rate, $\left| \bar{D}_{xx} \right|$.

Folds are more frequent in nature than pinch-and-swell structure because folding can occur in layered rock that deform according to viscous and power-law viscous flow laws while necking only occurs in rock with power-law viscous behaviour. Furthermore, stresses during folding (compression) can be significantly larger than stresses during necking (extension) before the rock fails by fracture. Hence, brittle boudinage is more frequently observed

than continuous necking.

Future challenges are to quantify the coupling of folding and necking with other processes acting during rock deformation such as fracturing, shear heating, grain-size evolution, fluid-flow and metamorphic reactions. The concept of continuum mechanics can provide the system of equations that describes these coupled processes and numerical algorithms will be able to solve these equations. However, these equations and related numerical

simulations will include many parameters and one of the biggest challenges may be to determine the controlling parameters (e.g. via dimensional analysis) and to make the coupled thermodynamic processes comprehensible. In that sense, one of the main objectives for future research on folding and necking fits well with the famous statement of J. Willard Gibbs (1880, acceptance letter of Rumford Prize):"One of the principal objectives of theoretical research in any department of knowledge is to find the point of view from which the subject appears in its greatest

simplicity".

**Acknowledgements**

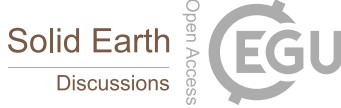

We thank Editor Susanne Buiter for her invitation to contribute this review to Solid Earth. SMS is grateful for many lessons, discussions and constructive critics of Yuri Podladchikov over the last 20 years. SMS thanks Sergei

Medvedev for his valuable help in improving appendix 1. SMS is also grateful for many fruitful and stimulating discussions on folding and necking with Ray Fletcher, Dani Schmid, Sergei Medvedev, Marcel Frehner, Marcin Dabrowski and Marta Adamuszek over the years. NSM thanks John Ramsay for firing an interest in the beauty of folds and boudins, and in the use of analogue models to study their development. NSM also thanks Yuri Podladchikov, Daniel Schmid, Boris Kaus, Guy Simpson and Marcel Frehner for discussion and feedback on

numerical modelling methods in general and on mechanisms of folding and boudinage.





**Appendix**

**Appendix 1: Derivation of the thin-plate equation from the 2D force balance equation**

Mathematical folding studies either use the thin-plate equation or the stability analysis, which is based on a

stream function solution for the full 2D force balance equations. We show here how the thin-plate equation can be

derived from the full 2D force balance equation, based on the derivation of a general extended thin-sheet equation

by Medvedev and Podladchikov (1999a, b). The thin-plate equation for folding is essentially derived by vertical

integration and approximation of the 2D equilibrium equations.

The equilibrium equations in 2D without gravity are

$$\frac{\partial \sigma_{xx}}{\partial x} + \frac{\partial \sigma_{xy}}{\partial y} = 0 \qquad\qquad\qquad \text{(A.1)}$$

$$\frac{\partial \sigma_{xy}}{\partial x} + \frac{\partial \sigma_{yy}}{\partial y} = 0 \qquad\qquad\qquad \text{(A.2)}$$

where $x$, $y$, $\sigma_{xx}$, $\sigma_{yy}$ and $\sigma_{xy}$ are the horizontal (layer-parallel) coordinate, the vertical coordinate, the horizontal

total stress, the vertical total stress and the shear stress, respectively. We apply the equilibrium equations to a layer

whose width in the x-direction is larger than its thickness in the y-direction. The bottom boundary ($Sb(x)$) and the

top boundary ($St(x)$) of the layer are described by continuous functions along the x-direction. The stresses along

the layer boundaries can be related to tractions on the top boundary $\vec{T_t} = \{T_{tx}, T_{ty}\}$ and on the bottom boundary

$\vec{T_b} = \{T_{bx}, T_{by}\}$. The tractions are related to the stress tensor for the top, $\boldsymbol{\sigma}|_{St}$, and bottom, $\boldsymbol{\sigma}|_{Sb}$, layer boundaries and

the outer unit normal vectors on both boundaries, $\vec{n_t}$ and $\vec{n_b}$, by the Cauchy formula:

$$\begin{aligned}\vec{T_t} &= \boldsymbol{\sigma}|_{St} \cdot \vec{n_t} \\ \vec{T_b} &= \boldsymbol{\sigma}|_{Sb} \cdot \vec{n_b}\end{aligned} \qquad\qquad \text{(A.3)}$$

The components of the outward unite normal vectors can be approximated for small slopes (i.e. dropping square root

terms) on the layer boundaries by





$$\vec{n}_t = \left\{ -\frac{\partial St(x)}{\partial x}, 1 \right\} \bigg/ \sqrt{\left( \frac{\partial St(x)}{\partial x} \right)^2 + 1} \approx \left\{ -\frac{\partial St(x)}{\partial x}, 1 \right\}$$

$$\vec{n}_b = \left\{ \frac{\partial Sb(x)}{\partial x}, -1 \right\} \bigg/ \sqrt{\left( \frac{\partial Sb(x)}{\partial x} \right)^2 + 1} \approx \left\{ \frac{\partial Sb(x)}{\partial x}, -1 \right\}$$

(A.4)

The components of the tractions at the top and bottom layer boundaries can then be expressed as

$$T_{tx} = -\sigma_{xx}\big|_{St(x)} \frac{\partial St(x)}{\partial x} + \sigma_{xy}\big|_{St(x)}$$

$$T_{bx} = \sigma_{xx}\big|_{Sb(x)} \frac{\partial Sb(x)}{\partial x} - \sigma_{xy}\big|_{Sb(x)}$$

$$T_{ty} = \sigma_{yy}\big|_{St(x)} - \sigma_{xy}\big|_{St(x)} \frac{\partial St(x)}{\partial x}$$

$$T_{by} = -\sigma_{yy}\big|_{Sb(x)} + \sigma_{xy}\big|_{Sb(x)} \frac{\partial Sb(x)}{\partial x}$$

(A.5)

Vertical integration of Eq. (A.2) while changing the order of integration and differentiation by using the rules of

differentiation of integrals with variable integration boundaries (Bronstein et al., 1997) yields

$$\int_{Sb(x)}^{St(x)} \frac{\partial \sigma_{xy}}{\partial x} dy + \sigma_{yy}\big|_{Sb(x)}^{St(x)} = \frac{\partial}{\partial x}\left( \int_{Sb(x)}^{St(x)} \sigma_{xy} dy \right) - \frac{\partial St(x)}{\partial x}\sigma_{xy}\big|_{St(x)} + \frac{\partial Sb(x)}{\partial x}\sigma_{xy}\big|_{Sb(x)} + \sigma_{yy}\big|_{St(x)} - \sigma_{yy}\big|_{Sb(x)} = 0 \quad \text{(A.6)}$$

Using the formulas for the components of the tractions (Eq. (A.5)) in Eq. (A.6) yields

$$\frac{\partial}{\partial x}\left( \int_{Sb(x)}^{St(x)} \sigma_{xy} dy \right) + T_{ty} + T_{ty} = 0$$

(A.7)

Similarly, vertical integration of the horizontal equilibrium Eq. (A.1) yields

$$\frac{\partial}{\partial x}\left( \int_{Sb(x)}^{St(x)} \sigma_{xx} dy \right) + T_{tx} + T_{bx} = 0$$

(A.8)

The integral in the first term in Eq. (A.7) can be written in different form using the rules of integration by parts


$$\int_{Sb(x)}^{St(x)} \sigma_{xy} dy = -\int_{Sb(x)}^{St(x)} (y - A)\frac{\partial \sigma_{xy}}{\partial y} dy + (y - A)\sigma_{xy}\big|_{St(x)} - (y - A)\sigma_{xy}\big|_{Sb(x)}$$

(A.9)

Integration by parts of two functions $u$ and $v$ can be generally expressed as

$$\int_a^b u\frac{dv}{dy} dy = -\int_a^b \frac{du}{dy} v dy + uv\big|_a^b$$

(A.10)





In Eq. (A.9) $\sigma_{xy}$ represents $u$ and $y - A$ represents $v$. The $y - A$ is the distance from the middle line of the layer, $A$, in the $y$-direction and

$$A = \frac{St(x) + Sb(x)}{2} \tag{A.11}$$

To find a relation between the vertically integrated vertical gradient of the shear stress and the vertically integrated horizontal gradient of the normal stress, we multiply Eq. (A.1) by $y - A$, integrate it vertically and apply the product rule of differentiation, which provides

$$-\int_{Sb(x)}^{St(x)} (y-A)\frac{\partial \sigma_{xy}}{\partial y}dy = \int_{Sb(x)}^{St(x)} (y-A)\frac{\partial \sigma_{xx}}{\partial x}dy =$$
$$\int_{Sb(x)}^{St(x)} \left( \frac{\partial}{\partial x}\big( (y-A)\sigma_{xx} \big) - \frac{\partial(y-A)}{\partial x}\sigma_{xx} \right) dy = \int_{Sb(x)}^{St(x)} \frac{\partial}{\partial x}\big( (y-A)\sigma_{xx} \big) dy + \int_{Sb(x)}^{St(x)} \frac{\partial A}{\partial x}\sigma_{xx}dy \tag{A.12}$$

The right hand side of Eq. (A.12) can be further modified by changing the order of integration and differentiation to

$$\int_{Sb(x)}^{St(x)} \frac{\partial}{\partial x}\big( (y-A)\sigma_{xx} \big) dy + \int_{Sb(x)}^{St(x)} \frac{\partial A}{\partial x}\sigma_{xx}dy =$$
$$\frac{\partial}{\partial x}\int_{Sb(x)}^{St(x)} \big( (y-A)\sigma_{xx} \big) dy - \frac{\partial St(x)}{\partial x}(y-A)\sigma_{xx}\Big|_{St(x)} + \frac{\partial Sb(x)}{\partial x}(y-A)\sigma_{xx}\Big|_{Sb(x)} + \frac{\partial A}{\partial x}\int_{Sb(x)}^{St(x)} \sigma_{xx}dy \tag{A.13}$$

The integral in Eq. (A.7) can be finally expressed as

$$\int_{Sb(x)}^{St(x)} \sigma_{xy}dy = \frac{\partial}{\partial x}\int_{Sb(x)}^{St(x)} \big( (y-A)\sigma_{xx} \big) dy + \frac{\partial A}{\partial x}\int_{Sb(x)}^{St(x)} \sigma_{xx}dy +$$
$$\frac{H}{2}\left( \sigma_{xy}\Big|_{St(x)} + \sigma_{xy}\Big|_{Sb(x)} - \frac{\partial St(x)}{\partial x}\sigma_{xx}\Big|_{St(x)} - \frac{\partial Sb(x)}{\partial x}\sigma_{xx}\Big|_{Sb(x)} \right) =$$
$$\frac{\partial}{\partial x}\int_{Sb(x)}^{St(x)} \big( (y-A)\sigma_{xx} \big) dy + \frac{\partial A}{\partial x}\int_{Sb(x)}^{St(x)} \sigma_{xx}dy + \frac{H}{2}\big( T_{tx} - T_{bx} \big) \tag{A.14}$$

where $H/2 = St(x) - A = -\big[ Sb(x) - A \big]$. Substituting Eq. (A.14) into Eq. (A.7) yields

$$\frac{\partial^2}{\partial x^2}\left( \int_{Sb(x)}^{St(x)} (y-A)\sigma_{xx}dy \right) + \frac{\partial}{\partial x}\left( \frac{\partial A}{\partial x}\int_{Sb(x)}^{St(x)} \sigma_{xx}dy \right) + \frac{\partial}{\partial x}\left( \frac{H}{2}\big( T_{tx} - T_{bx} \big) \right) + T_{ty} + T_{by} = 0 \tag{A.15}$$

Expanding the second term in Eq. (A.15) and using Eq. (A.8) provides





$$\frac{\partial^2}{\partial x^2}\left(\int_{Sb(x)}^{St(x)}(y-A)\sigma_{xx}dy\right)+\frac{\partial^2 A}{\partial x^2}\left(\int_{Sb(x)}^{St(x)}\sigma_{xx}dy\right)-\frac{\partial A}{\partial x}\left(T_{tx}+T_{bx}\right)+\frac{\partial}{\partial x}\left(\frac{H}{2}(T_{tx}-T_{bx})\right)+T_{ty}+T_{by}=0 \qquad \text{(A.16)}$$

Eq. (A.16) includes effects of both horizontal and vertical tractions on the layer boundaries and the only assumptions

made so far are that slopes on the layer boundaries are small so that square root terms in Eq. (A.4) are negligible.

The thin-plate approach of Biot (1961) assumes that only vertical tractions act on the layer boundaries, that

horizontal tractions are negligible and that $H$ = constant. Under these assumptions and using the terminology

$T_{ty}+T_{by}=q$ , Eq. (A.16) reduces to

$$\frac{\partial^2}{\partial x^2}\left(\int_{Sb(x)}^{St(x)}(y-A)\sigma_{xx}dy\right)+\frac{\partial^2 A}{\partial x^2}\left(\int_{Sb(x)}^{St(x)}\sigma_{xx}dy\right)+q=0 \qquad \text{(A.17)}$$

As in the thin-plate approach, we assume also that the origin of the vertical coordinate is in the centre of the layer

and use $St(x)=H/2$ and $Sb(x)=-H/2$ . Eq. (A.17) then becomes

$$\frac{\partial^2}{\partial x^2}\left(\int_{-H/2}^{H/2}y\sigma_{xx}dy\right)+\left(\int_{-H/2}^{H/2}\sigma_{xx}dy\right)\frac{\partial^2 A}{\partial x^2}+q=0 \qquad \text{(A.18)}$$

Eq. (A.18) has already the basic form of the thin-plate equation with the three terms representing the bending

moment due to flexure (left term), the moment due to compression (middle term) and the resistance of the

embedding medium (right term). To arrive at the thin-plate equation for viscous folding we have to make

assumptions about the stress distribution and the rheology. We assume that stresses are viscous and that the

horizontal total stress $\sigma_{xx}$ is composed of a constant layer-parallel stress due to a bulk shortening rate, $\bar{\sigma}_{xx}=4\eta\bar{D}_{xx}$ ,

and of a fibre (bending) stress $\tilde{\sigma}_{xx}$ , which is only related to the bending (flexure) of the layer but not to the

compression. The fibre stress depends on the flexural strain rate which can be approximated using Eq. (7) so that

$\tilde{\sigma}_{xx}=-4\eta y\partial^3 A/\partial t\partial x^2$ . The total horizontal stress can then be written as (Schmalholz et al., 2002)


$$\sigma_{xx}=\bar{\sigma}_{xx}+\tilde{\sigma}_{xx}=4\eta\bar{D}_{xx}-4\eta y\frac{\partial^3 A}{\partial t\partial x^2} \qquad \text{(A.19)}$$

The separation of the total stress into a stress due to a bulk shortening rate and a stress due to flexure is similar to the

separation of the stress into a basic state stress and a perturbed stress which is done in the stability analysis.

Substituting Eq. (A.19) into (A.18) and evaluating the integrals yields





$$-\frac{\eta H^3}{3}\frac{\partial^5 A}{\partial x^4 \partial t} + \bar{\sigma}_{xx} H \frac{\partial^2 A}{\partial x^2} + q = 0 \qquad (A.20)$$

The component of $\bar{\sigma}_{xx}$ vanishes by performing the integral in the bending moment (because the stress is multiplied

by $y$ in the left term in Eq. (A.18)) while the component of $\tilde{\sigma}_{xx}$ vanishes by performing the integral in the middle

term of Eq. (A.18) (because there the stress is not multiplied by $y$). Hence, the bending moment is only controlled by

flexural stresses while the moment due to compression is only controlled by the stress due to bulk shortening rate.

Eq. (A.20) corresponds to the thin-plate equation (4.8) in Biot (1961) and has been here derived from the general 2D

force balance equations (A.1) and (A.2). Using now $q = -4\eta k dA / dt$ (see Eq. (A.35) and text below) yields

$$-\frac{\eta H^3}{3}\frac{\partial^5 A}{\partial x^4 \partial t} + \bar{\sigma}_{xx} H \frac{\partial^2 A}{\partial x^2} - 4\eta_M k \frac{\partial A}{\partial t} = 0 \qquad (A.21)$$

Eq. (A.21) is identical to the equation (4.8) used in Biot (1961; he used the symbol $P$ instead of $\bar{\sigma}_{xx}$) to derive the

formula for the dominant wavelength. Alternative thin-plate equations for elastic material or for gravity as the

resisting mechanism against folding can be derived by using an elastic rheology and/or expressing $q$ by

gravitational stresses in Eq. (A.18).





**Appendix 2: Stream function approach and matrix resistance of viscous embedding medium**

An essential step in the derivation of the dominant wavelength solution was the derivation of a correct term

for the resistance of the viscous embedding medium, which depends not only on the amplitude, $A$, but also on the

wavelength, $L$ (see Eq. (8)). The derivation below follows essentially the one in Turcotte and Schubert (1982). The

constitutive equations (rheology) for linear viscous fluids are

$$\sigma_{xx} = -P + 2\eta \frac{\partial v_x}{\partial x}$$

$$\sigma_{yy} = -P + 2\eta \frac{\partial v_y}{\partial y} \tag{A.22}$$

$$\sigma_{xy} = \eta \left( \frac{\partial v_x}{\partial y} + \frac{\partial v_y}{\partial x} \right)$$

where $P = -\left( \sigma_{xx} + \sigma_{yy} \right) / 2$ (pressure or negative mean stress) and $v_x$ and $v_y$ are the velocities in the $x$- and $y$-

direction, respectively. Substituting Eqs. (A.22) into the 2D equilibrium equations (A.1) and (A.2), and assuming a

constant viscosity yields

$$\eta \left( \frac{\partial^2 v_x}{\partial x^2} + \frac{\partial^2 v_x}{\partial y^2} \right) - \frac{\partial P}{\partial x} = 0$$

$$\eta \left( \frac{\partial^2 v_y}{\partial x^2} + \frac{\partial^2 v_y}{\partial y^2} \right) - \frac{\partial P}{\partial y} = 0 \tag{A.23}$$

where we have used the relations from the incompressibility condition:

$$\frac{\partial v_x}{\partial x} = -\frac{\partial v_y}{\partial y}$$

$$\frac{\partial}{\partial y} \frac{\partial v_x}{\partial x} = -\frac{\partial^2 v_y}{\partial y^2} \tag{A.24}$$

$$-\frac{\partial^2 v_x}{\partial x^2} = \frac{\partial}{\partial x} \frac{\partial v_y}{\partial y}$$

The top of the above equations is the equation for the conservation of mass if density is constant. Taking the

derivative with respect to $y$ of the top equation in (A.23), taking the derivative with respect to $x$ of the bottom

equation in (A.23) and then subtracting both equations eliminates the pressure terms and yields





$$-\frac{\partial^3 v_y}{\partial x^3}+\frac{\partial^3 v_x}{\partial x^2 \partial y}-\frac{\partial^3 v_y}{\partial x \partial y^2}+\frac{\partial^3 v_x}{\partial y^3}=0 \tag{A.25}$$

The two unknown velocities can be represented by the derivatives of a so-called stream function

$$v_x=-\frac{\partial \varphi}{\partial y}$$
$$v_y=\frac{\partial \varphi}{\partial x} \tag{A.26}$$

Substituting equations (A.26) into (A.25) yields

$$\frac{\partial^4 \varphi}{\partial x^4}+2\frac{\partial^4 \varphi}{\partial x^2 \partial y^2}+\frac{\partial^4 \varphi}{\partial y^4}=0 \tag{A.27}$$

The above equation represents the force balance in a viscous medium with constant viscosity, and the stream function is a function of both $x$ and $y$. Usually one assumes a periodic behaviour in the direction along the layer, i.e. the $x$-direction, and writes the stream function as

$$\varphi(x,y)=\psi(y)\sin(kx) \tag{A.28}$$

Equation (A.27) then becomes

$$k^4 \psi \sin(kx)-2k^2 \frac{\partial \psi}{\partial y^2}\sin(kx)+\frac{\partial^4 \psi}{\partial y^4}\sin(kx)=0$$
$$\frac{\partial^4 \psi}{\partial y^4}-2k^2 \frac{\partial \psi}{\partial y^2}+k^4 \psi=0 \tag{A.29}$$

The general solution of equation (A.29) is

$$\psi=C_1 e^{-ky}+C_2 e^{-ky} y+C_3 e^{ky}+C_4 e^{ky} y \tag{A.30}$$

Seeking a solution for a half space where the velocities and stresses vanish at large distance $y \to \infty$ one has to set coefficients $C_3$ and $C_4$ to zero. A solution for the stream function is then

$$\varphi(x,y)=[C_1+C_2 y]e^{-ky}\sin(kx) \tag{A.31}$$





Calculating $v_x$ via Eq. (A.31) from Eq. (A.26) and assuming that $v_x = 0$ at $y = 0$ (which represents the interface

between layer boundary and embedding medium) provides $C_2 = kC_1$. The velocities are then using Eq. (A.31) and

(A.26)

$$v_x = C_1 k^2 \, e^{-ky} \, y \sin(kx)$$
$$v_y = C_1 (1 + ky) k \, e^{-ky} \cos(kx) \tag{A.32}$$

Substituting $v_x$ from Eq. (A.32) in the horizontal force balance (top of Eqs. (A.23)) and integrating with respect to x

to solve for the pressure yields

$$P = 2\eta C_1 k^2 \, e^{-ky} \cos(kx) \tag{A.33}$$

Evaluation of $v_y$ and $D_{yy} = dv_y / dy$ with Eq. (A.32) at the interface between matrix and layer (y = 0) yields

$$v_y (y = 0) = C_1 k \cos(kx)$$
$$D_{yy} (y = 0) = -C_1 k^3 \, e^{-ky} \cos(kx) \, y = 0 \tag{A.34}$$

The result $D_{yy}(y = 0) = 0$ indicates that there is no vertical deviatoric stress acting on the layer interface and hence

that $\sigma_{yy} = -P$. Incompressibility requires $D_{yy} = -D_{xx}$ and hence there is also no horizontal deviatoric stress acting on

the layer boundary. The resistance of the viscous medium corresponds hence to the resistance of a visocus fluid at

rest. The $v_y(y = 0)$ must be equal to the time derivative of the deflection of the layer, $v_y(y = 0) = dA/dt$. Hence,

$C_1 = (dA/dt)/k \cos(kx)$ and the pressure at $y = 0$ is

$$P(y = 0) = 2\eta k \frac{dA}{dt} = -\sigma_{yy} \tag{A.35}$$

The value of $P(y = 0) = -\sigma_{yy}$ is identical at the top and bottom layer boundaries if we assume that the material

above and below the layer are identical and that the deflection $A$ is identical, which is guaranteed if a constant

thickness of the layer is assumed. The vertical resistance of the matrix against folding therefore results from the

pressure at the layer boundary (Eq. (A.35)). The total resistance of the embedding medium above and below the

layer is then $q = 2P(y = 0) = 4\eta k dA/dt$ with $k = 2\pi/L$, using the convention, opposite to Biot (1961), that $q$ is

positive when acting against the positive direction of the deflection $A$.

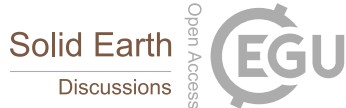

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





Table 1. Frequently used symbols and their consistent meaning throughout the text.

| Symbol | Meaning | Symbol | Meaning |
|---|---|---|---|
| $L_d$ | Dominant wavelength | $\alpha_d$ | Dimensionless maximal amplification rate (scaled by $\left|\bar{D}_{xx}\right|$) |
| $L$ | Wavelength | $\alpha$ | Dimensionless amplification rate (scaled by $\left|\bar{D}_{xx}\right|$) |
| $H$ | Layer thickness | $\dot{\varepsilon}, D_{xx}, D_{xy}$ | Strain rate and components |
| $H_M$ | Thickness of matrix | $\tau, \tau_{xx}, \tau_{yy}$ | Deviatoric stress and components |
| $H_{tot}$ | Total thickness of multilayer | $\sigma, \sigma_{xx}$ | Total stress and components |
| $A$ | Amplitude | $P$ | Pressure (mean stress) |
| $k$ | $2\pi / L$ | $F$ | Force per unit length |
| $s$ | $2\pi H / L$ | $t$ | Time |
| $\eta$ | Reference viscosity of layer | $\left|\bar{D}_{xx}\right|$ | Absolute value of basic state shortening/extension rate |
| $\eta_M$ | Reference viscosity of matrix | $D_{II}$ | Square root of second strain rate invariant |
| $R$ | $\eta / \eta_M$ | $\bar{D}_{xx}, \bar{\tau}_{xx}$ | Basic state variables |
| $n$ | Power law stress exponent of layer | $\tilde{D}_{xx}, \tilde{\tau}_{xx}$ | Perturbed variables |
| $n_M$ | Power law stress exponent of matrix | $\Delta\rho$ | Density difference |
| $g$ | Gravitational acceleration | $Ar_F, Ar_N$ | Argand number for folding and necking, respectively |





Table 2. Approximate solutions for dominant wavelength and maximal amplification rates for folding and necking.

| | Dominant wavelength | Maximal amplification rate (dimensionless) |
|---|---|---|
| **Single-layer folding** | | |
| **Power-law viscous folding** (embedded in infinite medium; linear viscous solution for $n, n_M = 1$) | $\dfrac{L_d}{H} = 2\pi \sqrt[3]{\dfrac{R}{6} \dfrac{n_M^{1/2}}{n}}$ | $\alpha_d = 1.21 \left( n\, n_M \right)^{1/3} R^{2/3}$ |
| **Large-scale folding** (Power-law viscous layer resting on inviscid medium) | $\dfrac{L_d}{H} = 2\pi \sqrt{\dfrac{2\eta \left| \overline{D}_{xx} \right|}{\Delta \rho\, g H}}$ | $\alpha_d = 6n \dfrac{2\eta \left| \overline{D}_{xx} \right|}{\Delta \rho\, g H}$ |
| **Detachment folding** (Power-law viscous layer resting on linear viscous layer with finite thickness $H_M$) | $\dfrac{L_d}{H} = 1.2\pi \left( \dfrac{R}{3n} \right)^{\frac{1}{6}} \sqrt{\dfrac{H}{H_M}}$ | $\alpha_d = 3n \left( \dfrac{R}{3n} \right)^{\frac{1}{3}} \dfrac{H_M}{H}$ |
| **Finite length power-law viscous folding** ($\eta_e = \eta \left( 1 + \left[ 2\eta / \eta_M a \right]^{n-1} \right)$, $D_a = \eta_e / \eta_M a$ and finite length solution valid for $D_a \gg 1$) | $\dfrac{L_d}{H} = 2\pi \left( \dfrac{1}{6n} \dfrac{\eta_e}{\eta_M} \right)^{\frac{1}{3}}$ | $\alpha_d = \dfrac{1}{1 + 2D_a} \left( \dfrac{4n}{3} \dfrac{\eta_e}{\eta_M} \right)^{2/3}$ |
| **Multilayer folding** | | |
| **Few linear viscous layers** (Multilayer embedded in viscous medium for $m \ll 2\pi \left( R/6 \right)^{1/3}$; $m$ is number of layers) | $\dfrac{L_d}{H} = 2\pi \left( m \dfrac{R}{6} \right)^{\frac{1}{3}}$ | $\alpha_d = 1.21 \left( mR \right)^{\frac{2}{3}}$ |
| **Many linear viscous layers** (Multilayer embedded in viscous medium for $m \gg 2\pi \left( R/6 \right)^{1/3}$; $m$ is number of layers) | $\dfrac{L_d}{H} = 2\pi \left( m \dfrac{R}{6} \right)^{\frac{1}{3}}$ | $\alpha_d = R$ |
| **Internal linear viscous folding** (Confined multilayer with weak layers of significant thickness. $a = H_M / \left( H_M + H \right)$; $m$ is half number of layers) | $\dfrac{L_d}{\sqrt{H\, H_{tot}}} = 1.9 \left( 1 + 3.63 \dfrac{\eta}{\eta_M} \dfrac{a^2}{m} \right)^{1/6}$ | |
| **Internal linear viscous folding** (Confined multilayer with weak layers of insignificant thickness. $H_M \to 0$) | $\dfrac{L_d}{\sqrt{H\, H_{tot}}} = 1.9$ | |
| **Single-layer necking** | | |
| **Power-law viscous necking** (embedded in infinite medium; linear viscous solution for $n, n_M = 1$) | $\dfrac{L_d}{H} = 2\pi \sqrt[3]{\dfrac{R}{6} \dfrac{n_M^{1/2}}{n}}$ | $\alpha_d = n - 1$ |
| **Large-scale necking** (Power-law layer resting on viscous medium with exponentially decaying viscosity for typical crustal rheologies and densities) | $\dfrac{L_d}{H} \approx 3.7 \pm 0.3$ | |





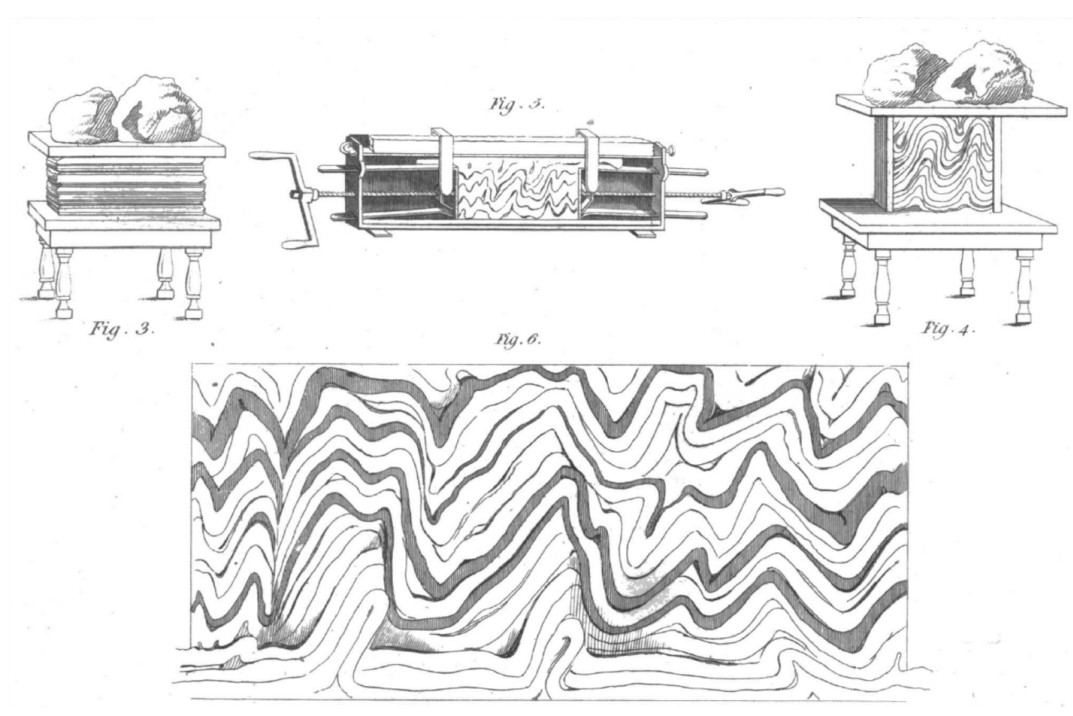


Figure 1. Original sketch of James Hall's (1815) folding experiment.

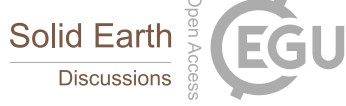

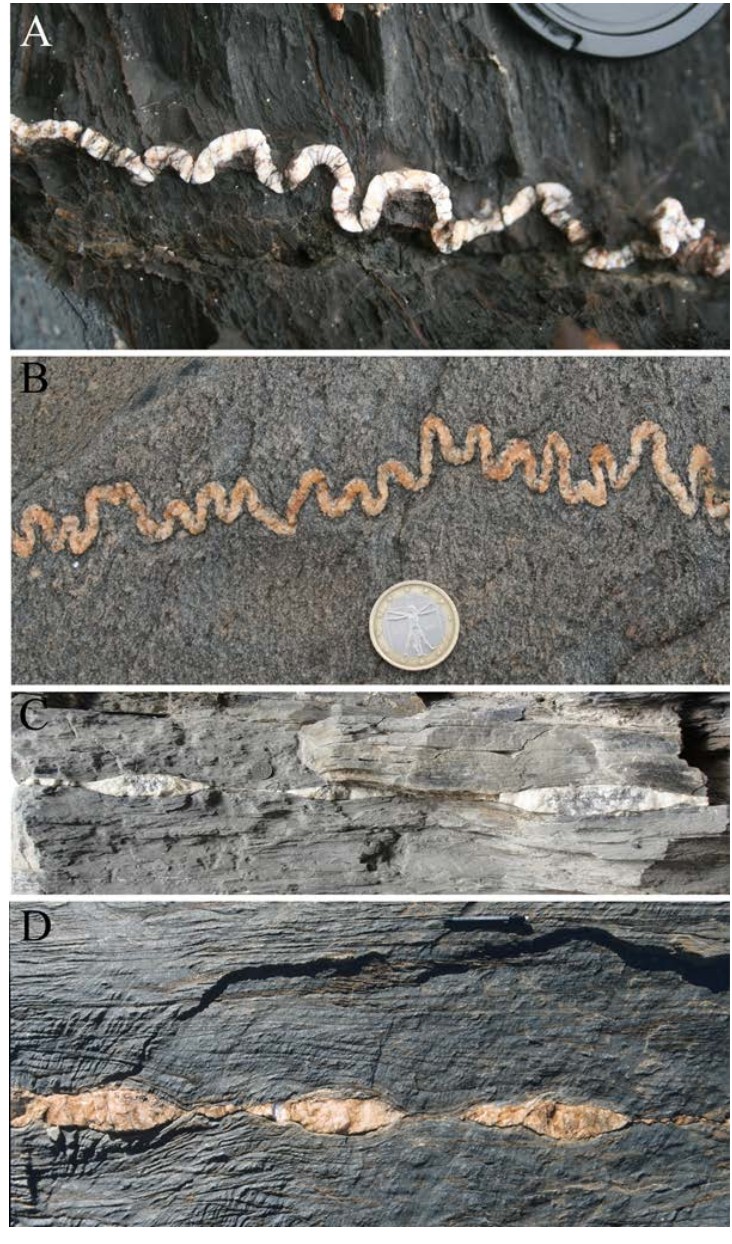

Figure 2. Natural single-layer folds (A and B) and pinch-and-swell structure (C and D). A) Folded quartz vein

around Val Figueiras, Portugal. B) Folded quartz vein from Cap de Creus, Spain. C) Extended calcite vein in finer

grained calcite marble from the Doldenhorn nappe, Switzerland. D) Extended quartz vein in grey calcite marble,

Ugab region, northern Namibia.




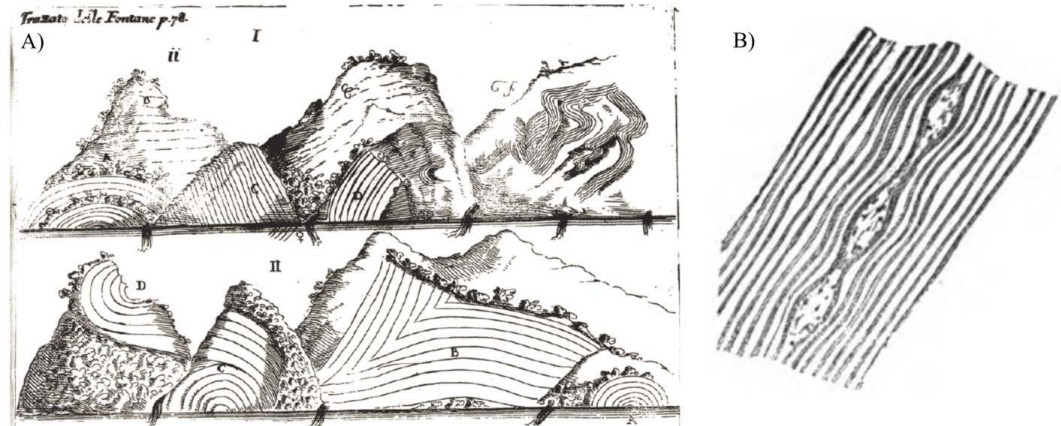

Figure 3. A) Fold observations in the European Alps around the Lake Uri in Switzerland. The panel shows a part of

a larger sketch of Johann Scheuchzer, which was published by Antonio Vallisneri (1715; see Luzzini, 2011; Vaccari,

2004). B). Sketch of necking (pinch-and-swell structure) from Ramsay (1866).




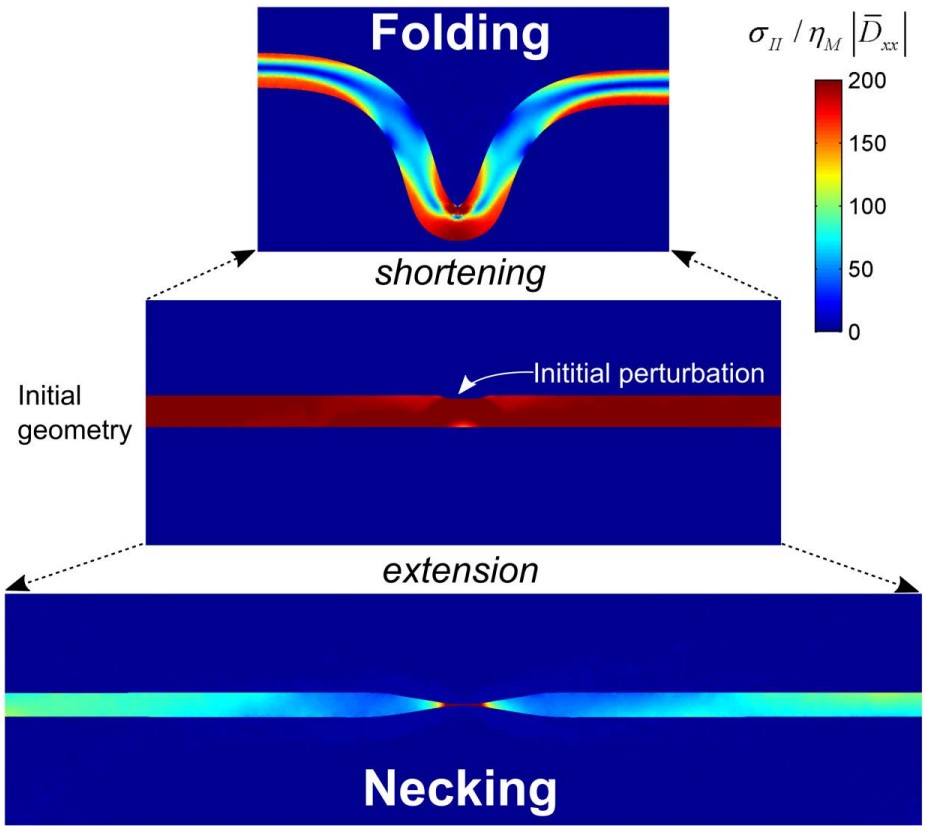

Figure 4. Numerical results of folding and necking simulations resulting from layer-parallel shortening and extension, respectively, of the same competent layer with an initial lateral thickness variation (initial perturbation). The geometries are the result of finite element simulations with a reference viscosity ratio of 100, a power-law exponent of the layer of 10 and of the embedding medium of 3. The colors indicate the square root of the second invariant of the stress tensor ($\sigma_{II} = \left( \tau_{xx}^2 + \tau_{xy}^2 \right)^{1/2}$) which is non-dimensionalized by dividing it by the product of matrix reference viscosity to bulk deformation rate ($\eta_M \left| \bar{D}_{xx} \right|$). For the initial geometry the value of $\sigma_{II}$ in the layer is close to the absolute value of the basic state deviatoric stress $\bar{\tau}_{xx} = 2\eta_{ref} \bar{D}_{xx}$. With progressive folding and necking the average stress of the layer-matrix system decreases. For necking, high stresses are localized in the neck. The corresponding structural softening of the folding and necking simulations is shown in Figure 18.



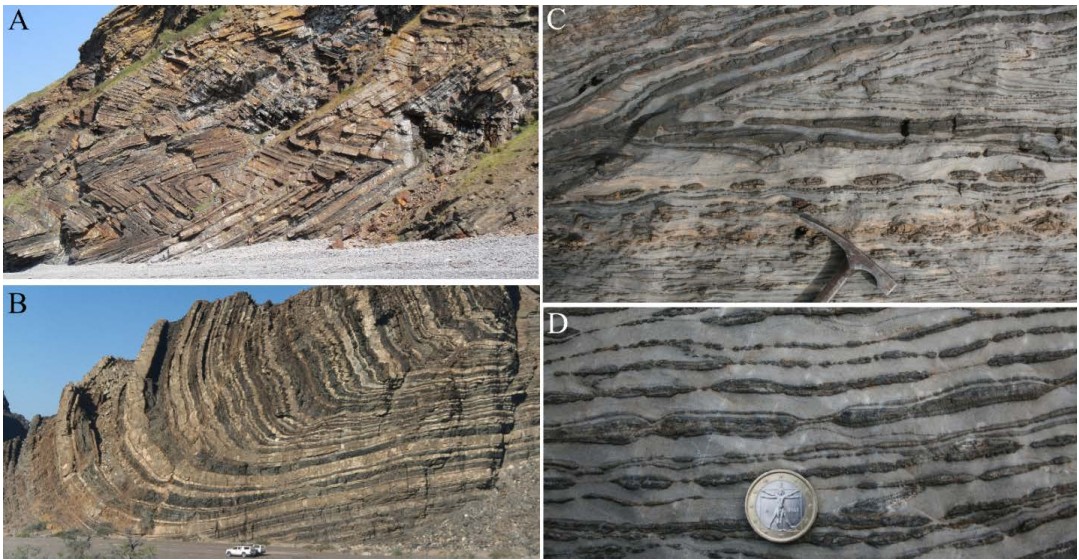

Figure 5. Multilayer folds (A, B), multilayer folds with boudins (C) and multilayer pinch-and-swell with boudins (D). A) Folded carboniferous sandstones and shales at Millook Haven, Cornwall, England. B) Multilayer folds in alternating turbiditic sandstone and shale layers, lower Rhino Wash, Ugab region, northern Namibia. C) Folded and extended layers of calc-silicate in marble, Monte Frerone, Adamello region, Italy. D) Extended multilayer of calc-silicate in marble around Monte Frerone, Adamello region, Italy.




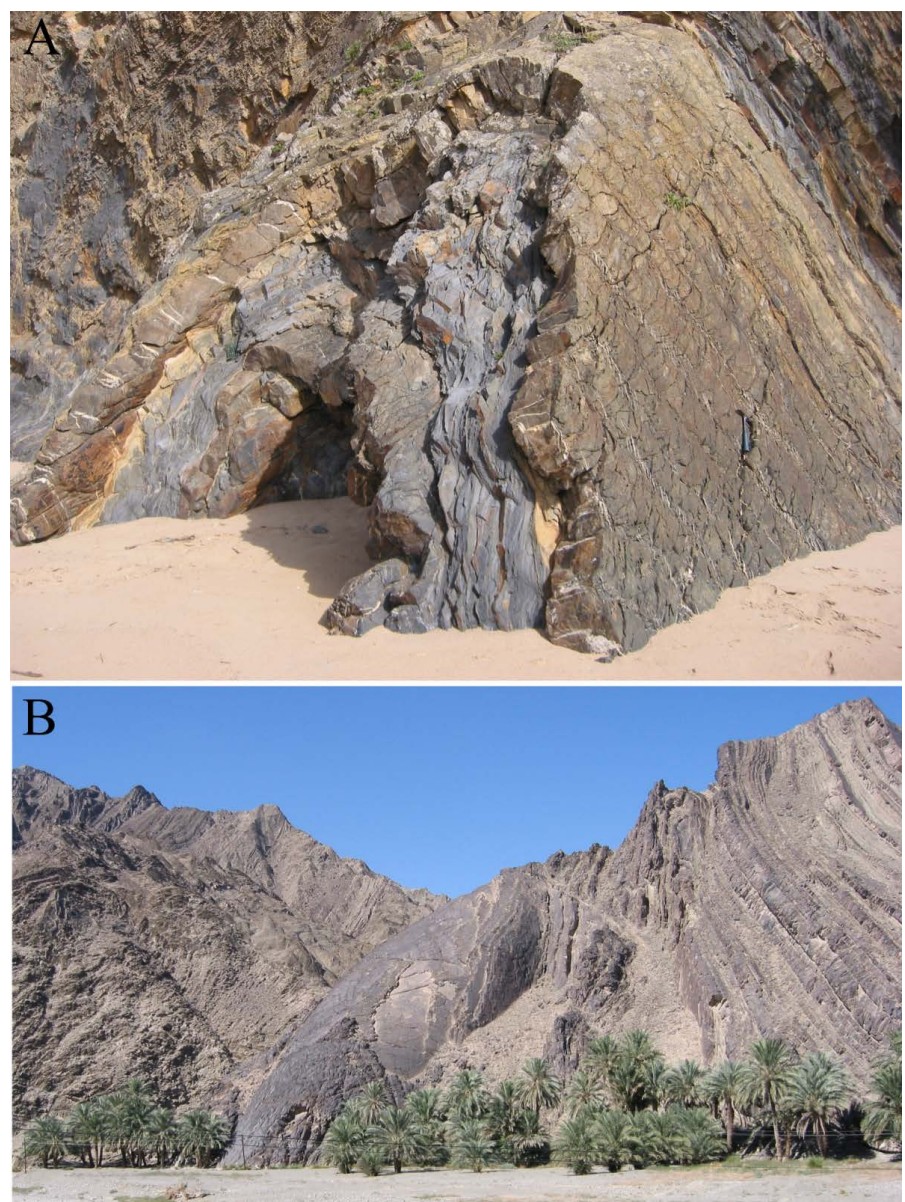

Figure 6. Outcrops showing the three-dimensional geometry of folds. A) Folded turbiditic sequence from

Almograve, Portugal. B) Folded turbiditic sequences with a large fold hinge plunging towards the viewer from

Makran region, Iran.



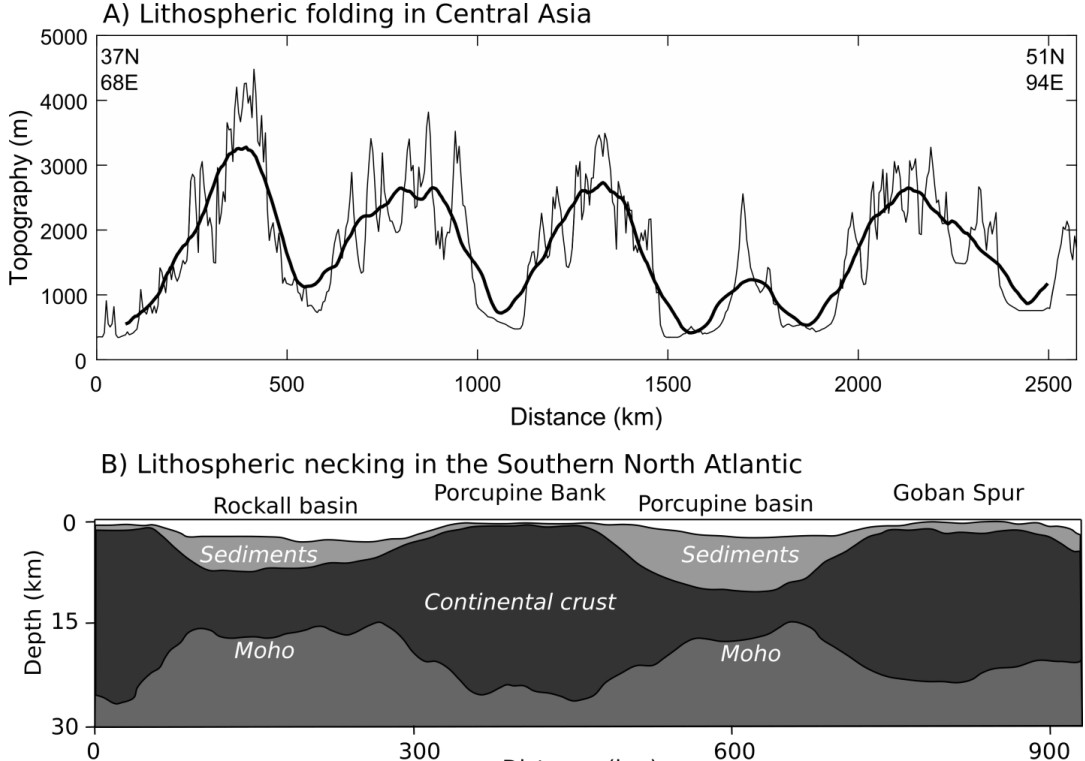

Figure 7. Lithospheric folding and necking. A) Topography across Central Asia with data from Geocontext-Profiler. The thin solid line is the original data and the thick solid line shows the running average topography within a 150 km wide window. The fold-like topography has been interpreted as the result of lithospheric folding (Burov et al., 1993). B) Crustal geometry across the Rockall and Porcupine basins modified after Mohn et al. (2014) and Welford et al. (2012). The thinned continental crust has been interpreted as the result of necking.






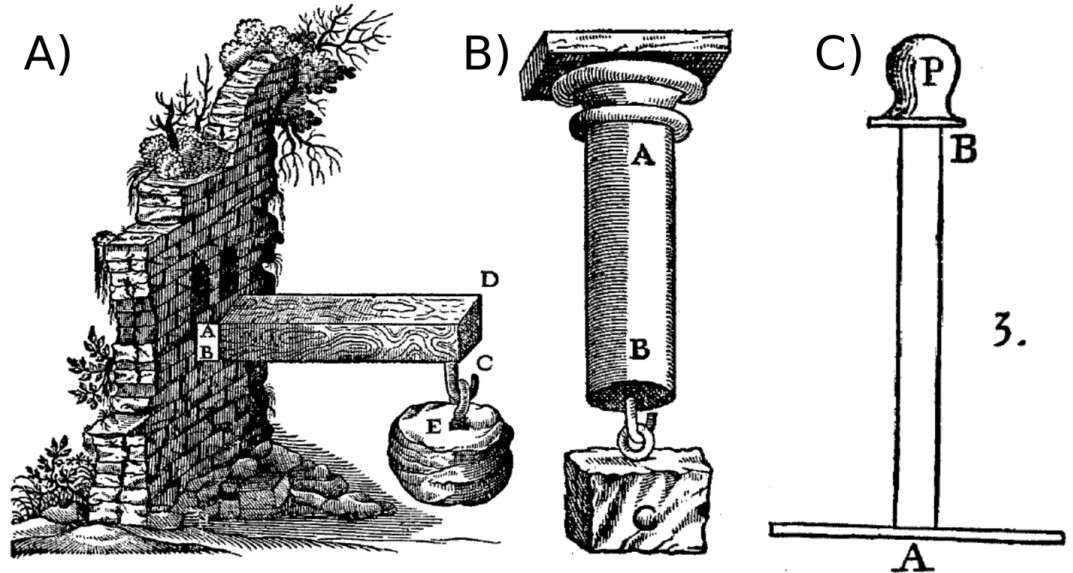

Figure 8. A) and B) shows sketches of Galilei (1638) who studied the strength of beams under loading (A) and the

tensile strength of columns (B). C) shows a sketch of Euler (1744) who studied the so-called elastic curves (elastica)

and the critical load for buckling of loaded columns.





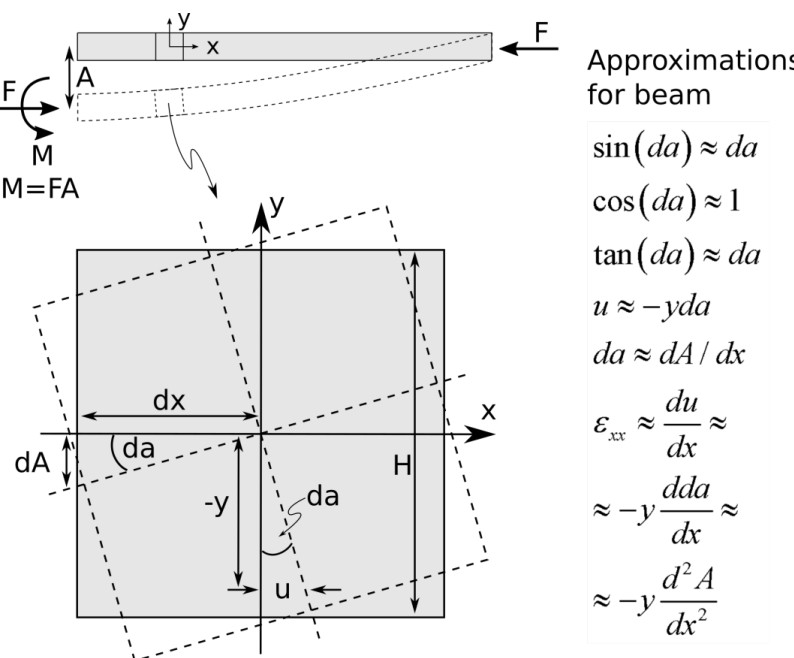

Figure 9. Sketch illustrating the Euler-Bernoulli beam theory and the related thin-plate approximations.



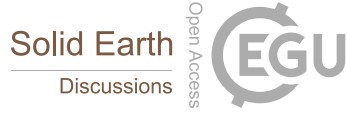

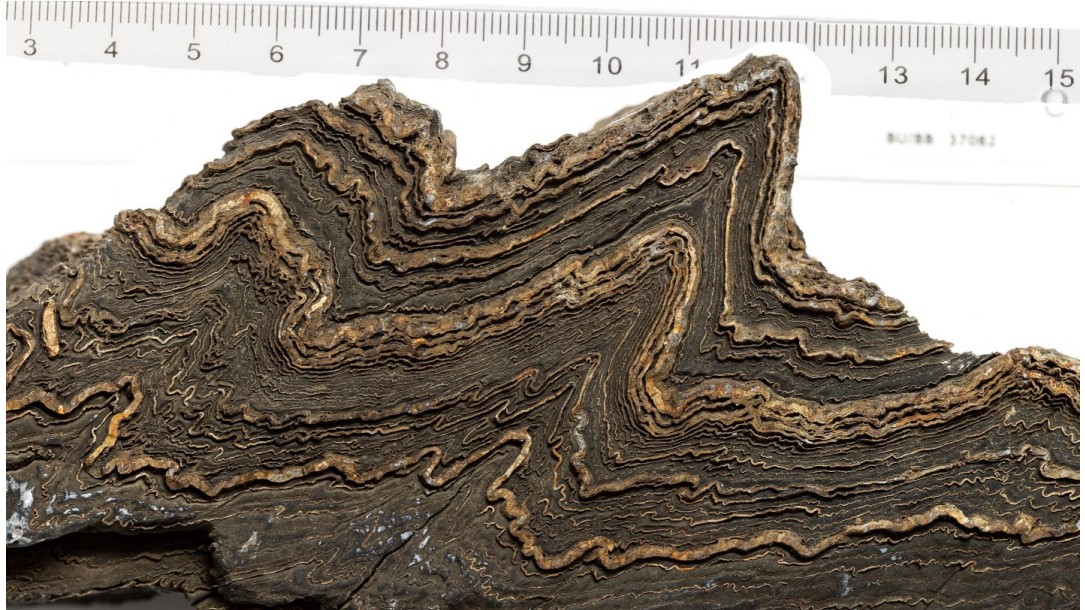

Figure 10. Multilayer folds showing that the size of individual folds (quantified by amplitude and distance between hinges) is related to their layer thickness and that folds become systematically smaller as their layer thickness

becomes thinner. Carbonates with silicate-rich layers belonging to the Jurassic El Quemado formation. The sample was found by Stéphane Leresche around the Mount Fitz Roy, Southern Patagonia, and the photo was made by Yoann Jaquet.




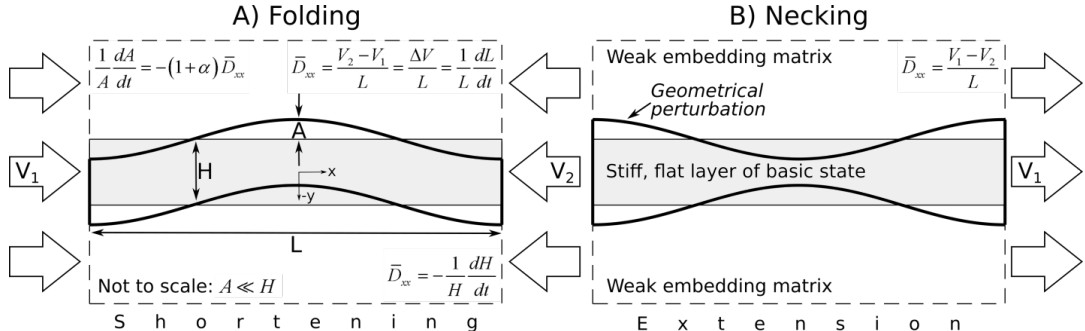

Figure 11. Configuration for analytical folding (A) and necking (B) models with some basic equations. $A$ is

amplitude, $H$ is layer thickness, $L$ is wavelength, $V_{1,2}$ are horizontal boundary velocities, $t$ is time, $\alpha$ is

amplification rate and $\bar{D}_{xx}$ is the applied bulk rate of deformation.





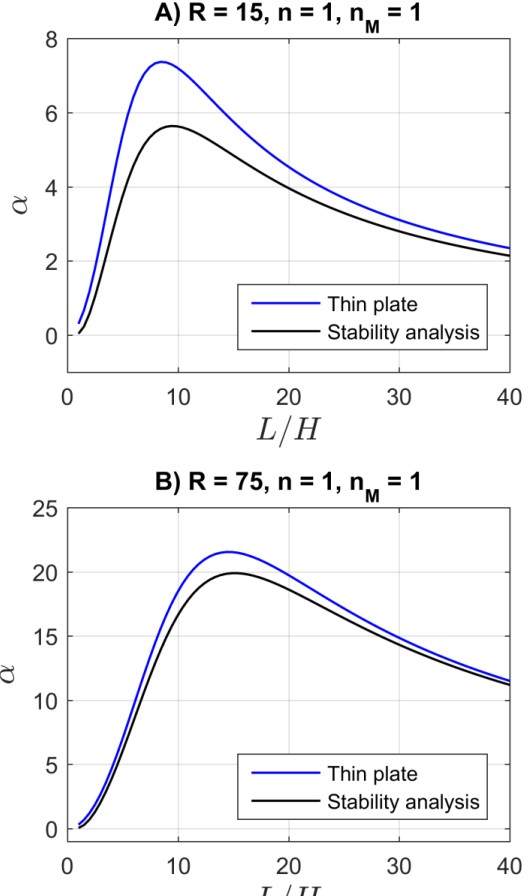

Figure 12. Dimensionless amplification rate, $\alpha$, versus wavelength to thickness ratio, $L/H$, for linear viscous

layer and embedding medium and a viscosity ratio $R = 15$ (A) and $R = 75$ (B). Approximate results are based on the

thin-plate approach and exact results (for infinitesimal amplitudes) are based on the stability analysis.



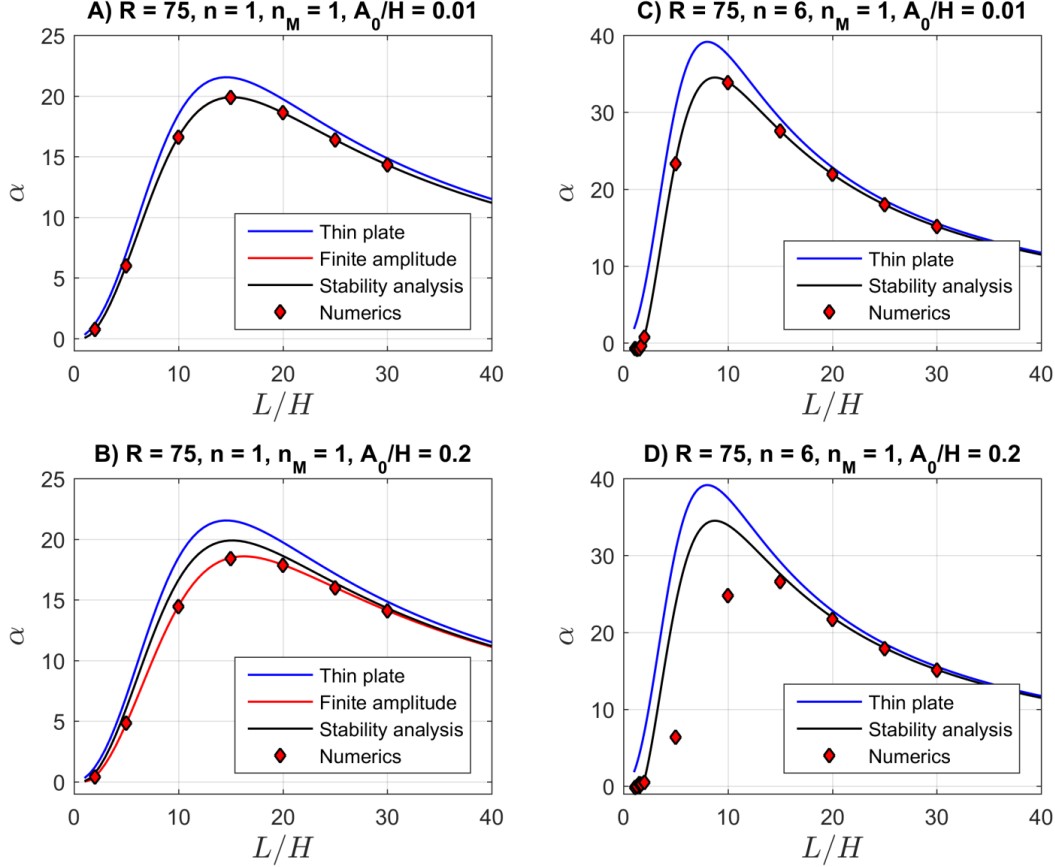


Figure 13. Dimensionless amplification rate, $\alpha$, versus wavelength to thickness ratio, $L/H$. All results have been calculated with the software Folder (Adamuszek et al., 2016) including the numerical results of finite element simulations indicated with red diamond symbols. A) Layer and matrix are linear viscous (power-law exponent of layer and matrix $n$, $n_M = 1$), the viscosity ratio $R = 75$ and the initial ratio of amplitude to layer thickness $A_0/H = 0.01$. B) Like A) but with $A_0/H = 0.02$. For linear viscous folding the finite amplitude solution of Adamuszek et al. (2013b) can be used to calculate the amplification rate for finite amplitudes. C) Layer is power-law viscous and matrix is linear viscous ($n = 6$ and $n_M = 1$), the viscosity ratio $R = 75$ and the initial ratio of amplitude to layer thickness $A_0/H = 0.01$. D) Like C) but with $A_0/H = 0.02$.




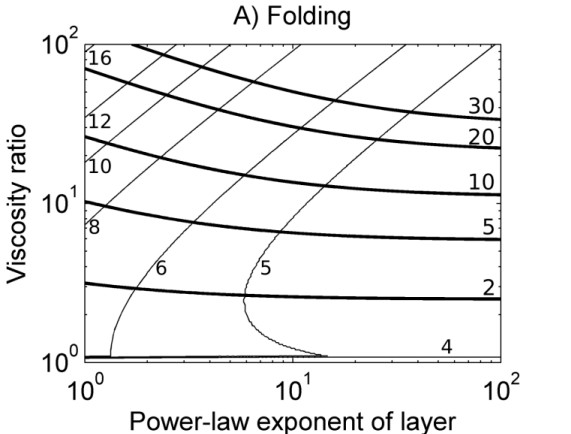
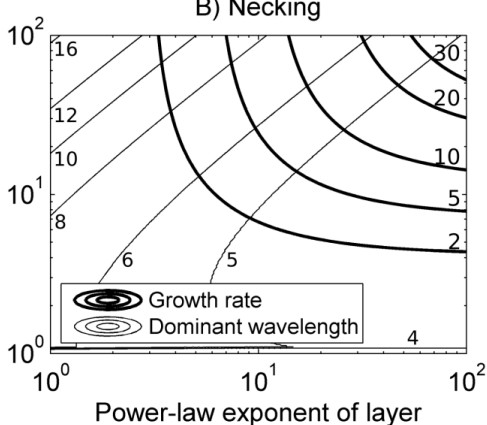

Figure 14. Ratio of dominant wavelength to layer thickness (thinner lines) and dimensionless amplification rate

(thicker lines) for folding (A) and necking (B) as function of the reference viscosity ratio and the power-law stress

exponent of the layer. The embedding medium is linear viscous. For folding the amplification rates are significant

also for small values of the power-law exponent (< 5) while for necking the amplification rates are insignificant (<

10) for power-law exponents < 10.




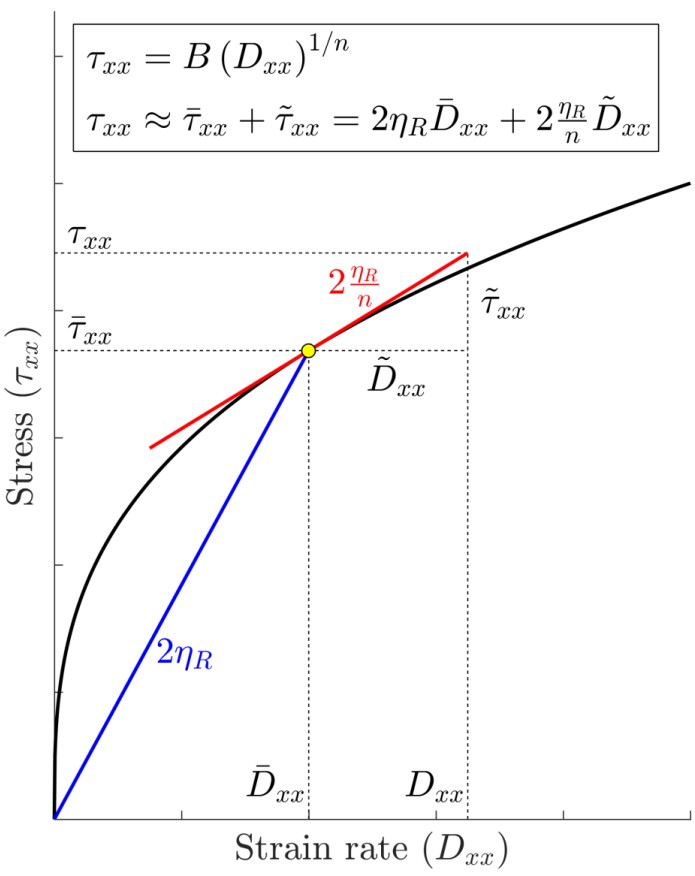

Figure 15. Sketch illustrating the linearization of power-law flow law by using basic state variables ($\bar{\tau}_{xx}$, $\bar{D}_{xx}$) and

perturbed variables ($\tilde{\tau}_{xx}$, $\tilde{D}_{xx}$). The reference viscosity for the basic state deformation, $\eta_R$, is related to the secant at

the point $\left(\bar{D}_{xx}, \bar{\tau}_{xx}\right)$ while the viscosity for the perturbed deformation is related to the tangent at the point $\left(\bar{D}_{xx}, \bar{\tau}_{xx}\right)$.

The slope of the tangent is a factor $n$ smaller than the slope of the secant.



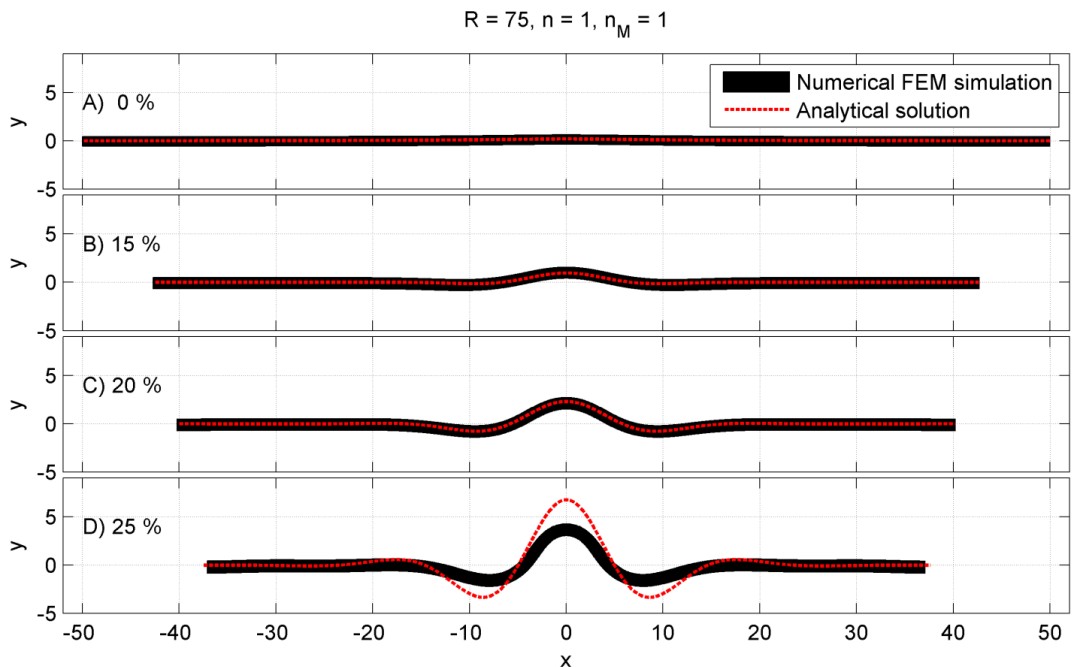


Figure 16. Evolution of fold geometry for an initial geometrical bell-shaped perturbation calculated with an

analytical solution (Eq. (27)) and a numerical finite element simulation. The parameters for the bell-shaped

perturbation are $a = 10$ and $b = 0.2$ (initial layer thickness is 1), and the linear viscosity ratio is 75.





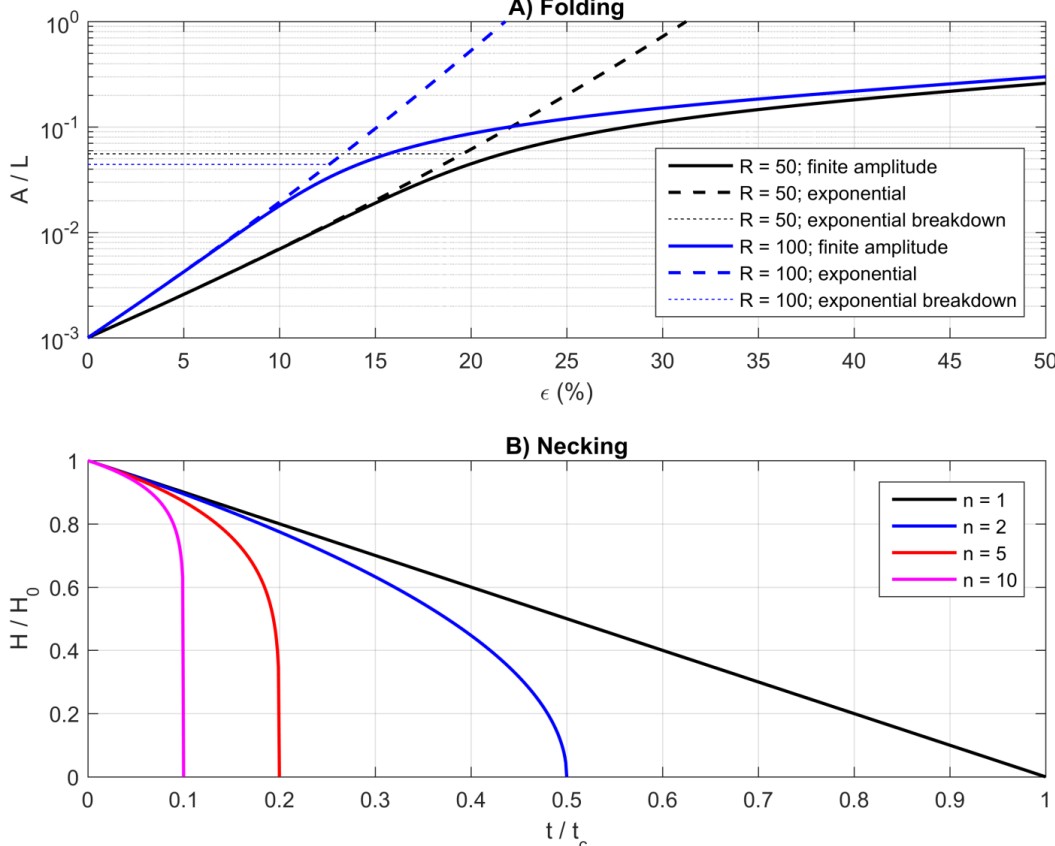

Figure 17. Approximate finite amplitude solutions for folding (A; Eq. (29)) and necking (B; Eq. (41)). For folding the layer and embedding medium are linear viscous, while for necking the layer is power-law viscous and not embedded in a viscous medium (free layer or embedded in inviscid medium). A) Ratio of amplitude to wavelength ($A / L$) versus horizontal shortening ($\varepsilon$) for a viscosity ratio ($R$) of 50 and 100. Both the exponential and finite amplitude solution are plotted. The thin dashed linas indicate the breakdown of the exponential solution as quantified by Eq. (30). B) Ratio of layer thickness to initial layer thickness ($H / H_0$) versus the dimensionless time ($t / t_c$; see text below Eq. (41)) for different values of the power-law exponent ($n$). For $n = 1$ no necking instability is active and thinning is due to homogeneous pure shear thinning only. The time for complete thinning ($H / H_0 = 0$) is given by $t / t_c = 1/n$ (see text).




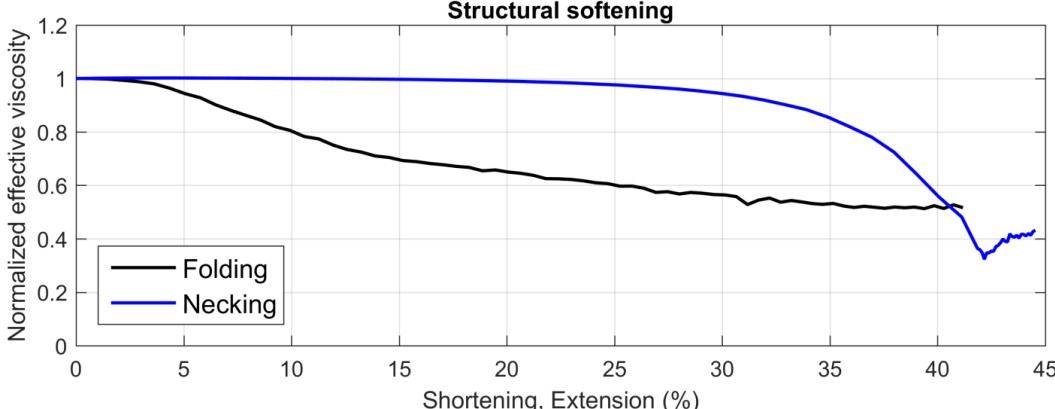

Figure 18. Structural softening during folding (A) and necking (B) of the simulations shown in Figure 4. The effective viscosity is calculated by the ratio of the area-averaged square root of the second invariant of the deviatoric stress tensor, $\sigma_{II}$, to the absolute value of the bulk rate of deformation, $\bar{D}_{xx}$, which was constant during the simulations. The effective viscosities are divided by the initial value of the effective viscosity and are plotted versus the bulk shortening for folding and the bulk extension for necking. Structural softening starts earlier for folding than for necking but the structural softening for necking is more intense.