# Peer review of "Folding and necking across the scales: a review of theoretical"

_Solid Earth, 2016_

## Referee Comment (RC1) · P. Hudleston (Referee) · 6 Jul 2016

P. Hudleston (Referee)

hudle001@umn.edu

This is an outstanding, well-written review of the large body of work carried out on folding and necking/pinch-and-swell, focusing on theoretical and experimental perspectives. The period covered extends back beyond the time when geology first became a discipline in its own right, with references to early work on the strength of beams and columns in the 17th and 18th centuries. Links to relevant engineering and mechanics literature are made throughout. The abstract and introduction by themselves make an excellent mini review of the topic.

The basic physics and mathematics involved in folding and necking are clearly explained, in the text and in the appendices, and the development of ideas nicely traced.

[Figure]

The similarities and differences between the response of layered materials to shortening and extension - by folding and necking respectively – are clearly documented.

The paper is long and there are a few instances of repetitions that could be reduced or eliminated. For instance, Neurath and Smith (1982) is cited in several places in the text and their work is introduced twice, first on p. 815 and then on p. 968. There could be some modest reduction in text by changing the second of these to reflect the earlier introduction.

The appropriateness of the choice of necking rather than pinch-and-swell to use as the general term to describe the phenomenon in extension might be debated, since necking generally implies (to me at least) a single point of thinning, whereas folding encompasses a range of behaviors from a single structure to a fold train. Pinch-and-swell of course implies repetition along the layer. The range of behaviors from single necking to "periodic" pinch-and-swell is covered in this paper.

It might be useful, since this is a review, to define "competent" when the term is first used (line 34), as some readers may be unfamiliar with this usage. And while considering use of terms, it might be worth pointing out somewhere in the text, as was done by Smith, that the non-linear rheology that leads to necking and pinch-and-swell need not be power law.

There are a few issues with the misspelling of proper nouns - Rockhall (line 1067), Dieteric (line 1700), and the MS should be checked for these. Also, it should be checked that all citation's in the text appear in the reference list and vice versa. Examples that (perhaps not surprisingly) I noticed are Hudleston (1973) that is cited in the text but not in the reference list and Hudleston and Stephansson (1972) that is in the reference list but not cited in the text (and in any case should be 1973).

It would be helpful to the reader if references to monographs give page numbers where the citations appear in the text.

The illustrations are excellent, but it would be nice to add one at least that illustrates in some way the dependence of dominant wavelength on spacing of the competent layers .

Specific Comments

line 402. It might be worth pointing out here that the amount of bulk shortening that is undergone before the analytical solution significantly overestimates amplification depends on both the viscosity ratio and the initial amplitude of the perturbation. This of course is discussed elsewhere in the text.

lines 412. I believe Johnson and Fletcher (1994) first used the expression "preferred wavelength. "Sherwin and Chapple do not use the term, but rather consider the dominant wavenumber a function of shortening. Fletcher and Sherwin also do not use the term preferred wavelength, but refer throughout their paper to the "wavelength that has received maximum amplification."

line 569. Reference here to Paterson and Weiss (1966) for development of chevron folds from kinks as described in first part of (iv)?

line 627. Perhaps a reference here to authors who considered folding of the crust or lithosphere to be impossible (Ramberg is one, but I don't recall the reference)..

lines 810-815. This phenomenon is also nicely illustrated in Ramsay and Huber (1987), as reflected by cleavage development in the competent layer.

line 1000. Although there is no layer-parallel shear in the competent layer during necking, there must be in the adjacent matrix.

line 1004. The decrease in amplification rate with increasing extension for necking must be for a different reason than the decrease in this rate for folding – for necking the rate presumably decreases because the wavelength is increasing and progressively further away from the dominant wavelength. For folding the reason is as explained by Schmalholz and Podladchikov (2000).

line 1273. A good natural example of this phenomenon is described (so far only) in a GSA abstract by myself and others (Hudleston, P., McEvoy, M.E., Watkins, W.D., Porter, M., 2015. Rheological information inferred from small-scale folds, Northern Snake Range, Nevada. Geol. Soc. Abstr. with Prog., 47 (7), p. 720.

Fig. 2. The lithology of the matrix in which the quartz veins in parts A and B are embedded should be given.

Peter Hudleston

---

## Short Comment (SC1) · 11 Jul 2016

Comments on the manuscript:

**Folding and necking across the scales: a review of theoretical and experimental results and their applications**

by S. M. Schmalholz and N. Mancktelow

submitted to Solid Earth (se-2016-80)

**Comment Regarding LAF**

The initial stages of fold development in the infinitesimal amplitude limit are well described by thin plate (if the viscosity ratio is high enough) and thick plate stability analysis (which is the exact solution). Both predict exponential growth of the amplitude with strain according to the respective growth rate spectra, which do not evolve in time. This model does not do a very good job in predicting large amplitude fold geometries with strain. There are two main processes that modify the exponential growth of single components.

1) As the layer folds it 'escapes' the applied far-field shortening, i.e. the actual layer (arc length) shortening rate is less than the shortening of the entire system. This leads to a slowdown of fold amplification as well as a structural softening of the system. This process was studied by Schmalholz and co-workers in a number of papers and is referred to as the finite amplitude solution (FAS).

2) The growth rate is a function of the wavelength to thickness ratio. The layer thickness increases due to the progressive shortening (at least as long as the individual limbs have not rotated too much) and wavelength decreases. This means that the growth rate of a given component (initial wavelength to thickness ratio) must change if its wavelength and thickness evolve. The component that has initially the highest amplification rate will in the next moment not be the fastest component any longer. Instead a component with an initially larger wavelength to thickness ratio will have the largest integrated amplification. This process is referred to as preferred wavelength development and was originally discussed in Sherwin and Chapple and then by Fletcher and Sherwin.

In LAF, we combine the thick plate stability analysis with both of these processes. We introduce various improvements and obtain a conceptually simple model that is capable of predicting fold development up to large amplitudes. The importance of LAF does not so much lie in the details of thick versus thin plate or how the elliptical integrals of the arc length evolution are approximated – its importance is that it combines all the key processes.

**Line 453-onwards**

A rather unfortunate way to introduce LAF. It is discredited before it is properly introduced in the next paragraph. We disagree with the statement that LAF developed for multiple waveforms does not represent a great improvement compared to the single waveform solution. Below we show a modified version of Fig. 16 illustrating the results of the fold shape evolution for the numerical (FEM, grey fold), LAF for single (blue), and LAF for multiple (red) wavelength solution.

LAF derived for single waveforms should not be used in multiple waveform cases. After 25% of shortening, LAF for multiple waveform and single waveform differ significantly. At 25% shortening the multiple waveform solution still fits the FEM model well. The limb dip at this stage is ~ 45°. In our opinion this is a great improvement compared to the superimposed single sinusoidal solutions.

[Figure]

**Fig. 1 Evolution of the fold geometry perturbed with a bell shape function shown for numerical results (grey layer) and derived analytically using LAF for single sinusoidal waveforms (blue) and multiple waveform (red) solutions. Top four figures show the evolution from 0 to 25% shortening. The bottom figure is a zoomed version of the final stage.**

**Detailed Comments**

**Line 410**: it should be $ds/dt = -2\bar{D}_{xx}s$. s0 is incorrect in this context.

**Line 415 - onwards**: This paragraph is misleading. Here, a new equation should be presented that introduces the correction to Eq. 16. Otherwise the correction $\bar{D}_{xx}$ applies to both passive and dynamic amplification. Why not to show the expression for the correction?
When the authors refer to finite amplitude solution they should specify which one (i.e. which version of FAS, LAF, ?). In Schmalholz (2006), there are two expressions for the growth rate correction provided, i.e. Eq. 18 and 19. The second expression is a slight modification of Eq. 18 that is claimed to only differ by 10%. However, the difference in the growth rate spectrum is large. We show this with a modified version of Fig. 13b (from the paper submitted by Schmalholz and Mancktelow), where different analytical solutions are provided for a linear viscous material and large initial amplitude $A_0/H=0.2$ (Fig. 2A). This large initial amplitude is the reason for the mismatch between thick plate and numerical (FEM) growth rates. The two versions of FAS correct the thin plate prediction, on which they are based, towards lower growth rates. One correction (Eq. 19, green line) essentially matches the thick plate solution while the other one (Eq. 18, blue line) yields lower values. Only LAF produces a perfect match of the numerically computed growth rate spectrum.

[Figure]

**Fig. 2 Growth rate solutions for different theoretical models (refers to the Fig. 13b). We use in A) thin plate and in B) thick plate solution in the FAS model.**

The question arises how FAS would perform if it would be based on thick plate. This is shown in Fig. 2b. The green dashed curve (FAS, Eq. 19) essentially coincides with the numerical and the LAF results. The remaining minor differences are due to the different ways of evaluating the elliptical integral of the arc length evolution. In LAF we solve the actual set of equations while in

FAS several (cascading) simplifications are made. Which approach is better is a moot point. Elliptic integrals are special but such are sine and other functions.

Fig. 2 shows that both FAS and LAF do a good job predicting the growth rate for this static case of relatively large initial amplitude. But only LAF considers how the wavelength and thickness of individual components evolve with strain and therefore does a better job reproducing the finite amplitude evolution.

**Line 440**: The exact solution has been presented by Adamuszek et al. 2013.

**Line 504**: Shouldn't be "-" here? Writing equal sign here can be also misleading. Write (-$D_{xx}t \approx 0.2$)

**Line 1540**: What is k? For consistency, it would be better to use s.

---

## Referee Comment (RC2) · B.J.P. Kaus (Referee) · 26 Jul 2016

This is a very extensive and nice review on folding and necking instabilities, which brings the reader up-to-speed with the available literature on the topic and I appreciate the effort that the authors have put into it. I believe it is well suited for publication in SE; yet I feel that a few topics could and should be discussed a bit more extensively, as it will make the review more complete and as I suspect that we will have to wait quite a few years for the next review on this topic.

1) Effect of brittle/plastic or Mohr-Coulomb rheology This review mainly focusses on the mechanics of elastic or (nonlinear) viscous folding & necking instabilities. If the number of citations is a measure of the scientific interest in a topic, a quick (and unrepresenta-

tive) google scholar seems to at least suggests that the papers that get most citations deal with lithospheric-scale folding theories. As you already state at several places in the introduction, the upper crust (and potentially the mantle lithosphere) predominantly deforms in a brittle manner, rather than viscous or elastic. Most in-situ stress measurements suggests that Byerlees law represent the state of the stress of the crust pretty well and that the (upper) crust is close to failure, which can be reasonably well mimicked by Mohr-Coulomb plasticity. Most papers (from the 70ie & 80ies) that deal with lithospheric deformation take this into account by approximating the brittle layers as powerlaw layers with a high powerlaw exponent (in some cases with a depth dependent pre-factor).

Yet, given the linearization involved, it is not all that clear to which extend those results are actually correct in the nonlinear regime. In my experience, the analytical models are inconsistent with those obtained in numerical models in which the overburden has a pressure-dependent rheology (which you can also see by inserting n=infinity in your equation for powerlaw detachment folding in table 2, which results in a dominant wavelength of zero). In Yamato et al. (2011, geology) we looked at this and found that numerically computed growthrate diagrams are capable of predicting the folding/faulting boundary (and the wavelength developing in finite strain numerical models). The growthrate diagrams are quite similar to the growthrate diagrams you discuss here for viscous folding (with a single maximum). Yet, it remains unclear how to best reproduce this with purely mechanical/analytical models. There are a number of papers on this subject (e.g, Johnson, 1980 Tectonophysics; Erickson 1996 JSG; Simpson 2009 JSG), and you do discuss some of the experimental studies. Yet, a separate and more extensive discussion of this topic would be very helpful.

2) Crustal-scale folds Given that the title of your manuscript has "across the scales", I feel you move from outcrop-scale folds to lithospheric-scale folding a bit too quickly, while missing many of the spectacular and well documented 3D examples of crustal scale folding (such as in the Zagros). I believe there has been quite a bit of progress in

recent years in understanding why there are folds (and not faults) in some areas, how this is related to the crustal scale structure and geometry, and how the 3D evolution of folds fits with constraints from uplift and river network deflection. I believe this fits well to the topic of your review and it would be great if you can include both figures of crustal-scale folds & a discussion on (mechanical) research on this.

3) LAS Dani Schmid & coworkers have already pointed it out in their review that the large amplitude solution should and can be discussed more completely here. To me, the main breakthrough in folding literature over the last say 15 years or so is that we can now model (ductile) folding instabilities with approximate analytical solutions in 2D and 3D up to large amplitudes. The importance of that cannot be emphasized enough as it really gives us a deeper understanding of physics and as this can serve as a starting point to start looking at some of the complications that play a role on a lithospheric scale (temperature and depth-dependency of viscosity & brittle effects). I would suggest to include the figures of Schmid & co.

4) Open questions Do you consider research on folding a finished topic or is there work to be done for young students? If yes, what are some of those open questions? Can you include a section on that as potential encouragement for young readers?

Minor points:

l. 17: including ON the lithospheric

l. 44: "competent" ->this is a very vague term. Do you mean a layer with a higher viscosity, a layer that has larger stresses if deformed? It would be good to explain how you define competent

l. 182: remove "easily" -> what is easy is relative for different readers.. Fig. 12: Mention in the figure caption that it was computed with equation 12.

l.262: From looking at figure 12, is is not so clear to me that the thin plate solution does not go to zero as you don't plot things up to zero. Maybe add a blowup in the figure?

l. 275/fig. 13c: Same argument here; I don't see the oscillating behavior on the plots. Can you add a zoom-in of that in the figure ? Also, is this oscillating behavior reproduced with numerical models or is it an artifact of the analytics? A small discussion on the underlying causes would be appreciated.

l. 310/fig. 15: Explain better in the figure caption what the colored lines represent.

l. 664: "A reasonable value" -> I suppose that you think it is reasonable as it is similar in magnitude as ridge push forces; please explain add that here.

l. 997: How do you know that the analytical solution provides "reasonable accurate results"? Numerics?

Boris Kaus, Mainz

Additional note:

I published quite a few papers with the first author, including recent ones, which is something that I pointed out to the editors of SE before accepting to do this review. In SE there are currently no official guidelines regarding this, and as the review process is open anyone can comment on it. I do not feel that this has biased my review, but do want to point it out.

---

## Editor Comment (EC1) · S. Buiter (Editor) · 4 Aug 2016

This is not a formal review or editorial recommendation, but just a couple of minor suggestions to the authors for consideration while preparing their revisions.

Section 2.1.1 on single-layer folding is (understandably) pretty long. Please consider some further subdivision (for example, according to rheology and/or analysis method?).

3D viscous folding is briefly introduced at the end of section 2.1.1. How well do 2D studies in general do in approximating 3D settings? Are we fine to first order with 2D or may we be oversimplifying essential components?

Symbol 'm' is used for number of layers at line 527 and for half the number of layers at

line 552, please make consistent.

Some symbols are used for different parameters. I understand this may be difficult to avoid seeing the large number of equations, but could you have a look? E.g., 'b' in equation 26 and line 590, 'm' on lines 527/552 and 935

Clearly some kind of selection of the vast literature of experimental results needed to be made to keep the review practical. But could you say how you chose the studies to discuss? (section 2.2)

Sections 2.2 and 3.2: Would it be possible to add a brief summary statement that discusses what the experiments have contributed to the theoretical analyses, that is, how they have brought understanding of folding and necking further?

Section 3.1.3, lines 1046-1053 has repetitions, necking is defined twice, as is the width of the necking zone

Figures:

(a) Figs. 2, 5, 6: Please attribute the photographs; (b) Fig. 2C: Add a scale; (c) Fig. 4: Please add the size of the domain, the size of the initial perturbation, the numerical method that was used, and that the results were computed for this study (I assume); (d) Figs. 12, 13, 17 and 18: Please increase the colour contrast between the black and blue lines (or dash one); (e) Fig. 14: Add also symbols to the axes and in the caption to agree with type of labelling in previous figs 12 and 13; (f) Fig 17: Does this figure need values for A0, S0 and eta_m?

---

## Author Comment (AC1) · 22 Aug 2016

We thank Susanne Buiter for her helpful comments.

Section 2.1.1 on single-layer folding is (understandably) pretty long. Please consider some

further subdivision (for example, according to rheology and/or analysis method?).

More than three levels of subdivisions are not allowed in Solid Earth. We hence keep the original

subdivision.

3D viscous folding is briefly introduced at the end of section 2.1.1. How well do 2D studies in

general do in approximating 3D settings? Are we fine to first order with 2D or may we be

oversimplifying essential components?

3D folding: The answer depends on the fold geometry. Many natural folds seem to have a

cylindrical structure, that is, their lateral extend along the fold axis is considerably larger than

their wavelength. For such cylindrical fold shapes 2D solutions are applicable. However, other

natural folds are clearly not cylindrical (such as dome and basin fold shapes) or have been

formed by two consecutive events of deformation. For such fold shapes, 3D models are required.

Symbol 'm' is used for number of layers at line 527 and for half the number of layers at

Meaning of symbol $m$ has been made consistent.

line 552, please make consistent.

Some symbols are used for different parameters. I understand this may be difficult to avoid

seeing the large number of equations, but could you have a look? E.g., 'b' in equation 26 and line

590, 'm' on lines 527/552 and 935

We replaced $b$ in line 590 with $H_m$. We also replaced $m$ in line 935ff with $r$.

Clearly some kind of selection of the vast literature of experimental results needed to be made to

keep the review practical. But could you say how you chose the studies to discuss? (section 2.2)

The sentence below has been added in the introductory paragraph to give a rationale for our selection – although the selection must remain personal.

"Only a limited personal selection of the many published studies can be presented here but, in keeping with the overall theme of the review, we particularly highlight those studies that either specifically constrained analytical solutions or attempted to extend them to higher amplitudes typical of natural geometries."

Sections 2.2 and 3.2: Would it be possible to add a brief summary statement that discusses what the experiments have contributed to the theoretical analyses, that is, how they have brought understanding of folding and necking further?

We think the short answer to this is actually already given in the sentence we have added in response to the query above as to why we selected particular experimental studies to include in this review – they provided the necessary constraint and extension to higher amplitudes (until this could be done efficiently numerically – now this can be done on most desktop computers but that was not the case until quite recently).

Section 3.1.3, lines 1046-1053 has repetitions, necking is defined twice, as is the width of the necking zone

We removed the repetitions.

Figures:

(a) Figs. 2, 5, 6: Please attribute the photographs; (b) Fig. 2C: Add a scale; (c) Fig. 4: Please add the size of the domain, the size of the initial perturbation, the numerical method that was used, and that the results were computed for this study (I assume);

(d) Figs. 12, 13, 17 and 18: Please increase the colour contrast between the black and blue lines (or dash one); (e) Fig. 14: Add also symbols to the axes and in the caption to agree with type of labelling in previous figs 12 and 13; (f) Fig 17: Does this figure need values for A0, S0 and eta_m?

We attributed the photographs in the captions.

Fig. 2C has a scale; it is a coin above the calcite layer in the middle-left of the photo.

For Fig. 4 the requested information have been given.

Color contrast: Figs. 12, 13, 17 and 18 have been modified.

Fig. 14: symbols have been added.

Fig. 17: Caption has been elaborated.

Stefan Schmalholz and Neil Mancktelow

---

## Author Comment (AC2) · 22 Aug 2016

We thank Peter Hudleston for his helpful and constructive review.

We shortened the text concerning the work of Neurath and Smith to some extent.

We clarified our usage of the terms necking and pinch-and-swell at the beginning of the introduction following the reviewer's suggestion.

We define now competent (i.e. one having a higher mechanical strength or greater resistance to deformation) and mention that necking can also occur for non-linear rheologies other than power-law.

We corrected the misspelling and references.

We included a new figure which visualises the dependence of the dominant wavelength and maximal amplification rate on the spacing of the competent layer. The results are from Schmid and Podladchikov (2006) for embedded, linear viscous multilayers which we consider the most applicable configuration for multi-layered rocks for which approximate analytical solutions exist. For example, the dominant wavelength solution of Biot displayed in our Eqn. (33) indicates that the dominant wavelength increases continuously with increasing layer spacing (because the parameter a goes to 1 but the total thickness of the multilayer increases) and hence this formula does not capture the "real single layer" mode of Schmid and Podladchikov (2006). If the spacing is larger than the dominant wavelength of the individual layers, then the individual layers will develop the single layer dominant wavelength independent of a continuously increasing spacing.

Specific comments

Line 402. Done.

Line 412. We modified the text.

Line 569. We added the reference.

Line 627. We added two references.

Line 810-815. We would be very happy to cite Ramsay and Huber (1987) on this point but we are not sure what figure is being referred to.

Line 1000. We modified the text to make this clear.

Line 1004. The decrease in amplification rate for necking of layers embedded in a viscous matrix is rather caused by the increasing shear resistance of the embedding matrix around the necking zone. If the layer is not embedded in a matrix (free plate) then the amplification rates are actually increasing with progressive necking which agrees

with analytical solutions for necking of free plates (see e.g. Fig. 9 in Schmalholz et al., Journal of Structural Geology, 2008). We added a sentence to explain this.

Line 1273. We added a sentence stating that the authors (Adamuszek et al., Terra Nova, 2013) applied their theory to a folded sequence of alternating nodular limestone and shale.

Figure 2. Done.

Stefan Schmalholz and Neil Mancktelow

---

## Author Comment (AC3) · 22 Aug 2016

We thank Boris Kaus for his helpful and constructive review.

We agree with Boris Kaus concerning his points 1) and 2) and we added a paragraph on the folding of Mohr-Coulomb layers and on crustal folding and fold-and-thrust belts. We also included the mentioned references. However, we kept it short because the review is already very long.

We modified the discussion of the LAF model.

We mentioned already the future challenges in the last paragraph of the summary and conclusions chapter (Lines 1441-1450). We extended this part a bit.

[Figure]

Minor points

Line 17. Done.

Line 44. Done.

Line 182. Done.

Line 262. We removed the statement about the "amplification rate not going to zero", because this statement applies actually to the variation of the maximal amplification rate with decreasing viscosity ratio and not to a variation of the amplification rate with varying ratio of wavelength to thickness as displayed in Fig. 12. Furthermore, the statement that the maximal amplification rate of the thin-plate approach does not tend to zero for a viscosity ratio of 1 is already given in line 203-205.

Line 275/Fig. 13c: We added a zoom in Fig. 13c and d. The numerical results show that the oscillation (i.e. negative amplification rates) disappear with finite amplitudes. We modified the text.

Fig. 15. Done.

Line 664. Done.

Line 997. Yes, because of comparison with numerical results. We modified the text.

Stefan Schmalholz and Neil Mancktelow

---

## Author Comment (AC4) · 22 Aug 2016

We thank Dani Schmid, Marta Adamuszek and Marcin Dabrowski for their short comment (referred to here as Schmid et al., 2016) which will help us to clarify certain statements concerning their theoretical model for Large Amplitude Folding (LAF; Adamuszek et al., 2013) during the revision of our manuscript.

The aim of our statement (line 454 on) concerning the "not dramatic" improvement of the LAF model with respect to the Exponential Single Waveform Solution (ESWS) of Biot et al. (1961) displayed in our Eqn. (27) (and in this reply again in Eqn. (1)) was not to discredit the LAF model, but to underline the usefulness and strength of the simple ESWS. Therefore, this statement came before the description of the LAF model, but we agree that the statement can be misunderstood and we will modify the text. The LAF model is important for understanding finite fold amplification, because it combines three major features of fold amplification: (i) the wavelength dependent amplification rates based on the stability analysis, (ii) the shortening of waveform components and the variation of the corresponding amplification rates ("preferred wavelength"), and (iii) the difference between the shortening rate of the layer's arc length and the bulk shortening rate. The LAF model is described by a coupled system of ordinary differential equations (time derivatives) and is hence considerably simpler than a full two-dimensional (2D) fluid mechanics model described by a system of partial differential equations which has to be solved numerically by, for example, the finite element method.

As clear in our original Fig.16 and highlighted in the Fig. 1 in Schmid et al. (2016), for the geometry and rheology taken as an example, the simple ESWS reproduces the finite fold geometry of the amplified bell-shaped initial perturbation rather well up to a shortening value of ca. 20% (here ~25° limb dip), but the fit then rapidly worsens. The LAF model continues to

provide a better approximation up to ca. 25% shortening (~45° limb dip), but even at this stage the predicted fold shape does differ noticeably from the numerical model. If accuracy is to be maintained, the transition must be made at some stage to two-dimensional (2D) numerical modelling: in the case of the simple ESWS at ca. 20% shortening, for the significantly more complicated LAF solution at around 23-25%. For the example we presented in Fig. 16, this was the basis of our statement that in this case the improvement was "not dramatic". By recently providing the Folder package (Adamuszek et al., 2016), the authors have made the switch to 2D numerical modelling of such an isolated perturbation (or indeed any layer shape) very easy and straightforward, so that the transition from (semi-)analytical to numerical has never been easier.

A major aim of our review is to focus on simple analytical solutions and to show that these simple solutions are extremely useful to get fundamental insight into folding mechanics and to make first order estimates. The ESWS of Biot et al. (1961) is a simple and comprehensible solution which can accurately predict fold amplification up to limb dips of around 25°. At such limb dips the general final fold shape (e.g. localized or regular, symmetric or asymmetric etc.) can be anticipated. If the Fourier transform of the initial geometrical perturbation of a layer is known, then the fold amplification can simply be calculated with the ESWS by modifying this Fourier transform. For an initial bell-shaped function the fold shape (represented here by the vertical coordinate of the central line in the folding layer, $y$) can be calculated by the ESWS equation:

$$y(x,t) = ab \int_0^\infty \exp\left( \underbrace{\alpha t}_{\substack{\text{exponential} \\ \text{time evolution}}} - ak \right) \cos(kx)\, dk \tag{1}$$

The fold shape given by $y(x,t)$ can be calculated for any time with Eqn. (1) by simply including an exponential growth term, $\alpha t$ ($\alpha$ is amplification rate and $t$ is time), into the Fourier

transform of the initial perturbation which is given by Eqn. (1) for $\alpha t = 0$. The ESWS is described by a single and simple equation and we hence consider the ESWS as a comprehensible (or transparent) solution which provides fundamental insight into fold shape evolution. The LAF solution is more accurate than the ESWS, but in turn the LAF model is much less transparent because it is described by a coupled system of ordinary differential equations for which the time evolution must be calculated numerically.

We have a similar reply concerning the comparison of the LAF solution with the Finite Amplitude Solution (FAS) of Schmalholz and Podladchikov (2000) which is displayed in our Eqn. (29) and is given here again:

$$\frac{L_0}{L} = \underbrace{\left(\frac{A}{L}\frac{L_0}{A_0}\right)^{\frac{1}{2+\alpha_d}}}_{\text{Exponential solution}} \underbrace{\left(\frac{S}{L}\frac{L_0}{S_0}\right)^{\frac{\alpha_d}{2+\alpha_d}}}_{\substack{\text{Finite amplitude} \\ \text{correction}}} \quad \text{with} \quad S = L + \pi^2 \frac{A^2}{L} \tag{2}$$

The first term on the right hand side including amplitude, $A$ (subscript 0 indicates initial values), and wavelength, $L$, is the classical exponential solution (e.g. Johnson and Fletcher, 1994, also showed such "power-law version" of the exponential solution; $\alpha_d$ is the dominant amplification rate) and the second term represents the finite amplitude correction due to the change of the arc length, $S$, with respect to the wavelength, $L$. The FAS is an analytical solution and provides an algebraic relationship between shortening (quantified by $L_0 / L$) and the fold amplification (quantified by $A / L$). The FAS provides hence fundamental insight into finite fold amplification and can be used, for example, to analytically derive algebraic expressions to estimate the limb dip at which the exponential solution breaks down (see our review). Again, the LAF model provides more accurate results than the FAS but the finite amplitude evolution is much more transparent from Eqn. (2) than from the ordinary differential equations of the LAF model.

Simple analytical solutions provide fundamental insight into fold amplification and are useful to make first order estimates. If accurate fold shape calculations are needed, then 2D (or 3D) numerical simulations are required. Between the simple and approximate analytical solutions and the accurate numerical simulations exist several mathematical models of "intermediate accuracy" such as the LAF model or the third-order analysis presented in Johnson and Fletcher (1994). All models have their justifications, can reveal different aspects of fold amplification and are essential for a thorough understanding of fold amplification. Nevertheless, the computational power has increased dramatically in the last decades. The best example is the free software Folder by Adamuszek et al. (2016) which enables 2D finite element simulations of folding and necking, and also the accurate, finite element based calculation of amplification rates for nearly any model configuration on a standard laptop within few seconds to several minutes. Therefore, if accurate fold shapes or amplification rates must be calculated, then 2D or 3D numerical simulations are most useful, but when fundamental insights and simple first order predictions are needed, then the simple analytical solutions are most useful.

Stefan Schmalholz and Neil Mancktelow

**References**

Adamuszek, M., Dabrowski, M., and Schmid, D. W.: Folder: A numerical tool to simulate the development of structures in layered media, Journal of Structural Geology, 84, 85-101, 2016.

Adamuszek, M., Schmid, D. W., and Dabrowski, M.: Theoretical analysis of large amplitude folding of a single viscous layer, Journal of Structural Geology, 48, 137-152, 2013.

Biot, M. A., Ode, H., and Roever, W. L.: Experimental verification of the theory of folding of stratified viscoelastic media, Geo. Soc. Am. Bull., 72, 1621-1632, 1961.

Johnson, A. M. and Fletcher, R. C.: Folding of viscous layers, Columbia University Press, New York, 1994.

Schmalholz, S. M. and Podladchikov, Y. Y.: Finite amplitude folding: transition from exponential to layer length controlled growth, Earth and Planetary Science Letters, 181, 617-633, 2000.

Schmid, D. W., Adamuszek, M. and Dabrowski, M.: Interactive comment on Solid Earth Discuss., doi:10.5194/se-2016-80, 2016.